# Stable Velocity: A Variance Perspective on Flow Matching

**Donglin Yang** [† 1]  **Yongxing Zhang** [2 3]  **Xin Yu** [1]  **Liang Hou** [4]  **Xin Tao** [4]  **Pengfei Wan** [4]  **Xiaojuan Qi** [‡ 1]
**Renjie Liao** [‡ 2 3 5]

## Abstract

While flow matching is elegant, its reliance on single-sample conditional velocities leads to high-variance training targets that destabilize optimization and slow convergence. By explicitly characterizing this variance, we identify 1) a *high-variance regime* near the prior, where optimization is challenging, and 2) a *low-variance regime* near the data distribution, where conditional and marginal velocities nearly coincide. Leveraging this insight, we propose **Stable Velocity**, a unified framework that improves both training and sampling. For training, we introduce Stable Velocity Matching (StableVM), an unbiased variance-reduction objective, along with Variance-Aware Representation Alignment (VA-REPA), which adaptively strengthen auxiliary supervision in the *low-variance regime*. For inference, we show that dynamics in the *low-variance regime* admit closed-form simplifications, enabling Stable Velocity Sampling (StableVS), a finetuning-free acceleration. Extensive experiments on ImageNet $256 \times 256$ and large pretrained text-to-image and text-to-video models, including SD3.5, Flux, Qwen-Image, and Wan2.2, demonstrate consistent improvements in training efficiency and more than $2\times$ faster sampling within the *low-variance regime* without degrading sample quality. Our code is available at https://github.com/linYDTHU/StableVelocity.

## 1. Introduction

Recent years have seen major advances in generative modeling driven by diffusion (Sohl-Dickstein et al., 2015; Ho et al., 2020; Song et al., 2020; Karras et al., 2022b), flow matching (Lipman et al., 2022; Liu et al., 2023), and stochastic interpolants (Albergo et al., 2025; Ma et al., 2024; Yu et al., 2024). These approaches transform a simple prior distribution, *e.g.*, standard normal, into a complex data distribution through a stochastic or deterministic dynamic process, admitting a unified formulation that has enabled scalable training and state-of-the-art performance across image generation (Labs, 2024; Esser et al., 2024; Wu et al., 2025a) and restoration (Rombach et al., 2022; Lin et al., 2024), and video generation (Brooks et al., 2024; Wan et al., 2025).

Among these approaches, Conditional Flow Matching (CFM) (Tong et al., 2023) provides an elegant objective for learning the probability flow without explicitly simulating the forward SDE or PF-ODE. By training a neural network to predict conditional velocity fields, CFM enjoys both theoretical guarantees and practical scalability.

Despite its elegance, CFM suffers from a fundamental yet underexplored limitation: the variance of its training target. In practice, the conditional velocity $v_t(x_t \mid x_0)$ is a single-sample Monte Carlo estimate of the true marginal velocity $v_t(x_t)$, which can exhibit high variance, particularly at timesteps where the marginal distribution remains close to the prior. Such high-variance targets not only slow convergence, but also induce a mismatch between the empirical optimization dynamics and the ideal population objective. While prior work has empirically observed variance-related inefficiencies in diffusion training (Karras et al., 2022a; Choi et al., 2022; Xu et al., 2023), a principled variance-theoretic understanding within flow matching and stochastic interpolants has been largely missing.

In this work, we develop a *variance-based perspective* on stochastic interpolants. By explicitly characterizing the variance of conditional velocity targets, we reveal a two-regime structure that governs both training and inference: a *high-variance regime* near the prior, where optimization is inherently noisy, and a *low-variance regime* near the data distribution, where conditional and marginal velocities coincide. This insight naturally leads to the **Stable Velocity** frame-

---

† This work was conducted during the author's internship at Kling Team. ‡ Project Lead. [1]University of Hong Kong, Hong Kong [2]University of British Columbia, Canada [3]Vector Institute for AI, Toronto, Canada [4]Kling Team, Kuaishou Technology [5]Canada CIFAR AI Chair. Correspondence to: Xiaojuan Qi <xjqi@eee.hku.hk>, Renjie Liao <rjliao@ece.ubc.ca>.

work. For training, we introduce Stable Velocity Matching (StableVM), an unbiased variance-reduction objective, together with Variance-Aware Representation Alignment (VA-REPA), which adaptively strengthen auxiliary supervision when the variance is low. For inference, we show that dynamics in the low-variance regime admit closed-form simplifications, enabling finetuning-free acceleration via Stable Velocity Sampling (StableVS).

We validate our proposed approach on ImageNet $256 \times 256$ and standard text-to-image and text-to-video benchmarks using state-of-the-art pretrained models. StableVM and VA-REPA consistently outperform REPA (Yu et al., 2024) across a wide range of model scales and training variants, including REG (Wu et al., 2025b) and iREPA (Singh et al., 2025) (Sec. 4.2). Meanwhile, StableVS achieves more than $2\times$ inference acceleration in the *low-variance regime* for recent flow-based models, such as SD3.5 (Esser et al., 2024), Flux (Labs, 2024), Qwen-Image (Wu et al., 2025a), and Wan2.2 (Wan et al., 2025), without perceptible degradation in sample quality (Sec. 4.3).

## 2. Variance Analysis of Flow Matching

We first briefly review flow matching and stochastic interpolants. Details as well as their connections to score-based diffusion models are deferred to Appendix A.

**Flow Matching and Stochastic Interpolants.** Given samples from an unknown data distribution $q(\boldsymbol{x}_0)$ over $\mathbb{R}^d$, flow matching (Lipman et al., 2022; Liu et al., 2023; Lipman et al., 2024a) and stochastic interpolants (Albergo et al., 2025; Albergo & Vanden-Eijnden, 2023) define a continuous-time corruption process

$$\boldsymbol{x}_t = \alpha_t \boldsymbol{x}_0 + \sigma_t \boldsymbol{\varepsilon}, \quad \boldsymbol{\varepsilon} \sim \mathcal{N}(0, \boldsymbol{I}), \quad (1)$$

where $\alpha_t$ and $\sigma_t$ are differentiable functions satisfying $\alpha_t^2 + \sigma_t^2 > 0$ for all $t \in [0,1]$, with boundary conditions $\alpha_0 = \sigma_1 = 1$ and $\alpha_1 = \sigma_0 = 0$. Most works (Ma et al., 2024; Yu et al., 2024; Lipman et al., 2022; Liu et al., 2023; Lipman et al., 2024a) adopt a simple linear interpolant $\alpha_t = 1 - t$, $\sigma_t = t$. This process induces a conditional velocity field

$$\boldsymbol{v}_t(\boldsymbol{x}_t \mid \boldsymbol{x}_0) = \frac{\sigma_t'}{\sigma_t}(\boldsymbol{x}_t - \alpha_t \boldsymbol{x}_0) + \alpha_t' \boldsymbol{x}_0, \quad (2)$$

where $\alpha_t'$ and $\sigma_t'$ denote the time derivative. The corresponding marginal velocity field

$$\boldsymbol{v}_t(\boldsymbol{x}_t) = \mathbb{E}_{p_t(\boldsymbol{x}_0 \mid \boldsymbol{x}_t)}[\boldsymbol{v}_t(\boldsymbol{x}_t \mid \boldsymbol{x}_0)]. \quad (3)$$

**Conditional Flow Matching (CFM).** Training typically uses the CFM objective (Tong et al., 2023):

$$\min_{\boldsymbol{\theta}} \ \mathbb{E}_{t, q(\boldsymbol{x}_0), p_t(\boldsymbol{x}_t \mid \boldsymbol{x}_0)} \lambda_t \left\| \boldsymbol{v}_{\boldsymbol{\theta}}(\boldsymbol{x}_t, t) - \boldsymbol{v}_t(\boldsymbol{x}_t \mid \boldsymbol{x}_0) \right\|^2, \quad (4)$$

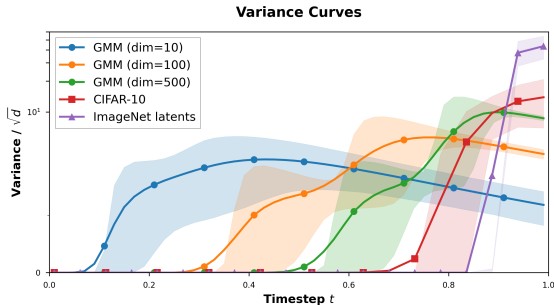

*Figure 1.* **Variance curves of $\mathcal{V}_{\mathrm{CFM}}(t)$ with 15%–85% quantile bands.** Evaluated on GMMs of varying dimensionality, CIFAR-10 images, and $256 \times 256$ ImageNet latents obtained by the Stable Diffusion VAE. The $y$-axis reports $\mathcal{V}_{\mathrm{CFM}}(t)$ normalized by the square root of the data dimension. See Appendix F.2 for details.

where $\lambda_t$ is a positive weighting function, and $\boldsymbol{v}_{\boldsymbol{\theta}}(\cdot, \cdot) :$ $[0, 1] \times \mathbb{R}^d \to \mathbb{R}^d$ is a neural velocity field parameterized by $\boldsymbol{\theta}$. Here, the conditional path distribution is $p_t(\boldsymbol{x}_t \mid \boldsymbol{x}_0) = \mathcal{N}(\boldsymbol{x}_t \mid \alpha_t \boldsymbol{x}_0, \sigma_t^2 \boldsymbol{I})$. The minimizer of Eq. (4) is provably the true marginal velocity field (Tong et al., 2023; Xu et al., 2023; Ma et al., 2024):

$$\boldsymbol{v}_{\boldsymbol{\theta}}^*(\boldsymbol{x}_t, t) = \mathbb{E}_{p_t(\boldsymbol{x}_0 \mid \boldsymbol{x}_t)}[\boldsymbol{v}_t(\boldsymbol{x}_t \mid \boldsymbol{x}_0)] = \boldsymbol{v}_t(\boldsymbol{x}_t). \quad (5)$$

**Variance of CFM.** Although unbiased, the CFM target is only a single-sample Monte Carlo estimator of Eq. (3), which can exhibit high variance (Owen, 2013; Elvira & Martino, 2021). Following Xu et al. (2023), we quantify this variance by the average trace of the conditional velocity covariance at time $t$:

$$\mathcal{V}_{\mathrm{CFM}}(t) = \mathbb{E}_{p_t(\boldsymbol{x}_t)}\big[\mathrm{Tr}\big(\mathrm{Cov}_{p_t(\boldsymbol{x}_0 \mid \boldsymbol{x}_t)}\big(\boldsymbol{v}_t(\boldsymbol{x}_t \mid \boldsymbol{x}_0)\big)\big)\big]$$
$$= \mathbb{E}_{q(\boldsymbol{x}_0), p_t(\boldsymbol{x}_t \mid \boldsymbol{x}_0)}\big[\|\boldsymbol{v}_t(\boldsymbol{x}_t \mid \boldsymbol{x}_0) - \boldsymbol{v}_t(\boldsymbol{x}_t)\|^2\big]. \quad (6)$$

To characterize the behavior of $\mathcal{V}_{\mathrm{CFM}}(t)$, we evaluate it on Gaussian mixture models (GMMs), CIFAR-10 (Krizhevsky, 2009), and ImageNet latent codes produced by the pretrained Stable Diffusion VAE (Rombach et al., 2022). As shown in Fig. 1, two consistent patterns emerge:

- **Low- vs. high-variance regimes.** $\mathcal{V}_{\mathrm{CFM}}(t)$ remains close to zero at small $t$ but increases rapidly as $t$ grows, naturally separating the process into a *low-variance regime* ($0 \leq t < \xi$) and a *high-variance regime* ($\xi \leq t \leq 1$).

- **Effect of dimensionality.** As data dimensionality increases, the split point $\xi$ shifts toward 1, enlarging the low-variance regime while also increasing the overall variance magnitude.

Fig. 2 illustrates the two regimes: in the *low-variance regime* the posterior concentrates on a single reference sample,

while in the *high-variance regime* it spreads over multiple samples, leading to large variance. Increasing dimensionality delays this mixing effect and shifts the split point $\xi$ closer to 1. Although the exact split point $\xi$ depends on the unknown data distribution, our variance analysis suggests a clear dimensionality-dependent trend, which provides practical guidance for choosing $\xi$ in real-world settings.

In summary, these observations naturally suggest two key questions: (1) How can we reduce training variance in the high-variance regime without altering the global minimizer? (2) How can the low-variance regime be exploited for stronger supervision and faster sampling?

# 3. Variance-Driven Optimization of Training and Sampling

In this section, we address the two questions posed in Sec. 2 from a unified, variance-driven perspective.

## 3.1. Stable Velocity Matching

We propose Stable Velocity Matching (StableVM), a variance-reduced yet unbiased alternative to CFM. The key idea is to replace the single-sample conditional velocity target with a multi-sample, self-normalized aggregation over reference data points under the multi-sample conditional path. This reduces training variance while preserving the exact same global minimizer as CFM in Eq. (5).

**Multi-sample conditional path.** Inspired by (Xu et al., 2023), we introduce $n$ reference samples $\{\boldsymbol{x}_0^i\}_{i=1}^n$, drawn *i.i.d.* from the data distribution $q(\boldsymbol{x}_0)$. We then define a composite conditional probability path $p_t^{\text{GMM}}\left(\boldsymbol{x}_t \mid \{\boldsymbol{x}_0^i\}_{i=1}^n\right) := \sum_{i=1}^n \frac{1}{n} p_t(\boldsymbol{x}_t \mid \boldsymbol{x}_0^i)$, which is essentially a mixture of conditional probabilities associated with each reference sample. The posterior under the GMM path admits a simple mixture form (Prop. E.1), and preserves the original marginal path in expectation, a property that is crucial for unbiasedness.

**StableVM target.** Based on the reference samples, we define the StableVM target $\widehat{\boldsymbol{v}}_{\text{StableVM}}$ as the self-normalized importance weighted average of the conditional velocities:

$$\widehat{\boldsymbol{v}}_{\text{StableVM}}(\boldsymbol{x}_t; \{\boldsymbol{x}_0^i\}_{i=0}^n) := \frac{\sum_{k=1}^n p_t(\boldsymbol{x}_t \mid \boldsymbol{x}_0^k) \boldsymbol{v}_t(\boldsymbol{x}_t \mid \boldsymbol{x}_0^k)}{\sum_{j=1}^n p_t(\boldsymbol{x}_t \mid \boldsymbol{x}_0^j)}. \quad (7)$$

Compared to the CFM target $\boldsymbol{v}_t(\boldsymbol{x}_t \mid \boldsymbol{x}_0)$, this can be viewed as a multi-sample Monte Carlo estimator of the same marginal velocity field $\boldsymbol{v}_t(\boldsymbol{x}_t)$.

**Training objective.** We train a neural velocity field $\boldsymbol{v}_\theta(\boldsymbol{x}_t, t)$ by minimizing

$$\begin{aligned} &\mathcal{L}_{\text{StableVM}}(\boldsymbol{\theta}) \\ &= \mathbb{E}_{\substack{t, \{\boldsymbol{x}_0^i\} \sim q^n \\ \boldsymbol{x}_t \sim p_t^{\text{GMM}}}} \left\| \boldsymbol{v}_\theta(\boldsymbol{x}_t, t) - \widehat{\boldsymbol{v}}_{\text{StableVM}}(\boldsymbol{x}_t; \{\boldsymbol{x}_0^i\}_{i=0}^n) \right\|^2. \end{aligned} \quad (8)$$

Our StableVM target is compatible with general stochastic interpolant framework. Under the special case of VP diffusion, it resembles the STF objective (Eq. (23)) in form, but differs fundamentally in how the noisy training input $\boldsymbol{x}_t$ is constructed. STF generates $\boldsymbol{x}_t$ by perturbing a *single* reference sample, i.e., $\boldsymbol{x}_t \sim p_t(\boldsymbol{x}_t \mid \boldsymbol{x}_0^1)$, and then forms a self-normalized weighted target over the remaining references. In contrast, StableVM explicitly samples $\boldsymbol{x}_t$ from a composite conditional path $p_t^{\text{GMM}}$ over the entire reference batch, which yields an unbiased target and extends naturally beyond VP diffusion. Details are provided in Appendix C.

**Unbiasedness and optimality.** The following theorem establishes two key properties of StableVM: it remains unbiased and admits the same global minimizer as CFM.

**Theorem 3.1.** *(a) The StableVM target is unbiased. That is, for any $\boldsymbol{x}_t$, we have*

$$\mathbb{E}_{\{\boldsymbol{x}_0^i\} \sim p_t^{GMM}(\cdot|\boldsymbol{x}_t)} \left[ \widehat{\boldsymbol{v}}_{StableVM}(\boldsymbol{x}_t; \{\boldsymbol{x}_0^i\}_{i=0}^n) \right] = \boldsymbol{v}_t(\boldsymbol{x}_t).$$

*(b) The global minimizer $\boldsymbol{v}^*(\boldsymbol{x}_t, t)$ of the StableVM objective $\mathcal{L}_{StableVM}$ is the true velocity field $\boldsymbol{v}_t(\boldsymbol{x}_t)$.*

Proofs are provided in Appendix E.2.

**Variance of StableVM.** While remaining unbiased, StableVM strictly reduces the variance of the training target. Following Eq. (6), we derive the average trace-of-covariance

$$\begin{aligned} &\mathcal{V}_{\text{StableVM}}(t) \\ &= \mathbb{E}_{\boldsymbol{x}_t \sim p_t} \left[ \text{Tr} \left( \text{Cov}_{\{\boldsymbol{x}_0^i\} \sim p_t^{\text{GMM}}} \left( \widehat{\boldsymbol{v}}_{\text{StableVM}} \right) \right) \right] \\ &= \mathbb{E}_{\substack{\boldsymbol{x}_t \sim p_t \\ \{\boldsymbol{x}_0^i\} \sim p_t^{\text{GMM}}}} \left\| \boldsymbol{v}_t(\boldsymbol{x}_t) - \widehat{\boldsymbol{v}}_{\text{StableVM}}(\boldsymbol{x}_t; \{\boldsymbol{x}_0^i\}_{i=0}^n) \right\|^2. \end{aligned} \quad (9)$$

**Theorem 3.2.** *Fix $t \in [0, 1]$. Assume that the conditional velocity field is affine and the event $\{\boldsymbol{x}_0^1 = \cdots = \boldsymbol{x}_0^n\}$ has probability 0. Then we have $\mathcal{V}_{StableVM}(t) < \mathcal{V}_{CFM}(t)$ strictly.*

In fact, we can prove the following stronger variance bound stating that the variance decays in the rate of $O(1/n)$. Proofs are provided in Appendix E.3.

**Theorem 3.3.** *Fix $t \in [0, 1]$. Assume $\left( (n-1) \| \widehat{\boldsymbol{v}}_t - \boldsymbol{v}_t(\boldsymbol{x}_t) \|^2 \right)_{n=1}^\infty$ is uniformly integrable. Define $r_t(\boldsymbol{x}_t, \boldsymbol{x}_0) := \frac{p_t(\boldsymbol{x}_0|\boldsymbol{x}_t)}{q(\boldsymbol{x}_0)}$ and $\Delta_t(\boldsymbol{x}_t, \boldsymbol{x}_0) := \| \boldsymbol{v}_t(\boldsymbol{x}_t \mid \boldsymbol{x}_0) - \boldsymbol{v}_t(\boldsymbol{x}_t) \|^2$.*

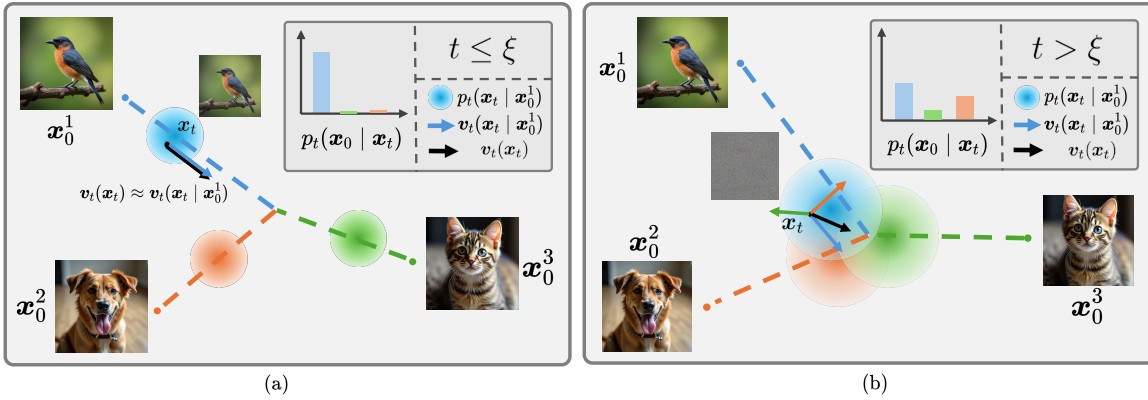

Figure 2. **Illustration of CFM variance** $\mathcal{V}_{\textbf{CFM}}(t)$. (a) The *low-variance regime* ($t \leq \xi$), where the posterior $p_t(\boldsymbol{x}_0 \mid \boldsymbol{x}_t)$ is sharply concentrated and the conditional velocity $\boldsymbol{v}_t(\boldsymbol{x}_t \mid \boldsymbol{x}_0)$ nearly coincides with the true velocity $\boldsymbol{v}_t(\boldsymbol{x}_t)$, yielding $\mathcal{V}_{\text{CFM}}(t) \approx 0$. (b) The *high-variance regime* ($t > \xi$), the posterior spreads over multiple reference samples, causing the conditional velocity to fluctuate and resulting in a large $\mathcal{V}_{\text{CFM}}(t)$.

*Then, for large enough n, we have*

$$
\mathcal{V}_{StableVM}(t) = \frac{1}{n-1}\Bigg(\mathcal{V}_{CFM}(t)
$$

$$
+ \mathop{\mathbb{E}}_{\substack{\boldsymbol{x}_0 \sim q \\ \boldsymbol{x}_t \sim p_t(\cdot|\boldsymbol{x}_0)}} \Big[ \big( r_t(\boldsymbol{x}_t, \boldsymbol{x}_0) - 1 \big) \Delta_t(\boldsymbol{x}_t, \boldsymbol{x}_0) \Big] \Bigg) + o\Big(\frac{1}{n}\Big).
\tag{10}
$$

The training procedure is summarized in Alg. 1.

**Extension to Class-Conditional Generation with Classifier-Free Guidance.** Extending StableVM to conditional generation introduces sparsity challenges, as only a small subset of reference samples may match a given label or prompt. To address this, we maintain a class-conditional memory bank of capacity $K$, pre-populated from the training dataset and updated using a FIFO policy. This allows StableVM to construct the mixture-based noisy input and target field using a sufficiently large and diverse reference set even when per-batch class frequency is low. Crucially, this mechanism preserves unbiasedness, as all references are drawn from the true data distribution, while effectively amortizing reference sampling over time. The full algorithm is provided in Alg. 2.

### 3.2. Variance-Aware Representation Alignment

Recent work on representation alignment (REPA) (Yu et al., 2024) and its variants (Leng et al., 2025; Wu et al., 2025b; Singh et al., 2025) demonstrates that auxiliary semantic supervision can substantially accelerate the training of diffusion transformers (Peebles & Xie, 2023a; Ma et al., 2024). From a variance-regime perspective, we empirically find that the effectiveness of REPA largely arises when applied in the *low-variance regime*.

As shown in Sec. 2, the generative process decomposes into low- and high-variance regimes. In the *low-variance regime*, $\boldsymbol{x}_t$ remains strongly coupled to $\boldsymbol{x}_0$, preserving semantic information and making representation alignment well-conditioned. In contrast, in the *high-variance regime* near pure noise, $\boldsymbol{x}_t$ carries little information about $\boldsymbol{x}_0$, and the posterior $p(\boldsymbol{x}_0 \mid \boldsymbol{x}_t)$ becomes highly multimodal, rendering deterministic alignment ill-posed.

Fig. 3 empirically illustrates this regime dependence. When evaluated on a well-trained model, the per-timestep representation-alignment loss remains low and learnable in the low-variance regime, but saturates at high values in the high-variance regime. Consistently, restricting representation alignment to early timesteps yields substantially better FID than applying it uniformly, while applying it only in the *high-variance regime* provides negligible benefit.

Motivated by these observations, we propose variance-aware representation alignment (VA-REPA): semantic alignment should be applied selectively in the *low-variance regime* where the supervision signal is informative. This principle is independent of the specific representation or loss formulation and applies uniformly to REPA and its variants.

Let $\ell_{\text{RA}}(\boldsymbol{x}_t)$ denote a per-sample representation-alignment loss. We introduce a nonnegative weighting function $w(t) \in [0, 1]$ and define the overall training objective as

$$
\mathcal{L} = \mathcal{L}_{\text{StableVM}} + \lambda_{\text{RA}} \frac{\mathbb{E}_{t, \boldsymbol{x}_t}[w(t)\, \ell_{\text{RA}}(\boldsymbol{x}_t)]}{\mathbb{E}_t[w(t)]}.
\tag{11}
$$

The normalization by $\mathbb{E}_t[w(t)]$ ensures that the alignment term is scaled by the number of *effective samples*, preventing vanishing gradients when most samples fall in the *high-variance regime*.

**Weighting functions.** We consider three weighting schemes to modulate the alignment objective: (i) a hard threshold $w_{\text{hard}}(t) = \mathbb{I}[t < \xi]$ for explicit ablation; (ii) a sigmoid relax-

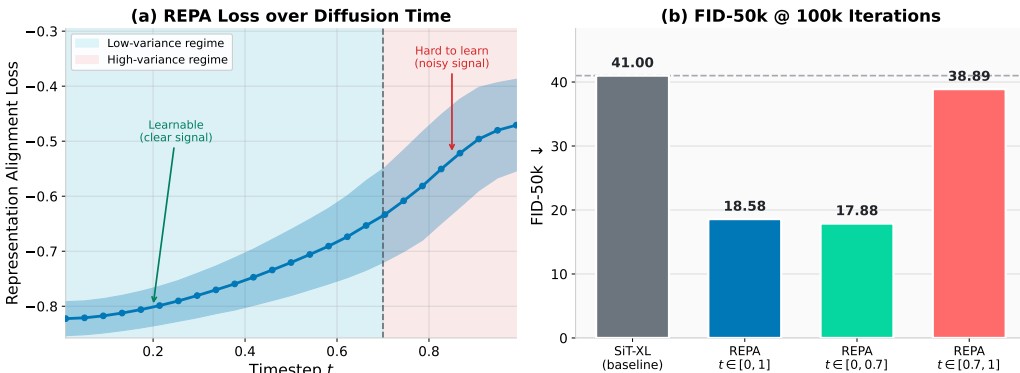

*Figure 3.* **Motivation for variance-aware representation alignment.** **(a)** In the *low-variance regime*, the alignment loss remains consistently low on a pretrained model from REPA (Yu et al., 2024), indicating a learnable and informative supervision signal. In contrast, in the *high-variance regime*, the loss stays high, reflecting the ill-posed nature of semantic recovery from noise. **(b)** Restricting representation alignment to the *low-variance regime* yields the best FID, while applying it only in the *high-variance regime* provides minimal meaningful improvement over the baseline. These results indicate that representation alignment should be activated adaptively rather than uniformly along the diffusion trajectory.

ation $w_{\text{sigmoid}}(t) = \sigma(k(\xi - t))$, where $k$ controls the sharpness of the transition; and (iii) an SNR-based weighting $w_{\text{SNR}}(t) = \frac{\text{SNR}(t)}{\text{SNR}(t)+\text{SNR}(\xi)}$, with $\text{SNR}(t) = \alpha_t^2/\sigma_t^2$, which anchors the midpoint ($w = 0.5$) at $t = \xi$. All weights are applied per sample and normalized within each minibatch.

### 3.3. Stable Velocity Sampling

We introduce Stable Velocity Sampling (StableVS), a principled, finetuning-free acceleration strategy for sampling in the *low-variance regime*. As shown in Sec. 2 (Fig. 2), the conditional variance $\mathcal{V}_{\text{CFM}}(t)$ becomes negligible over a range of early timesteps. In this regime, the instantaneous velocity $\boldsymbol{v}_t(\boldsymbol{x}_t)$ is effectively determined by a single dominant data point $\boldsymbol{x}_0$, allowing the true velocity field to be well approximated by the conditional velocity $\boldsymbol{v}_t(\boldsymbol{x}_t \mid \boldsymbol{x}_0)$. StableVS exploits this structure to enable stable, large-step integration without degrading sample quality, provided the model accurately recovers $\boldsymbol{v}_t(\boldsymbol{x}_t)$.

**StableVS for SDE.** For the reverse SDE (Eq. (17)), we derive the following DDIM-style (Song et al., 2021a) posterior:

$$p_\tau(\boldsymbol{x}_\tau \mid \boldsymbol{x}_t, \boldsymbol{v}_t(\boldsymbol{x}_t)) = \mathcal{N}\left(\boldsymbol{\mu}_{\tau|t},\ \beta_t^2 \mathbf{I}\right), \quad (12)$$

where $\beta_t = f_\beta \sigma_\tau$ and $f_\beta \in [0, 1]$ is a controllable parameter. We then define the noise ratio $\rho_t := \sqrt{(\sigma_\tau^2 - \beta_t^2)/\sigma_t^2}$ and the velocity coupling coefficient $\lambda_t := (\alpha_\tau - \alpha_t\rho_t)/(\alpha_t' - \alpha_t\sigma_t'/\sigma_t)$. The posterior mean $\boldsymbol{\mu}_{\tau|t}$ is then given by:

$$\boldsymbol{\mu}_{\tau|t} = \left(\rho_t - \lambda_t \frac{\sigma_t'}{\sigma_t}\right)\boldsymbol{x}_t + \lambda_t \boldsymbol{v}_t(\boldsymbol{x}_t). \quad (13)$$

The full derivation is provided in Appendix E.4.

**StableVS for ODE.** For the probability flow ODE, we define the integral factor $\Psi_{t,\tau} := \frac{1}{C_t} \int_t^\tau \frac{C(s)}{\sigma_s} \, \mathrm{d}s$, where $C(s) = \alpha_s' - \alpha_s \sigma_s'/\sigma_s$. The exact solution at timestep $\tau$ is:

$$\boldsymbol{x}_\tau = \sigma_\tau \left[\left(\frac{1}{\sigma_t} - \frac{\sigma_t'}{\sigma_t}\Psi_{t,\tau}\right)\boldsymbol{x}_t + \Psi_{t,\tau}\boldsymbol{v}_t(\boldsymbol{x}_t)\right]. \quad (14)$$

The derivation is in Appendix E.5.

In the special case of linear interpolant (*i.e.*, $\alpha_t = 1 - t$, $\sigma_t = t$), setting $\beta_t = 0$ in Eq. (12) makes two samplers coincide:

$$\boldsymbol{x}_\tau = \boldsymbol{x}_t + (\tau - t)\boldsymbol{v}_t(\boldsymbol{x}_t). \quad (15)$$

In the *low-variance regime*, the probability flow trajectory reduces to a straight line with constant velocity, allowing exact integration via Euler steps of arbitrary size.

## 4. Experiments

In this section, we empirically validate the three components of our variance-driven framework. Specifically, we address the following questions:

1. Can StableVM and VA-REPA improve generation performance and training speed? (Tab. 1, 2)

2. Can StableVM and VA-REPA generalize across different training settings? (Tab. 2, 3, 4, Fig. 4)

3. Can StableVS greatly reduce the sampling steps within the *low-variance regime* without performance degradation on pretrained T2I and T2V models? (Tab. 5, 6, Fig. 5)

### 4.1. Setup

**Implementation Details.** Unless otherwise specified, the implementations of StableVS and VA-REPA follow the con-

figuration of REPA (Yu et al., 2024). All models are trained on the ImageNet (Deng et al., 2009) training split. Input images are encoded into latent representations $\mathbf{z} \in \mathbb{R}^{32 \times 32 \times 4}$ using the pre-trained VAE from Stable Diffusion (Rombach et al., 2022). For StableVM and VA-REPA, we set the split point $\xi = 0.7$, the bank capacity $K = 256$, and adopt the sigmoid weighting function $w_{\text{sigmoid}}(t)$. StableVS uses 9 steps in the *low-variance regime*, with $\xi = 0.85$ and $f_\beta = 0$.

**Evaluation.** For StableVM and VA-REPA, we follow the ADM evaluation protocol (Dhariwal & Nichol, 2021). Generation quality is assessed using FID (Heusel et al., 2017), IS (Salimans et al., 2016), sFID (Nash et al., 2021), and precision/recall (Kynkäänniemi et al., 2019), all computed over 50K generated samples. Following Yu et al. (2024), we employ an SDE Euler-Maruyama sampler with 250 steps. For experiments with classifier-free guidance (CFG) (Ho & Salimans, 2022), we use a guidance scale of $w = 1.8$ with interval-based CFG schedule (Kynkäänniemi et al., 2024).

We further evaluate StableVS on standard text-to-image (T2I) and text-to-video (T2V) benchmarks. For T2I, we adopt the *GenEval* benchmark (Ghosh et al., 2023), which comprises 553 prompts spanning 6 categories. Following the official protocol, we generate four samples per prompt using random seeds $\{0, 1000, 2000, 3000\}$. For T2V, we evaluate on *T2V-CompBench* (Sun et al., 2025), which contains 1,400 text prompts covering seven aspects of compositionality. For both tasks, we additionally report reference-based metrics, including PSNR, SSIM, and LPIPS (Zhang et al., 2018), computed with respect to a 30-step baseline.

## 4.2. Evaluation of StableVM and VA-REPA

**Quantitative Evaluation.** We evaluate StableVM and VA-REPA against state-of-the-art latent diffusion transformers under both classifier-free guidance (CFG) and non-CFG settings. Tab. 1 reports CFG results using the SiT-XL backbone for all methods. With only 80 training epochs, our approach achieves the strongest overall performance among all compared methods, outperforming prior REPA-based approaches in both FID and IS, while remaining highly competitive with REPA-E (Leng et al., 2025). Although REPA-E attains a slightly lower FID, the gap is marginal, and our method reaches comparable generation quality at substantially lower training cost. Notably, REPA-E relies on expensive end-to-end fine-tuning of both the autoencoder and the diffusion transformer, whereas our pipeline keeps the autoencoder fixed, resulting in a more modular and computationally efficient training procedure.

Tab. 2 reports results without CFG over multiple SiT architectures and training checkpoints. Across all reported settings, StableVM and VA-REPA consistently improve FID, IS, precision, and recall over vanilla REPA, demonstrating strong scalability and stability with respect to both model

*Table 1.* **Comparison of latent diffusion transformers with CFG.** We compare our method against baselines including MaskDiT (Zheng et al., 2023), DiT-XL/2 (Peebles & Xie, 2023b), SiT-XL/2 (Ma et al., 2024), Faster-DiT (Yao et al., 2024), REPA (Yu et al., 2024), iREPA (Singh et al., 2025), REG (Wu et al., 2025b), and REPA-E (Leng et al., 2025). The first **Ours** block uses the standard REPA sampling protocol, while the second adopts class-balanced sampling (marked with [*]) following REPA-E. Methods marked with [†] require fine-tuning autoencoders.

| Model | Epochs | FID↓ | sFID↓ | IS↑ | Prec.↑ | Rec.↑ |
|---|---|---|---|---|---|---|
| *Latent Diffusion Transformers* | | | | | | |
| MaskDiT | 1600 | 2.28 | 5.67 | 276.6 | 0.80 | 0.61 |
| DiT-XL/2 | 1400 | 2.27 | 4.60 | 278.2 | 0.83 | 0.57 |
| SiT-XL/2 | 1400 | 2.06 | 4.50 | 270.3 | 0.82 | 0.59 |
| Faster-DiT | 400 | 2.03 | 4.63 | 264.0 | 0.81 | 0.60 |
| *Representation Alignment Methods* | | | | | | |
| REPA | 80 | 1.98 | 4.60 | 263.0 | 0.80 | **0.61** |
| | 800 | 1.42 | 4.70 | 305.7 | 0.80 | 0.65 |
| iREPA | 80 | 1.93 | 4.59 | 268.8 | 0.80 | 0.60 |
| REG | 80 | 1.86 | **4.49** | **321.4** | 0.76 | 0.63 |
| | 480 | 1.40 | 4.24 | 296.9 | 0.77 | 0.66 |
| **Ours** | 80 | **1.80** | 4.52 | 272.4 | **0.81** | 0.60 |
| | 400 | 1.47 | 4.51 | 300.3 | 0.80 | 0.63 |
| | 480 | 1.44 | 4.49 | 302.9 | 0.80 | 0.64 |
| REPA-E[†] | 80 | **1.67**[*] | **4.12**[*] | 266.3[*] | 0.80[*] | **0.63**[*] |
| | 800 | 1.12[*] | 4.09[*] | 302.9[*] | 0.79[*] | 0.66[*] |
| **Ours** | 80 | 1.71[*] | 4.54[*] | **274.2**[*] | **0.81**[*] | 0.61[*] |
| | 400 | 1.34[*] | 4.53[*] | 305.0[*] | 0.80[*] | 0.64[*] |
| | 480 | 1.33[*] | 4.46[*] | 307.8[*] | 0.80[*] | 0.64[*] |

size and training duration.

**Variation in REPA-based Methods.** StableVM and VA-REPA can plug into existing REPA-style pipelines without modifying the underlying method. Tab. 3 reports results at 100k iterations for vanilla REPA, REG, and iREPA. Integrating our approach consistently improves the performance across all variants, demonstrating that our contributions are orthogonal and provide reliable, drop-in performance gains.

**Ablation on Split Point $\xi$.** The split point $\xi$ defines the boundary between the *low-variance regime* and the *high-variance regime*, and determines the extent of representation alignment. Tab. 4 analyzes the effect of different $\xi$ values at multiple training stages. At early training (100k iterations), a smaller split point ($\xi = 0.6$) achieves the best FID, indicating that weaker alignment provides an easier supervisory signal that accelerates initial convergence. As training progresses, $\xi = 0.7$ consistently yields the best overall performance, particularly at 400k iterations, where it delivers the best performance. In contrast, a larger split point ($\xi = 0.8$) degrades performance, due to noisy supervision introduced from the high-variance regime. Based on these results, we adopt $\xi = 0.7$ as the default setting.

**Ablations on weighting schemes $w(t)$ and bank capacity $K$.** Fig. 4 studies the sensitivity of VA-REPA weight-

*Table 2.* **Variation in Model Scale and Checkpoints.** Comparison of our full method (StableVM + VA-REPA) against vanilla-REPA. Results are reported without CFG.

| Method | Iter | FID↓ | sFID↓ | IS↑ | Prec.↑ | Rec.↑ |
|---|---|---|---|---|---|---|
| *SiT-B/2* (130M) | | | | | | |
| REPA | 100k | 52.06 | 8.18 | 26.8 | 0.45 | 0.59 |
| **Ours** | 100k | **49.69** | 8.18 | **28.5** | **0.46** | **0.60** |
| *SiT-L/2* (458M) | | | | | | |
| REPA | 100k | 22.75 | 5.52 | 59.9 | 0.61 | 0.63 |
| **Ours** | 100k | **21.03** | **5.51** | **63.9** | **0.62** | 0.63 |
| *SiT-XL/2* (675M) | | | | | | |
| REPA | 100k | 18.59 | 5.39 | 70.6 | 0.64 | 0.62 |
| **Ours** | 100k | **17.12** | 5.39 | **74.8** | **0.65** | **0.63** |
| REPA | 200k | 11.04 | **5.02** | 101.9 | 0.68 | 0.64 |
| **Ours** | 200k | **10.56** | 5.03 | **105.4** | **0.69** | **0.64** |
| REPA | 400k | 8.13 | **5.01** | 124.5 | 0.69 | 0.66 |
| **Ours** | 400k | **7.58** | 5.03 | **127.6** | **0.70** | **0.66** |

*Table 3.* **Compatibility of StableVM + VA-REPA with REPA variants.** Integration results for vanilla REPA, REG, and iREPA. All metrics are reported at 100k iterations.

| Methods | FID↓ | sFID↓ | IS↑ | Prec.↑ | Rec.↑ |
|---|---|---|---|---|---|
| REPA | 18.59 | 5.39 | 70.6 | 0.64 | 0.62 |
| **+Ours** | **17.12** | 5.39 | **74.8** | **0.65** | **0.63** |
| REG | 8.90 | 5.50 | 125.3 | 0.72 | 0.59 |
| **+Ours** | **8.11** | **5.34** | **128.8** | **0.74** | **0.60** |
| iREPA | 16.62 | 5.31 | 76.7 | 0.65 | 0.63 |
| **+Ours** | **16.02** | **5.30** | **78.6** | **0.66** | 0.63 |

*Table 4.* **Ablation on split point** $\xi$**.** Impact of different split points across training stages. The default setting ($\xi = 0.7$) is highlighted.

| Iter | Split point $\xi$ | FID↓ | sFID↓ | IS↑ | Prec.↑ | Rec.↑ |
|---|---|---|---|---|---|---|
| | 0.6 | **17.38** | 5.36 | **73.7** | 0.65 | **0.63** |
| 100k | 0.7 | 17.63 | **5.33** | 73.2 | 0.65 | 0.62 |
| | 0.8 | 17.85 | 5.34 | 72.3 | 0.65 | 0.62 |
| | 0.6 | 10.57 | 5.02 | 104.2 | 0.69 | 0.64 |
| 200k | 0.7 | **10.56** | 5.03 | **105.4** | **0.69** | 0.64 |
| | 0.8 | 10.73 | 5.02 | 103.4 | 0.68 | **0.65** |
| | 0.6 | 7.97 | 5.04 | 124.3 | 0.70 | 0.66 |
| 400k | 0.7 | **7.58** | 5.03 | **127.6** | **0.70** | **0.66** |
| | 0.8 | 7.62 | 4.97 | 127.0 | 0.70 | 0.66 |

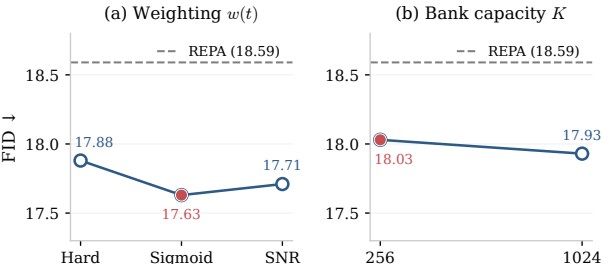

*Figure 4.* **Ablation on VA-REPA weighting and StableVM bank capacity.** Left: effect of different weighting schemes $w(t)$, showing that soft weightings outperform hard thresholding. Right: effect of memory bank capacity $K$, where $K = 256$ already achieves near-optimal performance. All results are evaluated at 100k iterations. REPA baseline is shown as a dashed line.

ing strategies and the StableVM memory bank capacity. Across all weighting strategies, incorporating VA-REPA consistently improves performance over the REPA baseline. Among them, soft weighting schemes outperform the hard threshold, with $w_{\text{sigmoid}}(t)$ achieving the best overall results. For the memory bank, $K = 256$ is sufficient to obtain stable variance reduction, while increasing to $K = 1024$ yields only marginal gains, indicating diminishing returns.

### 4.3. Evaluation of StableVS

**StableVS reduces sampling cost while preserving content across solvers and modalities.** Tab. 5 and 6 show that replacing the base solver with StableVS only in the *low-variance regime* substantially reduces the total number of sampling steps without degrading generation quality. Across SD3.5, Flux, Qwen-Image-2512, and Wan2.2, StableVS with 9 low-variance steps consistently matches or exceeds 30-step baselines, while naively shortening the base solver leads to noticeable drops in reference metrics. This behavior is solver-agnostic and holds for both images and videos: similar gains are observed across Euler, DPM-Solver++, and UniPC. On GenEval, StableVS recovers the perceptual fidelity lost by short-step baselines, producing outputs indistinguishable from 30-step results; on T2V-CompBench,

it maintains generative performance while significantly improving reference metrics, indicating that step reduction doesn't alter spatial structure, motion patterns, or semantic content. These findings directly support our analysis in Sec. 3.3: in the low-variance regime, the posterior effectively collapses, rendering the sampling trajectory deterministic. Consequently, replacing the base solver with StableVS in this regime changes the numerical integration path but not the resulting sample, a conclusion confirmed by qualitative comparisons in Fig. 5 under identical random seeds.

**Choice of split point $\xi$ for StableVS.** We note that the split point $\xi$ used for StableVS differs from that used for VA-REPA. Empirically, we find that a smaller split point adopted in VA-REPA ($\xi = 0.7$) yields samples that are closer to the original 30-step baseline, whereas a larger split point (*e.g.*, $\xi = 0.85$) allows more aggressive step reduction with minimal quality degradation. A detailed ablation over $\xi$ and other StableVS hyperparameters is provided in Tab. 9.

## 5. Related Works

**Training Acceleration of Diffusion and Flow Models.** Diffusion and flow matching models exhibit strong generative performance but incur substantial computational costs when

*Table 5.* **Evaluation on *T2V-CompBench* at** $640 \times 480$**p for *Wan2.2* (Wan et al., 2025).** StableVS replaces the base solver in the *low-variance regime* $[0, \xi]$ (steps in parentheses), keeping the *high-variance regime* $[\xi, 1]$ unchanged. Split point fixed at $\xi = 0.85$. Highlighted rows show StableVS matches or exceeds 30-step baselines with fewer steps.

| Solver configuration | | | Reference metrics | | | T2V-CompBench metrics | | | | | | |
|---|---|---|---|---|---|---|---|---|---|---|---|---|
| Base solver | Solver in $[0, \xi]$ | Total steps | PSNR ↑ | SSIM ↑ | LPIPS ↓ | Consist ↑ | Dynamic ↑ | Spatial ↑ | Motion ↑ | Action ↑ | Interact ↑ | Numeracy ↑ |
| | UniPC(19) | 30 | – | – | – | 0.842 | 0.120 | 0.607 | **0.299** | 0.749 | **0.708** | **0.476** |
| UniPC | UniPC(13) | 20 | 15.61 | 0.593 | 0.377 | 0.821 | 0.123 | **0.618** | 0.265 | 0.720 | 0.703 | 0.462 |
| | **StableVS(9)** | 20 | **31.10** | **0.942** | **0.036** | **0.843** | **0.123** | 0.610 | 0.289 | **0.753** | 0.699 | **0.476** |

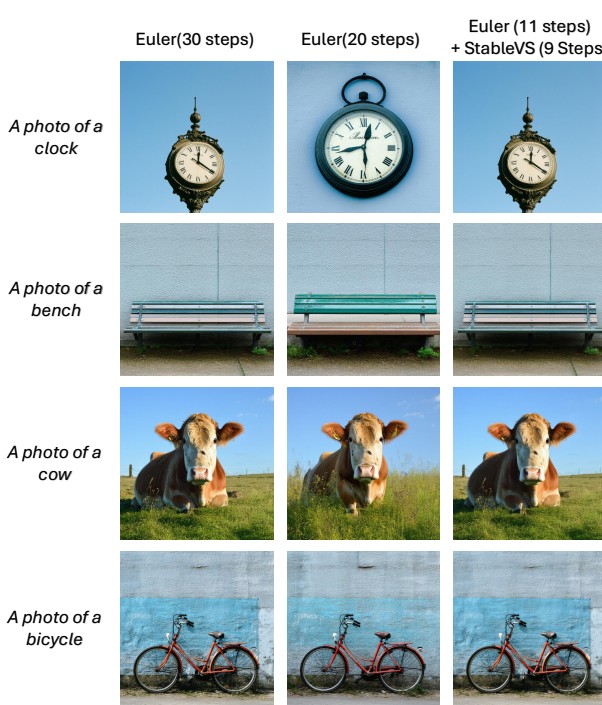

Euler(30 steps)  Euler(20 steps)  Euler (11 steps) + StableVS (9 Steps)

A photo of a clock

A photo of a bench

A photo of a cow

A photo of a bicycle

*Figure 5.* **Visual comparison across prompts on SD3.5 (Esser et al., 2024).** Results are generated using the Euler solver with 30 and 20 steps, and with StableVS replacing Euler in the *low-variance regime*, all under the same random seeds. Compared to the standard 20-step solver, StableVS yields outputs that more closely resemble the 30-step results. Zoom in for details. Additional qualitative comparisons are provided in Appendix H.

*Table 6.* **Evaluation on *GenEval* at** $1024 \times 1024$ **resolution.** StableVS replaces the base solver in the *low-variance regime* $[0, \xi]$, while keeping the *high-variance regime* $[\xi, 1]$ unchanged. Numbers in parentheses (e.g., Euler(19)) denote the number of sampling steps in the *low-variance regime*. The split point is fixed to $\xi = 0.85$ for all models. Highlighted rows demonstrate that StableVS achieves comparable results to the 30-step baseline with fewer total sampling steps. Full results are reported in Tab. 10.

| Solver configuration | | | Overall ↑ | Reference metrics | | |
|---|---|---|---|---|---|---|
| Base solver | Solver in $[0, \xi]$ | Total steps | | PSNR ↑ | SSIM ↑ | LPIPS ↓ |
| *SD3.5-Large* | | | | | | |
| | Euler(19) | 30 | 0.723 | – | – | – |
| Euler | Euler(13) | 20 | 0.710 | 16.93 | 0.753 | 0.333 |
| | **StableVS(9)** | 20 | **0.723** | **36.92** | **0.980** | **0.021** |
| DPM++ | DPM++(19) | 30 | **0.724** | – | – | – |
| | DPM++(13) | 20 | 0.717 | 17.42 | 0.784 | 0.287 |
| | **StableVS(9)** | 20 | 0.719 | **32.61** | **0.957** | **0.063** |
| *Flux-dev* | | | | | | |
| | Euler(19) | 30 | 0.660 | – | – | – |
| Euler | Euler(13) | 20 | 0.659 | 19.74 | 0.820 | 0.244 |
| | **StableVS(9)** | 20 | **0.666** | **35.45** | **0.968** | **0.025** |
| *Qwen-Image-2512* | | | | | | |
| | Euler(22) | 30 | **0.733** | – | – | – |
| Euler | Euler(12) | 17 | 0.721 | 17.01 | 0.767 | 0.277 |
| | **StableVS(9)** | 17 | 0.731 | **32.27** | **0.962** | **0.031** |

trained on high-resolution data. To mitigate this burden, prior work has explored three complementary directions. First, dimensionality reduction methods compress high-resolution data into lower-dimensional representations to reduce training cost. Latent diffusion (Rombach et al., 2022) pioneered this paradigm, with subsequent improvements focusing on more efficient autoencoders (Chen et al., 2025) or localized modeling (Wang et al., 2023). Second, several works incorporate auxiliary regularization or self-supervised objectives to stabilize optimization and accelerate convergence (Zheng et al., 2023; Yu et al., 2024; Wu et al., 2025b; Leng et al., 2025; Singh et al., 2025). Third, improvements to the training objective itself aim to reduce variance or imbalance across timesteps, including log-normal timestep

sampling and SNR-aware weighting (Karras et al., 2022b; Choi et al., 2022). More closely related to our work, recent studies employ self-normalized importance sampling (SNIS) estimators (Hesterberg, 1995) to reduce the variance of score estimation (Xu et al., 2023; Niedoba et al., 2024), albeit at the cost of introducing bias. In contrast, our work provides a variance-centric analysis that explicitly reveals a two-regime structure in stochastic interpolants, and introduces an unbiased, variance-reducing training objective StableVM together with VA-REPA, unifying variance reduction and auxiliary supervision under a single principle.

**Sampling Acceleration of Diffusion and Flow Models.** Reducing the number of sampling steps is critical for practical deployment of diffusion and flow-based generative models. Existing approaches can be broadly categorized into training-required and training-free methods. Training-required approaches leverage additional learning to enable few-step generation, including progressive distillation (Salimans & Ho, 2022; Sauer et al., 2024), consistency models (Song et al., 2023), inductive moment matching (Zhou et al., 2025), and mean flow models (Geng et al., 2026). Training-free approaches instead focus on improved numerical solvers for the reverse ODE, such as DDIM (Song et al.,

2021a), DPM-Solver and its variants (Lu et al., 2022; 2025), and UniPC (Zhao et al., 2023). While these methods improve integration accuracy, they treat the entire diffusion trajectory uniformly. Our StableVS departs from this view by exploiting the low-variance regime identified in our analysis, where the sampling dynamics become effectively deterministic. This regime-aware perspective enables aggressive step reduction without retraining, while remaining compatible with existing solvers in the high-variance regime.

## 6. Conclusion

In this work, we develop a variance-based perspective on stochastic interpolants and reveal a two-regime structure that fundamentally governs both training and sampling dynamics. Our analysis shows that high-variance conditional velocity targets hinder optimization, whereas in the *low-variance regime* the conditional and true velocities coincide, yielding both stable supervision and predictable dynamics. Building on this insight, we introduce the **Stable Velocity** framework, comprising StableVM for unbiased variance-reduced training, VA-REPA for selectively applying auxiliary supervision, and StableVS for finetuning-free acceleration at inference. Extensive experiments on ImageNet $256 \times 256$ and large pretrained T2I and T2V models demonstrate consistent improvements in training stability and substantial sampling speedups without sacrificing sample quality. Beyond the specific methods introduced here, our results suggest that explicitly modeling variance structure along the generative trajectory provides a principled foundation for designing more efficient training objectives and sampling algorithms in diffusion and flow-based generative models.

## Acknowledgments

This work was supported by Kuaishou Technology and also funded, in part, by the NSERC DG Grant (No. RGPIN-2022-04636), the Vector Institute for AI, Canada CIFAR AI Chair, NSERC Canada Research Chair (CRC), NSERC Discovery Grants, and a Google Gift Fund. Resources used in preparing this research were provided, in part, by the Province of Ontario, the Government of Canada through the Digital Research Alliance of Canada alliance.can.ca, and companies sponsoring the Vector Institute www.vectorinstitute.ai/#partners, and Advanced Research Computing at the University of British Columbia. Additional hardware support was provided by John R. Evans Leaders Fund CFI grant.

## Impact Statement

This work aims to advance the understandings of variance structure in deep generative models. While these insights can enable a range of beneficial applications, they also introduce potential risks, particularly related to the misuse of image and video generation technologies. We acknowledge these concerns and emphasize the importance of responsible deployment and appropriate safeguards. Our research does not involve human subjects or the use of personal data. All experiments are conducted on widely used, publicly available benchmarks. We are committed to transparency, reproducibility, and adherence to responsible AI research practices.

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

# A. Preliminaries

The probability flow ordinary differential equation (PF-ODE) for flow matching is defined as follows,

$$\mathrm{d}\boldsymbol{x}_t = \boldsymbol{v}_t(\boldsymbol{x}_t)\,\mathrm{d}t \tag{16}$$

induces marginal distributions that match that of Eq. (1) for all time $t \in [0, 1]$.

In addition, there exists a reverse stochastic differential equation (SDE) whose marginal $p_t(x)$ coincides with that of the PF-ODE in Eq. (16), but with an added diffusion term (Ma et al., 2024):

$$\mathrm{d}\boldsymbol{x}_t = \boldsymbol{v}_t(\boldsymbol{x}_t)\,\mathrm{d}t - \tfrac{1}{2}w_t \boldsymbol{s}_t(\boldsymbol{x}_t)\,\mathrm{d}t + \sqrt{w_t}\,\mathrm{d}\overline{\mathbf{w}}_t, \tag{17}$$

where $\overline{\mathbf{w}}_t$ is a standard Wiener process in backward time, $\sqrt{w_t}$ is the diffusion coefficient, and the score $\boldsymbol{s}_t(\boldsymbol{x}_t) = \nabla_{\boldsymbol{x}_t}\log p_t(\boldsymbol{x}_t)$ can be re-expressed using the velocity field (Ma et al., 2024):

$$\boldsymbol{s}_t(\boldsymbol{x}_t) = \sigma_t^{-1}(\alpha_t \boldsymbol{v}_t(\boldsymbol{x}_t) - \alpha'_t \boldsymbol{x}_t)/(\alpha'_t \sigma_t - \alpha_t \sigma'_t). \tag{18}$$

In diffusion models (Sohl-Dickstein et al., 2015; Ho et al., 2020; Song et al., 2020), the forward process can be modeled as an Itô SDE:

$$\mathrm{d}\boldsymbol{x} = \boldsymbol{f}(\boldsymbol{x}, t)\,\mathrm{d}t + g(t)\,\mathrm{d}\mathbf{w}, \tag{19}$$

where $\mathbf{w}$ is the standard Wiener process, $\boldsymbol{f}(\cdot, t) : \mathbb{R}^d \to \mathbb{R}^d$ is vector-valued function called drift coefficient of $x_t$ and $g(\cdot) : \mathbb{R} \to \mathbb{R}$ is a scalar function known as the diffusion coefficient of $\boldsymbol{x}_t$. It gradually transforms the data distribution $q$ to a known prior as time goes from $t = 0$ to $1$. With stochastic process defined in Eq. (19), Variance-Preserving (VP) diffusion model (Ho et al., 2020; Song et al., 2020; Karras et al., 2022a) implicitly define both $\alpha_t$ and $\sigma_t$ in Eq. (1) with an equilibrium distribution as prior. VP diffusion commonly chooses $\alpha_t = \cos(\frac{\pi}{2}t), \sigma_t = \sin(\frac{\pi}{2}t)$.

The diffusion model is trained to estimate the score of the marginal distribution at time $t$, $\nabla_{\boldsymbol{x}_t}p_t(\boldsymbol{x}_t)$, via a neural network. Specifically, the training objective is a weighted sum of the denoising score matching (DSM) (Vincent, 2011):

$$\min_{\boldsymbol{\theta}} \mathbb{E}_{t,q(\boldsymbol{x}_0),p_t(\boldsymbol{x}_t|\boldsymbol{x}_0)}\lambda_t \left\| \boldsymbol{s}_{\boldsymbol{\theta}}(\boldsymbol{x}_t, t) - \boldsymbol{s}_t(\boldsymbol{x}_t \mid \boldsymbol{x}_0) \right\|^2, \tag{20}$$

where $\lambda_t$ is a positive weighting function, $\boldsymbol{s}_{\boldsymbol{\theta}}(\cdot, \cdot) : [0, 1] \times \mathbb{R}^d \to \mathbb{R}^d$ is a time-dependent vector field parametrized as neural network with parameters $\boldsymbol{\theta}$, the conditional probability path $p_t(\boldsymbol{x}_t \mid \boldsymbol{x}_0) = \mathcal{N}(\boldsymbol{x}_t \mid \alpha_t \boldsymbol{x}_0, \sigma_t^2 \boldsymbol{I})$, and the conditional score function $\boldsymbol{s}_t(\boldsymbol{x}_t \mid \boldsymbol{x}_0) = \nabla_{\boldsymbol{x}_t}\log p_t(\boldsymbol{x}_t \mid \boldsymbol{x}_0)$. The Eq. (20) shares a very similar form with CFM target in Eq. (4). Also, similar to Eq. (5), the objective in Eq. (20) admits a closed-form minimizer (Song et al., 2020; Xu et al., 2023):

$$\boldsymbol{s}_{\boldsymbol{\theta}}^*(\boldsymbol{x}_t, t) = \mathbb{E}_{p_t(\boldsymbol{x}_0|\boldsymbol{x}_t)}\left[\boldsymbol{s}_t(\boldsymbol{x}_t \mid \boldsymbol{x}_0)\right] = \boldsymbol{s}_t(\boldsymbol{x}_t) \tag{21}$$

Here, the marginal probability path $p_t(\boldsymbol{x}_t)$ is a mixture of conditional probability paths $p_t(\boldsymbol{x}_t \mid \boldsymbol{x}_0)$ that vary with data points $\boldsymbol{x}_0$, that is,

$$p_t(\boldsymbol{x}_t) = \int p_t(\boldsymbol{x}_t \mid \boldsymbol{x}_0)q(\boldsymbol{x}_0)\,\mathrm{d}\boldsymbol{x}_0. \tag{22}$$

# B. More Discussion on Related Works

**Representation Alignment and Training Acceleration.** A growing body of work shows that shaping intermediate representations can substantially accelerate the training of diffusion and flow-based generative models. REPA (Yu et al., 2024) introduces an auxiliary objective that aligns hidden states of diffusion transformers with features from pretrained visual encoders, yielding significant gains under limited training budgets. Several follow-up methods explore alternative alignment strategies and regularization mechanisms. Dispersive Loss (Wang & He, 2025) removes the need for external teachers by encouraging internal feature dispersion, providing a lightweight, encoder-free form of representation regularization. HASTE (Wang et al., 2025) addresses over-regularization by restricting alignment to early training stages and disabling it later for improved stability. REG (Wu et al., 2025b) departs from explicit alignment altogether, instead entangling noisy latents with compact semantic tokens throughout the denoising trajectory. REPA-E (Leng et al., 2025) further extends REPA

by jointly fine-tuning both the VAE and diffusion model, substantially improving performance at the cost of increased training complexity. More recently, iREPA (Singh et al., 2025) demonstrates that the effectiveness of representation alignment is driven primarily by spatial structure rather than high-level semantics, enhancing REPA through spatially aware projections and normalization. Our variance-aware perspective is orthogonal to these approaches: rather than modifying the form or source of alignment, we identify when semantic supervision is well-defined along the generative trajectory. As a result, VA-REPA can be naturally combined with existing alignment methods to further improve training efficiency and stability.

**Variance Reduction for Diffusion and Flow Models.** Several works have explored variance reduction for diffusion models through importance sampling over timesteps, primarily aiming to reduce the variance of the diffusion ELBO and empirically improving training efficiency (Huang et al., 2021; Song et al., 2021b). Alternative approaches leverage control variates instead of importance sampling to stabilize training objectives (Wang et al., 2020; Jeha et al., 2024). More closely related to our work, recent studies employ self-normalized importance sampling (SNIS) estimators (Hesterberg, 1995) to reduce the variance of score estimation (Xu et al., 2023; Niedoba et al., 2024). These methods achieve variance reduction by aggregating multiple conditional scores, but typically introduce bias due to self-normalization. In contrast, StableVM focuses on a different source of variance that arises in flow-matching objectives, namely the variability induced by individual reference pairs $(x_0, \varepsilon)$ during target construction. We reduce this variance by aggregating reference samples into a composite conditional formulation, which enables variance reduction while preserving unbiasedness. As a result, StableVM complements existing diffusion-based variance reduction techniques and is particularly effective in settings where variance originates from reference-level stochasticity rather than timestep sampling.

## C. Comparison with Stable Target Field

The standard training objective for diffusion models (Ho et al., 2020; Song et al., 2020; Karras et al., 2022b) is based on *denoising score matching* (DSM) (Vincent, 2011), also suffers from high variance.

To address this issue, Xu et al. (2023) proposed the *Stable Target Field* (STF), which stabilizes training by leveraging a reference batch $\mathcal{B} = \{x_0^i\}_{i=1}^n \sim q(x_0)$. The STF objective is defined as

$$\mathcal{L}_{\text{STF}}(\boldsymbol{\theta}, t) = \mathbb{E}_{\{x_0^i\}_{i=1}^n \sim q(x_0),\, p_t(x_t|x_0^1)} \left\| v_{\boldsymbol{\theta}}(x_t, t) - \sum_{k=1}^n \frac{p_t(x_t \mid x_0^k)}{\sum_{j=1}^n p_t(x_t \mid x_0^j)}\, s_t(x_t \mid x_0^k) \right\|^2. \tag{23}$$

Unlike DSM ($n = 1$), STF forms a weighted average of scores over the reference batch, with weights determined by the conditional likelihoods $p_t(x_t \mid x_0^k)$. This reduces the covariance of the target by a factor of $n$, thereby lowering variance. While STF introduces bias, the minimizer of $\mathcal{L}_{\text{STF}}$ is given by

$$v^*(x_t, t) = \mathbb{E}_{p(x_0|x_t),\, \{x_0^i\}_{i=2}^n \sim q(x_0)} \sum_{k=1}^n \frac{p_t(x_t \mid x_0^k)}{\sum_{j=1}^n p_t(x_t \mid x_0^j)}\, s_t(x_t \mid x_0^k), \tag{24}$$

which deviates from the true score $s_t(x_t)$. However, as $n \to \infty$, this bias vanishes and the weighted estimator converges to the true score.

Our approach differs from STF in three important ways:

1. **General framework.** We extend the variance analysis and variance-reduction strategy to the flow matching as well as stochastic interpolant framework, which generalizes beyond VP diffusion and exhibits a distinct variance structure.

2. **Unbiased objective.** Instead of relying on a finite-sample weighted average, we propose a mixture of conditional probabilities that eliminates bias while still achieving variance reduction.

3. **Class-conditional extension.** While STF does not naturally extend to class-conditional settings, we design a tailored algorithm that maintains variance reduction under classifier-free guidance, improving both convergence and training efficiency.

To further elucidate these differences, we evaluate unconditional generation on CIFAR-10 (Krizhevsky, 2009). For a fair comparison, both STF and StableVM use the same reference batch size $n = 2048$. All metrics are computed over 50K generated samples.

*Table 7.* **Unconditional CIFAR-10 generation.** Performance comparison between CFM, STF, and StableVM. STF instantiated directly from Eq. 23 performs poorly, whereas StableVM achieves faster convergence and better sample quality via unbiased variance reduction. All results use $n = 2048$ and 50k generated samples.

| Iter | Model | FID↓ | IS↑ | sFID↓ | Prec.↑ | Rec.↑ |
|------|-------|------|-----|-------|--------|-------|
| 30k | CFM | 10.76 | 7.97 | 4.55 | 0.58 | 0.57 |
| | STF (Eq. 23) | 38.37 | 6.02 | 9.94 | 0.52 | 0.45 |
| | STF (original impl.) | 9.91 | 8.01 | **4.51** | 0.58 | **0.58** |
| | StableVM | **9.31** | **8.02** | 4.62 | **0.59** | 0.57 |
| 50k | CFM | 7.50 | 8.30 | 4.25 | 0.60 | 0.59 |
| | STF (Eq. 23) | 29.18 | 6.59 | 7.65 | 0.52 | 0.50 |
| | STF (original impl.) | 7.07 | 8.24 | **4.22** | 0.60 | 0.59 |
| | StableVM | **6.87** | **8.38** | 4.34 | **0.61** | **0.59** |
| 200k | CFM | 3.58 | **9.06** | **3.94** | 0.65 | 0.61 |
| | STF (Eq. 23) | 13.50 | 7.69 | 4.88 | 0.56 | 0.57 |
| | STF (original impl.) | 3.93 | 8.92 | 4.06 | 0.65 | **0.61** |
| | StableVM | **3.56** | 9.05 | 3.98 | **0.65** | 0.60 |

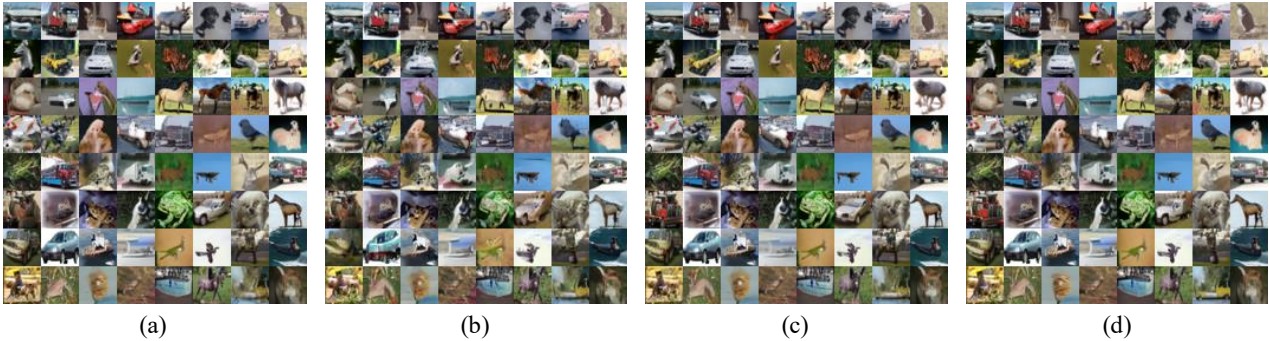

| (a) | (b) | (c) | (d) |

*Figure 6.* **Qualitative comparison of generated samples on CIFAR-10.** Samples generated by the checkpoint at 50k iterations (a) CFM, (b) STF instantiated directly from Eq. 23, (c) the original STF implementation, and (d) our StableVM. StableVM produces sharper and more coherent samples, consistent with its improved convergence and variance reduction.

In this experiment, the STF baseline strictly follows Eq. 23: the reference batch $\mathcal{B} = \{x_0^i\}_{i=1}^n$ is sampled from the data distribution, and the noisy input $x_t$ is generated by applying the forward process to the single reference sample $x_0^1$. While this matches the theoretical formulation in the original paper, it differs from the implementation used by Xu et al. (2023), which instead applies noise to the first $M$ samples in the reference batch and treats them as training inputs. This modification yields a near-unbiased estimator, since $x_t$ is no longer conditioned on a single reference point but effectively drawn from a composite distribution over multiple samples. This behavior closely resembles the composite conditional distribution $p_t^{\text{GMM}}(x_t \mid \{x_0^i\}_{i=1}^n)$ introduced in Sec. 3.1. By contrast, StableVM explicitly samples $x_t$ from a Gaussian mixture model constructed over the reference batch, ensuring unbiased targets while reducing the variance.

As shown in Tab. 7 and Fig. 6, STF instantiated directly from Eq. 23 underperforms even standard CFM, whereas StableVM consistently accelerates convergence and improves sample quality. While the original STF implementation performs substantially better than its direct instantiation from Eq. 23, yet still lags behind StableVM, this comparison highlights a key distinction: STF's empirical gains arise from an implicit deviation from its theoretical objective, whereas StableVM is explicitly designed to be unbiased, variance-reduced, and readily extensible to flow-matching settings.

# D. Algorithms

For clarity of presentation, we provide the detailed training procedures of StableVM in this appendix. The main text focuses on the variance-driven formulation and theoretical properties, while Algorithm 1 summarizes the core StableVM training loop. We further extend StableVM to the class-conditional setting with classifier-free guidance, with the complete procedure given in Algorithm 2.

StableVM introduces additional computation and memory overhead that scales linearly with the bank capacity, i.e. $\mathcal{O}(K)$. Specifically, the self-normalized target requires computing a weighted average over the reference batch for each class, and in class-conditioned generation, the memory bank stores latent representations per class. For example, in ImageNet generation, storing 256 latents for each of the 1,000 classes in fp16 precision requires approximately 2 GB of additional memory. The associated computation incurs only minor latency, which is negligible compared to the model's forward and backward passes.

---

**Algorithm 1** Stable Velocity Matching

---

**Require:** Training iteration $T$, initial model $v_{\theta}$, dataset $\mathcal{D}$, learning rate $\eta$
1: **for** $iter = 1 \dots T$ **do**
2:     Sample a batch $\{x_0^i\}_{i=0}^n$ from $\mathcal{D}$
3:     Uniformly sample time $t \sim q_t(t)$ from $[0, 1]$
4:     Sample perturbed batch $\{x_t^j\}_{j=1}^M$ from

$$p_t^{\text{GMM}}(x_t^j \mid \{x_0^i\}_{i=0}^n) = \sum_{i=0}^K \frac{1}{n} p_t(x_t^j \mid x_0^i)$$

5:     Calculate stable vector field for all $x_t^j$:

$$\hat{v}_{\text{StableVM}}(x_t^j; \{x_0^i\}_{i=0}^n) := \frac{\sum_{k=1}^n p_t(x_t \mid x_0^k) v_t(x_t \mid x_0^k)}{\sum_{j=1}^n p_t(x_t \mid x_0^j)}$$

6:     Calculate loss:

$$\mathcal{L}(\theta) = \frac{1}{M} \sum_{j=1}^M \lambda(t) \left\| v_{\theta}(x_t^j, t) - \hat{v}_{\text{StableVM}}(x_t^j; \{x_0^i\}_{i=0}^n) \right\|^2$$

7:     Update model: $\theta \leftarrow \theta - \eta \nabla \mathcal{L}(\theta)$
8: **end for**
9: **Return** $v_{\theta}$

---

---

**Algorithm 2** Stable Velocity Matching with Classifier-free Guidance

---

**Input:** training iterations $T$, model $\boldsymbol{v_\theta}$, dataset $\mathcal{D}$, learning rate $\eta$, batch size $B$, number of classes $C$, per-class bank capacity $K$, CFG dropout probability $p_{\mathrm{cfg}}$

Initialize memory bank $\mathcal{M} = \{\mathcal{M}_c\}_{c=0}^C$, where $c{=}0, \ldots, C{-}1$ denote class-conditional banks and $c{=}C$ is the unconditional bank

**for** $c = 0, \ldots, C$ **do**

    $\mathcal{M}_c \leftarrow$ prefilled FIFO queue of capacity $K$

**end for**

**for** $iter = 1 \ldots T$ **do**

    Sample times $\{t_i\}_{i=1}^B \sim q_t([0,1))$

    Uniformly sample labels $\{y^i\}_{i=1}^B$ from $\{0, \ldots, C{-}1\}$

    Set $y^i \leftarrow C$ with probability $p_{\mathrm{cfg}}$

    **for** $i = 1 \ldots B$ **do**

        Sample perturbed sample $\boldsymbol{x}_{t_i}^i$ from

$$p_{\mathrm{GMM}}(\boldsymbol{x}_{t_i}^i \mid \mathcal{M}_{y^i}) = \frac{1}{|\mathcal{M}_{y^i}|} \sum_{\boldsymbol{x}_0^{\mathrm{ref}} \in \mathcal{M}_{y^i}} p_t(\boldsymbol{x}_{t_i}^i \mid \boldsymbol{x}_0^{\mathrm{ref}})$$

    Compute stable field

$$\boldsymbol{v}_{\mathcal{M}_{y^i}}(\boldsymbol{x}_{t_i}^i) = \sum_{\boldsymbol{x}_0^{\mathrm{ref}} \in \mathcal{M}_{y^i}} \frac{p_t(\boldsymbol{x}_{t_i}^i \mid \boldsymbol{x}_0^{\mathrm{ref}})}{\sum_{\boldsymbol{y}_0^{\mathrm{ref}} \in \mathcal{M}_{y^i}} p_t(\boldsymbol{x}_{t_i}^i \mid \boldsymbol{y}_0^{\mathrm{ref}})} \, \boldsymbol{v}_t(\boldsymbol{x}_{t_i}^i \mid \boldsymbol{x}_0^{\mathrm{ref}})$$

    **end for**

    Compute loss

$$\mathcal{L}(\boldsymbol{\theta}) = \frac{1}{B} \sum_{i=1}^B \lambda(t_i) \big\| \boldsymbol{v_\theta}(\boldsymbol{x}_{t_i}^i, t_i, y^i) - \boldsymbol{v}_{\mathcal{M}_{y^i}}(\boldsymbol{x}_{t_i}^i) \big\|^2$$

    Update parameters $\boldsymbol{\theta} \leftarrow \boldsymbol{\theta} - \eta \nabla_{\boldsymbol{\theta}} \mathcal{L}(\boldsymbol{\theta})$

    Sample $\mathcal{B} = \{(\boldsymbol{x}_0^i, y_i)\}_{i=1}^B$ from $\mathcal{D}$

    **for** $i = 1 \ldots B$ **do**

        Push $\boldsymbol{x}_0^i$ into $\mathcal{M}_{y_i}$ (evict oldest if full)

        Push $\boldsymbol{x}_0^i$ into $\mathcal{M}_C$ (unconditional bank)

    **end for**

**end for**

**Output:** trained velocity field $\boldsymbol{v_\theta}$

---

# E. Proofs

## E.1. $p_t^{\mathrm{GMM}}$ Posterior

**Proposition E.1.** *The posterior* $p_t^{GMM}\left(\{\boldsymbol{x}_0^i\}_{i=1}^n \mid \boldsymbol{x}_t\right) = \frac{1}{n} \sum_{i=1}^n \left( p_t\left(\boldsymbol{x}_0^i \mid \boldsymbol{x}_t\right) \prod_{j \neq i} q(\boldsymbol{x}_0^j) \right)$.

*Proof.* This is a simple application of the Bayes rule. Note that we have

$$
\begin{aligned}
p_t^{\mathrm{GMM}}\left(\{\boldsymbol{x}_0^i\}_{i=1}^n \mid \boldsymbol{x}_t\right) &= \frac{p_t^{\mathrm{GMM}}\left(x_t \mid \{\boldsymbol{x}_0^i\}_{i=1}^n\right) \cdot \prod_{i=1}^n q(\boldsymbol{x}_0^i)}{p_t(\boldsymbol{x}_t)} \\
&= \frac{1}{p_t(\boldsymbol{x}_t)} \left( \sum_{i=1}^n \frac{1}{n} \cdot p_t(\boldsymbol{x}_t \mid \boldsymbol{x}_0^i) \right) \left( \prod_{i=1}^n q(\boldsymbol{x}_0^i) \right) \\
&= \frac{1}{n} \sum_{i=1}^n \left( \frac{p_t(\boldsymbol{x}_t \mid \boldsymbol{x}_0^i)}{p_t(\boldsymbol{x}_t)} \cdot \prod_{j=1}^n q(\boldsymbol{x}_0^j) \right) \\
&= \frac{1}{n} \sum_{i=1}^n \left( p_t(\boldsymbol{x}_0^i \mid \boldsymbol{x}_t) \prod_{j \neq i} q(\boldsymbol{x}_0^j) \right),
\end{aligned}
\tag{25}
$$

as desired. $\qquad\square$

### E.2. Proof of Unbiasedness (Theorem 3.1)

**Theorem 3.1.** *(a) The StableVM target is unbiased. That is, for any $\boldsymbol{x}_t$, we have*

$$\mathbb{E}_{\{\boldsymbol{x}_0^i\} \sim p_t^{GMM}(\cdot|\boldsymbol{x}_t)} \left[ \widehat{\boldsymbol{v}}_{StableVM}(\boldsymbol{x}_t; \{\boldsymbol{x}_0^i\}_{i=0}^n) \right] = \boldsymbol{v}_t(\boldsymbol{x}_t).$$

*(b) The global minimizer $\boldsymbol{v}^*(\boldsymbol{x}_t, t)$ of the StableVM objective $\mathcal{L}_{StableVM}$ is the true velocity field $\boldsymbol{v}_t(\boldsymbol{x}_t)$.*

*Proof.* (a) Using Eq. (25), we obtain

$$\mathbb{E}_{\{\boldsymbol{x}_0^i\}_{i=1}^n \sim p_t^{\mathrm{GMM}}(\cdot|\boldsymbol{x}_t)} \left[ \sum_{k=1}^n \frac{p_t(\boldsymbol{x}_t \mid \boldsymbol{x}_0^k) \boldsymbol{v}_t(\boldsymbol{x}_t \mid \boldsymbol{x}_0^k)}{\sum_{j=1}^n p_t(\boldsymbol{x}_t \mid \boldsymbol{x}_0^j)} \right]$$

$$= \int \frac{1}{n \cdot p_t(\boldsymbol{x}_t)} \left( \sum_{i=1}^n p_t(\boldsymbol{x}_t \mid \boldsymbol{x}_0^i) \right) \left( \prod_{i=1}^n q(\boldsymbol{x}_0^i) \right) \cdot \sum_{k=1}^n \frac{p_t(\boldsymbol{x}_t \mid \boldsymbol{x}_0^k) \boldsymbol{v}_t(\boldsymbol{x}_t \mid \boldsymbol{x}_0^k)}{\sum_{j=1}^n p_t(\boldsymbol{x}_t \mid \boldsymbol{x}_0^j)} \, \mathrm{d}\boldsymbol{x}_0^{1:n}$$

$$= \frac{1}{n \cdot p_t(\boldsymbol{x}_t)} \int \left( \prod_{i=1}^n q(\boldsymbol{x}_0^i) \right) \left( \sum_{k=1}^n p_t(\boldsymbol{x}_t \mid \boldsymbol{x}_0^k) \boldsymbol{v}_t(\boldsymbol{x}_t \mid \boldsymbol{x}_0^k) \right) \, \mathrm{d}\boldsymbol{x}_0^{1:n}$$

$$= \frac{1}{n \cdot p_t(\boldsymbol{x}_t)} \sum_{k=1}^n \int \left( \prod_{i=1}^n q(\boldsymbol{x}_0^i) \right) p_t(\boldsymbol{x}_t \mid \boldsymbol{x}_0^k) \boldsymbol{v}_t(\boldsymbol{x}_t \mid \boldsymbol{x}_0^k) \, \mathrm{d}\boldsymbol{x}_0^{1:n}$$

$$= \frac{1}{n \cdot p_t(\boldsymbol{x}_t)} \sum_{k=1}^n \int \left( \prod_{i \neq k} q(\boldsymbol{x}_0^i) \right) p_t(\boldsymbol{x}_t, \boldsymbol{x}_0^k) \boldsymbol{v}_t(\boldsymbol{x}_t \mid \boldsymbol{x}_0^k) \, \mathrm{d}\boldsymbol{x}_0^{1:n}$$

$$= \frac{1}{n \cdot p_t(\boldsymbol{x}_t)} \sum_{k=1}^n \left( \left( \prod_{i \neq k} \int q(\boldsymbol{x}_0^i) \, \mathrm{d}\boldsymbol{x}_0^i \right) \cdot \int p_t(\boldsymbol{x}_t, \boldsymbol{x}_0^k) \boldsymbol{v}_t(\boldsymbol{x}_t \mid \boldsymbol{x}_0^k) \, \mathrm{d}\boldsymbol{x}_0^k \right)$$

$$= \frac{1}{n} \sum_{k=1}^n \int p_t(\boldsymbol{x}_0^k \mid \boldsymbol{x}_t) \boldsymbol{v}_t(\boldsymbol{x}_t \mid \boldsymbol{x}_0^k) \, \mathrm{d}\boldsymbol{x}_0^k = \boldsymbol{v}_t(\boldsymbol{x}_t),$$

as desired.

(b) Recall the StableVM objective Eq. (8)

$$\mathcal{L}_{\mathrm{StableVM}}(\boldsymbol{\theta}, t)$$

$$= \mathbb{E}_{\boldsymbol{x}_t \sim p_t, \{\boldsymbol{x}_0^i\}_{i=1}^n \sim p_t^{\mathrm{GMM}}(\cdot|\boldsymbol{x}_t)} \left[ \left\| \boldsymbol{v}_{\boldsymbol{\theta}}(\boldsymbol{x}_t, t) - \sum_{k=1}^n \frac{p_t(\boldsymbol{x}_t \mid \boldsymbol{x}_0^k) \boldsymbol{v}_t(\boldsymbol{x}_t \mid \boldsymbol{x}_0^k)}{\sum_{j=1}^n p_t(\boldsymbol{x}_t \mid \boldsymbol{x}_0^j)} \right\|^2 \right]$$

$$= \mathbb{E}_{\boldsymbol{x}_t \sim p_t} \left[ \int p_t^{\mathrm{GMM}}\left( \{\boldsymbol{x}_0^i\}_{i=1}^n \mid \boldsymbol{x}_t \right) \left\| \boldsymbol{v}_{\boldsymbol{\theta}}(\boldsymbol{x}_t, t) - \sum_{k=1}^n \frac{p_t(\boldsymbol{x}_t \mid \boldsymbol{x}_0^k) \boldsymbol{v}_t(\boldsymbol{x}_t \mid \boldsymbol{x}_0^k)}{\sum_{j=1}^n p_t(\boldsymbol{x}_t \mid \boldsymbol{x}_0^j)} \right\|^2 \mathrm{d}\boldsymbol{x}_0^{1:n} \right].$$

For each $\boldsymbol{x}_t$, let

$$\mathcal{L}_{\boldsymbol{x}_t}(\boldsymbol{v}) := \int p_t^{\mathrm{GMM}}\left( \{\boldsymbol{x}_0^i\}_{i=1}^n \mid \boldsymbol{x}_t \right) \left\| \boldsymbol{v} - \sum_{k=1}^n \frac{p_t(\boldsymbol{x}_t \mid \boldsymbol{x}_0^k) \boldsymbol{v}_t(\boldsymbol{x}_t \mid \boldsymbol{x}_0^k)}{\sum_{j=1}^n p_t(\boldsymbol{x}_t \mid \boldsymbol{x}_0^j)} \right\|^2 \mathrm{d}\boldsymbol{x}_0^{1:n}.$$

It suffices to show that for each $\boldsymbol{x}_t$, setting $\boldsymbol{v}$ to $\boldsymbol{v}_t(\boldsymbol{x}_t)$ above minimizes $\mathcal{L}_{\boldsymbol{x}_t}$. Note that $\mathcal{L}_{\boldsymbol{x}_t}$ is differentiable and strictly

convex in $\boldsymbol{v}$, so the minimizer $\boldsymbol{v}^*$ must satisfy $\nabla \mathcal{L}_{\boldsymbol{x}_t}(\boldsymbol{v}^*) = 0$. It follows that

$$
\begin{aligned}
0 &= 2 \int p_t^{\text{GMM}}\left(\{\boldsymbol{x}_0^i\}_{i=1}^n \mid \boldsymbol{x}_t\right) \left(\boldsymbol{v}^* - \sum_{k=1}^n \frac{p_t(\boldsymbol{x}_t \mid \boldsymbol{x}_0^k)\boldsymbol{v}_t(\boldsymbol{x}_t \mid \boldsymbol{x}_0^k)}{\sum_{j=1}^n p_t(\boldsymbol{x}_t \mid \boldsymbol{x}_0^j)}\right) \, \mathrm{d}\boldsymbol{x}_0^{1:n} \\
&= 2 \int p_t^{\text{GMM}}\left(\{\boldsymbol{x}_0^i\}_{i=1}^n \mid \boldsymbol{x}_t\right) \boldsymbol{v}^* \, \mathrm{d}\boldsymbol{x}_0^{1:n} \\
&\quad - 2 \int p_t^{\text{GMM}}\left(\{\boldsymbol{x}_0^i\}_{i=1}^n \mid \boldsymbol{x}_t\right) \sum_{k=1}^n \frac{p_t(\boldsymbol{x}_t \mid \boldsymbol{x}_0^k)\boldsymbol{v}_t(\boldsymbol{x}_t \mid \boldsymbol{x}_0^k)}{\sum_{j=1}^n p_t(\boldsymbol{x}_t \mid \boldsymbol{x}_0^j)} \, \mathrm{d}\boldsymbol{x}_0^{1:n} \\
&= 2\boldsymbol{v}^* - 2\boldsymbol{v}_t(\boldsymbol{x}_t),
\end{aligned}
$$

where we have used part (a) in the last step. Therefore, $\boldsymbol{v}_t(\boldsymbol{x}_t)$ is the unique minimizer of $\mathcal{L}_{\boldsymbol{x}_t}$. This finishes the proof.

$\square$

### E.3. Proofs of the Variance Bounds (Theorem 3.2 and Theorem 3.3)

In this section, we prove the variance reduction bound of our StableVM target as in Eq. (9). We first recall that the StableVM target is the following estimator

$$
\widehat{\boldsymbol{v}}_t := \sum_{k=1}^n \frac{p_t(\boldsymbol{x}_t \mid \boldsymbol{x}_0^k) \, \boldsymbol{v}_t(\boldsymbol{x}_t \mid \boldsymbol{x}_0^k)}{\sum_{j=1}^n p_t(\boldsymbol{x}_t \mid \boldsymbol{x}_0^j)} \in \mathbb{R}^d, \tag{26}
$$

where $\boldsymbol{x}_t \sim p_t$ and $\{\boldsymbol{x}_0^i\}_{i=1}^n \sim p_t^{\text{GMM}}(\cdot \mid \boldsymbol{x}_t)$.

We first prove the weaker version of the variance bound.

**Theorem 3.2.** *Fix $t \in [0,1]$. Assume that the conditional velocity field is affine and the event $\{\boldsymbol{x}_0^1 = \cdots = \boldsymbol{x}_0^n\}$ has probability 0. Then we have $\mathcal{V}_{\text{StableVM}}(t) < \mathcal{V}_{\text{CFM}}(t)$ strictly.*

*Proof.* We first note that for fixed $\{\boldsymbol{x}_0^i\}_{i=1}^n$ and $\boldsymbol{x}_t$, we have by Jensen's inequality that

$$
\left\| \boldsymbol{v}_t(\boldsymbol{x}_t) - \sum_{k=1}^n \frac{p_t(\boldsymbol{x}_t \mid \boldsymbol{x}_0^k)\boldsymbol{v}_t(\boldsymbol{x}_t \mid \boldsymbol{x}_0^k)}{\sum_{j=1}^n p_t(\boldsymbol{x}_t \mid \boldsymbol{x}_0^j)} \right\|^2 \leq \sum_{k=1}^n \frac{p_t(\boldsymbol{x}_t \mid \boldsymbol{x}_0^k)}{\sum_{j=1}^n p_t(\boldsymbol{x}_t \mid \boldsymbol{x}_0^j)} \left\| \boldsymbol{v}_t(\boldsymbol{x}_t) - \boldsymbol{v}_t(\boldsymbol{x}_t \mid \boldsymbol{x}_0^k) \right\|^2,
$$

where equality holds if and only if $\boldsymbol{v}_t(\boldsymbol{x}_t \mid \boldsymbol{x}_0^1) = \cdots = \boldsymbol{v}_t(\boldsymbol{x}_t \mid \boldsymbol{x}_0^n)$ (Cvetkovski, 2012), which happens if and only if $\boldsymbol{x}_0^1 = \cdots = \boldsymbol{x}_0^n$ for affine conditional velocity field.

We have

$$
\begin{aligned}
&\mathcal{V}_{\text{StableVM}}(t) \\
&= \mathbb{E}_{\{\boldsymbol{x}_0^i\}_{i=1}^n \sim q^n} \mathbb{E}_{\boldsymbol{x}_t \sim p_t^{\text{GMM}}(\cdot \mid \{\boldsymbol{x}_0^i\}_{i=1}^n)} \left\| \boldsymbol{v}_t(\boldsymbol{x}_t) - \sum_{k=1}^n \frac{p_t(\boldsymbol{x}_t \mid \boldsymbol{x}_0^k)\boldsymbol{v}_t(\boldsymbol{x}_t \mid \boldsymbol{x}_0^k)}{\sum_{j=1}^n p_t(\boldsymbol{x}_t \mid \boldsymbol{x}_0^j)} \right\|^2 \\
&= \mathbb{E}_{\{\boldsymbol{x}_0^i\}_{i=1}^n \sim q^n} \left[ \int \frac{1}{n}\left(\sum_{i=1}^n p_t(\boldsymbol{x}_t \mid \boldsymbol{x}_0^i)\right) \left\| \boldsymbol{v}_t(\boldsymbol{x}_t) - \sum_{k=1}^n \frac{p_t(\boldsymbol{x}_t \mid \boldsymbol{x}_0^k)\boldsymbol{v}_t(\boldsymbol{x}_t \mid \boldsymbol{x}_0^k)}{\sum_{j=1}^n p_t(\boldsymbol{x}_t \mid \boldsymbol{x}_0^j)} \right\|^2 \mathrm{d}\boldsymbol{x}_t \right].
\end{aligned}
$$

Now let $A$ be the event that $\{\boldsymbol{x}_0^1 = \cdots = \boldsymbol{x}_0^n\}$ and let $A^c$ be the complement of the event $A$. We have

$$
\begin{aligned}
&\mathcal{V}_{\text{StableVM}}(t) \\
&= \mathbb{E}_{\{\boldsymbol{x}_0^i\}_{i=1}^n \sim q^n} \left[ \mathbb{1}_A \cdot \int \frac{1}{n}\left(\sum_{i=1}^n p_t(\boldsymbol{x}_t \mid \boldsymbol{x}_0^i)\right) \left\| \boldsymbol{v}_t(\boldsymbol{x}_t) - \sum_{k=1}^n \frac{p_t(\boldsymbol{x}_t \mid \boldsymbol{x}_0^k)\boldsymbol{v}_t(\boldsymbol{x}_t \mid \boldsymbol{x}_0^k)}{\sum_{j=1}^n p_t(\boldsymbol{x}_t \mid \boldsymbol{x}_0^j)} \right\|^2 \mathrm{d}\boldsymbol{x}_t \right] \\
&\quad + \mathbb{E}_{\{\boldsymbol{x}_0^i\}_{i=1}^n \sim q^n} \left[ \mathbb{1}_{A^c} \cdot \int \frac{1}{n}\left(\sum_{i=1}^n p_t(\boldsymbol{x}_t \mid \boldsymbol{x}_0^i)\right) \left\| \boldsymbol{v}_t(\boldsymbol{x}_t) - \sum_{k=1}^n \frac{p_t(\boldsymbol{x}_t \mid \boldsymbol{x}_0^k)\boldsymbol{v}_t(\boldsymbol{x}_t \mid \boldsymbol{x}_0^k)}{\sum_{j=1}^n p_t(\boldsymbol{x}_t \mid \boldsymbol{x}_0^j)} \right\|^2 \mathrm{d}\boldsymbol{x}_t \right].
\end{aligned}
$$

But by assumption, $A$ has probability 0, so $\mathbb{1}_A = 0$ a.s., and the first expectation above becomes 0. Therefore, by the equality condition of Jensen's inequality, we have a strict inequality

$$
\begin{aligned}
&\mathcal{V}_{\text{StableVM}}(t) \\
&= \mathbb{E}_{\{\boldsymbol{x}_0^i\}_{i=1}^n \sim q^n} \left[ \mathbb{1}_{A^c} \cdot \int \frac{1}{n} \left( \sum_{i=1}^n p_t(\boldsymbol{x}_t \mid \boldsymbol{x}_0^i) \right) \left\| \boldsymbol{v}_t(\boldsymbol{x}_t) - \sum_{k=1}^n \frac{p_t(\boldsymbol{x}_t \mid \boldsymbol{x}_0^k) \boldsymbol{v}_t(\boldsymbol{x}_t \mid \boldsymbol{x}_0^k)}{\sum_{j=1}^n p_t(\boldsymbol{x}_t \mid \boldsymbol{x}_0^j)} \right\|^2 \mathrm{d}\boldsymbol{x}_t \right] \\
&< \mathbb{E}_{\{\boldsymbol{x}_0^i\}_{i=1}^n \sim q^n} \left[ \mathbb{1}_{A^c} \cdot \int \frac{1}{n} \left( \sum_{i=1}^n p_t(\boldsymbol{x}_t \mid \boldsymbol{x}_0^i) \right) \sum_{k=1}^n \frac{p_t(\boldsymbol{x}_t \mid \boldsymbol{x}_0^k)}{\sum_{j=1}^n p_t(\boldsymbol{x}_t \mid \boldsymbol{x}_0^j)} \left\| \boldsymbol{v}_t(\boldsymbol{x}_t) - \boldsymbol{v}_t(\boldsymbol{x}_t \mid \boldsymbol{x}_0^k) \right\|^2 \mathrm{d}\boldsymbol{x}_t \right].
\end{aligned}
$$

We then have

$$
\begin{aligned}
&\mathbb{E}_{\{\boldsymbol{x}_0^i\}_{i=1}^n \sim q^n} \left[ \mathbb{1}_{A^c} \cdot \int \frac{1}{n} \left( \sum_{i=1}^n p_t(\boldsymbol{x}_t \mid \boldsymbol{x}_0^i) \right) \sum_{k=1}^n \frac{p_t(\boldsymbol{x}_t \mid \boldsymbol{x}_0^k)}{\sum_{j=1}^n p_t(\boldsymbol{x}_t \mid \boldsymbol{x}_0^j)} \left\| \boldsymbol{v}_t(\boldsymbol{x}_t) - \boldsymbol{v}_t(\boldsymbol{x}_t \mid \boldsymbol{x}_0^k) \right\|^2 \mathrm{d}\boldsymbol{x}_t \right] \\
&\leq \mathbb{E}_{\{\boldsymbol{x}_0^i\}_{i=1}^n \sim q^n} \left[ \int \frac{1}{n} \left( \sum_{i=1}^n p_t(\boldsymbol{x}_t \mid \boldsymbol{x}_0^i) \right) \sum_{k=1}^n \frac{p_t(\boldsymbol{x}_t \mid \boldsymbol{x}_0^k)}{\sum_{j=1}^n p_t(\boldsymbol{x}_t \mid \boldsymbol{x}_0^j)} \left\| \boldsymbol{v}_t(\boldsymbol{x}_t) - \boldsymbol{v}_t(\boldsymbol{x}_t \mid \boldsymbol{x}_0^k) \right\|^2 \mathrm{d}\boldsymbol{x}_t \right] \\
&= \mathbb{E}_{\{\boldsymbol{x}_0^i\}_{i=1}^n \sim q^n} \left[ \int \frac{1}{n} \sum_{k=1}^n p_t(\boldsymbol{x}_t \mid \boldsymbol{x}_0^k) \left\| \boldsymbol{v}_t(\boldsymbol{x}_t) - \boldsymbol{v}_t(\boldsymbol{x}_t \mid \boldsymbol{x}_0^k) \right\|^2 \mathrm{d}\boldsymbol{x}_t \right] \\
&= \frac{1}{n} \cdot \mathbb{E}_{\{\boldsymbol{x}_0^i\}_{i=1}^n \sim q^n} \left[ \sum_{k=1}^n \int p_t(\boldsymbol{x}_t \mid \boldsymbol{x}_0^k) \left\| \boldsymbol{v}_t(\boldsymbol{x}_t) - \boldsymbol{v}_t(\boldsymbol{x}_t \mid \boldsymbol{x}_0^k) \right\|^2 \mathrm{d}\boldsymbol{x}_t \right] \\
&= \frac{1}{n} \cdot \mathbb{E}_{\{\boldsymbol{x}_0^i\}_{i=1}^n \sim q^n} \left[ \sum_{k=1}^n \mathbb{E}_{\boldsymbol{x}_t \sim p_t(\cdot \mid \boldsymbol{x}_0^k)} \left\| \boldsymbol{v}_t(\boldsymbol{x}_t) - \boldsymbol{v}_t(\boldsymbol{x}_t \mid \boldsymbol{x}_0^k) \right\|^2 \right] \\
&= \frac{1}{n} \sum_{k=1}^n \mathcal{V}_{\text{CFM}}(t) = \mathcal{V}_{\text{CFM}}(t).
\end{aligned}
$$

This proves that $\mathcal{V}_{\text{StableVM}}(t) < \mathcal{V}_{\text{CFM}}(t)$ as desired. $\qquad \square$

Before proceeding to proving the stronger bound, we first give an alternative interpretation of the posterior $p_t^{\text{GMM}}\left(\{\boldsymbol{x}_0^i\}_{i=1}^n \mid \boldsymbol{x}_t\right)$. Consider the following procedure conditioned on $\boldsymbol{x}_t$:

1. Sample a latent index $I$ uniformly from $\{1, \ldots, n\}$.

2. Sample $\boldsymbol{x}_0^I \sim p_t(\cdot \mid \boldsymbol{x}_t)$ and $\boldsymbol{x}_0^j \sim q$ for all $j \neq I$.

We claim the following:

**Lemma E.2.** *(a) The joint distribution of $\{\boldsymbol{x}_0^i\}_{i=1}^n$ sampled from the above procedure conditioned on $\boldsymbol{x}_t$ is exactly $p_t^{GMM}\left(\{\boldsymbol{x}_0^i\}_{i=1}^n \mid \boldsymbol{x}_t\right)$.*

*(b) $\{\boldsymbol{x}_0^i\}_{i=1}^n$ are independent conditioned on $\boldsymbol{x}_t$ and $I$.*

*Proof.* (a) Note that the joint distribution of $\{\boldsymbol{x}_0^i\}_{i=1}^n$ sampled from the above procedure is

$$
\frac{1}{n} \sum_{i=1}^n \left( p_t(\boldsymbol{x}_0^i \mid \boldsymbol{x}_t) \prod_{j \neq i} q(\boldsymbol{x}_0^j) \right),
$$

which matches the joint distribution given by the posterior of $p_t^{\text{GMM}}$.

(b) This is clear by construction.

$\qquad \square$

The following lemmas compute the variance of the StableVM estimator step-by-step.

**Lemma E.3.** *We have*
$$\mathrm{Cov}\left(\widehat{\boldsymbol{v}}_t \mid \boldsymbol{x}_t\right) = \mathbb{E}\left[\mathrm{Cov}\left(\widehat{\boldsymbol{v}}_t \mid \boldsymbol{x}_t, I\right) \mid \boldsymbol{x}_t\right],$$

*where the expectation on the right-hand side is over the random variable $I$.*

*Proof.* By the law of total covariance, we have
$$\mathrm{Cov}\left(\widehat{\boldsymbol{v}}_t \mid \boldsymbol{x}_t\right) = \mathbb{E}_I\left[\mathrm{Cov}\left(\widehat{\boldsymbol{v}}_t \mid \boldsymbol{x}_t, I\right) \mid \boldsymbol{x}_t\right] + \mathrm{Cov}_I\left(\mathbb{E}\left[\widehat{\boldsymbol{v}}_t \mid \boldsymbol{x}_t, I\right] \mid \boldsymbol{x}_t\right).$$

We claim that $\mathbb{E}\left[\widehat{\boldsymbol{v}}_t \mid \boldsymbol{x}_t, I\right]$ does not depend on $I$, so the second term above would be 0.

Let $i \in [n]$. We have
$$\mathbb{E}\left[\widehat{\boldsymbol{v}}_t \mid \boldsymbol{x}_t, I = i\right] = \int p_t(\boldsymbol{x}_0^i \mid \boldsymbol{x}_t) \prod_{j \neq i} q(\boldsymbol{x}_0^j) \cdot \sum_{k=1}^n \frac{p_t(\boldsymbol{x}_t \mid \boldsymbol{x}_0^k)\,\boldsymbol{v}_t(\boldsymbol{x}_t \mid \boldsymbol{x}_0^k)}{\sum_{j=1}^n p_t(\boldsymbol{x}_t \mid \boldsymbol{x}_0^j)}\, \mathrm{d}\boldsymbol{x}_0^{1:n}.$$

Let $\pi : \mathbb{R}^{nd} \to \mathbb{R}^{nd}$ that swaps $\boldsymbol{x}_0^1$ and $\boldsymbol{x}_0^i$, i.e. $\pi(\boldsymbol{x}_0^1, \ldots, \boldsymbol{x}_0^i, \ldots, \boldsymbol{x}_0^n) = (\boldsymbol{x}_0^i, \ldots, \boldsymbol{x}_0^1, \ldots, \boldsymbol{x}_0^n)$. Note that we have $|\det D\pi| = 1$ since the Jacobian $D\pi$ is a permutation matrix. Then by the change of variable formula, we have
$$\mathbb{E}\left[\widehat{\boldsymbol{v}}_t \mid \boldsymbol{x}_t, I = i\right] = \int p_t(\boldsymbol{x}_0^1 \mid \boldsymbol{x}_t) \prod_{j \neq 1} q(\boldsymbol{x}_0^j) \cdot \sum_{k=1}^n \frac{p_t(\boldsymbol{x}_t \mid \boldsymbol{x}_0^k)\,\boldsymbol{v}_t(\boldsymbol{x}_t \mid \boldsymbol{x}_0^k)}{\sum_{j=1}^n p_t(\boldsymbol{x}_t \mid \boldsymbol{x}_0^j)}\, \mathrm{d}\boldsymbol{x}_0^{1:n}$$
$$= \mathbb{E}\left[\widehat{\boldsymbol{v}}_t \mid \boldsymbol{x}_t, I = 1\right].$$

This shows that $\mathrm{Cov}_I\left(\mathbb{E}\left[\widehat{\boldsymbol{v}}_t \mid \boldsymbol{x}_t, I\right] \mid \boldsymbol{x}_t\right) = 0$, which finishes the proof. $\square$

For the next two lemmas, we fix $\ell \in [d]$ and only consider the $\ell^{\text{th}}$ component $\widehat{\boldsymbol{v}}_t^{(\ell)}$ for simplicity. Define
$$V_{n-1} := \sum_{k \neq i} p_t(\boldsymbol{x}_t \mid \boldsymbol{x}_0^k)\boldsymbol{v}_t^{(\ell)}(\boldsymbol{x}_t \mid \boldsymbol{x}_0^k), \quad P_{n-1} := \sum_{k \neq i} p_t(\boldsymbol{x}_t \mid \boldsymbol{x}_0^k),$$
$$v_i := p_t(\boldsymbol{x}_t \mid \boldsymbol{x}_0^i)\boldsymbol{v}_t^{(\ell)}(\boldsymbol{x}_t \mid \boldsymbol{x}_0^i), \quad p_i := p_t(\boldsymbol{x}_t \mid \boldsymbol{x}_0^i).$$

Then we have
$$\widehat{\boldsymbol{v}}_t^{(\ell)} = \frac{V_{n-1} + v_i}{P_{n-1} + p_i}.$$

Note that conditioned on $\boldsymbol{x}_t$ and $I = i$, the random variables $\boldsymbol{x}_0^k$ for $k \neq i$ are all independent and follow the data distribution $q$ by Lemma E.2.

Let us also first compute some expectations that will be useful later. We have for $k \neq i$,
$$\mathbb{E}\left[p_t(\boldsymbol{x}_t \mid \boldsymbol{x}_0^k) \mid \boldsymbol{x}_t, I = i\right] = \int p_t(\boldsymbol{x}_t \mid \boldsymbol{x}_0^k)q(\boldsymbol{x}_0^k)\, \mathrm{d}\boldsymbol{x}_0^k = \int p_t(\boldsymbol{x}_t, \boldsymbol{x}_0^k)\, \mathrm{d}\boldsymbol{x}_0^k = p_t(\boldsymbol{x}_t), \tag{27}$$

$$\mathbb{E}\left[p_t(\boldsymbol{x}_t \mid \boldsymbol{x}_0^k)\boldsymbol{v}_t(\boldsymbol{x}_t \mid \boldsymbol{x}_0^k) \mid \boldsymbol{x}_t, I = i\right] = \int p_t(\boldsymbol{x}_t \mid \boldsymbol{x}_0^k)\boldsymbol{v}_t(\boldsymbol{x}_t \mid \boldsymbol{x}_0^k)q(\boldsymbol{x}_0^k)\, \mathrm{d}\boldsymbol{x}_0^k$$
$$= \int p_t(\boldsymbol{x}_t, \boldsymbol{x}_0^k)\boldsymbol{v}_t(\boldsymbol{x}_t \mid \boldsymbol{x}_0^k)\, \mathrm{d}\boldsymbol{x}_0^k$$
$$= p_t(\boldsymbol{x}_t) \int \frac{p_t(\boldsymbol{x}_t, \boldsymbol{x}_0^k)}{p_t(\boldsymbol{x}_t)}\,\boldsymbol{v}_t(\boldsymbol{x}_t \mid \boldsymbol{x}_0^k)\, \mathrm{d}\boldsymbol{x}_0^k$$
$$= p_t(\boldsymbol{x}_t) \int p_t(\boldsymbol{x}_0^k \mid \boldsymbol{x}_t)\boldsymbol{v}_t(\boldsymbol{x}_t \mid \boldsymbol{x}_0^k)\, \mathrm{d}\boldsymbol{x}_0^k$$
$$= p_t(\boldsymbol{x}_t)\boldsymbol{v}_t(\boldsymbol{x}_t), \tag{28}$$

$$\mathbb{E}_{\boldsymbol{x}_0 \sim p_t(\cdot | \boldsymbol{x}_t)} \left[ \boldsymbol{v}_t(\boldsymbol{x}_t \mid \boldsymbol{x}_0) \mid \boldsymbol{x}_t \right] = \int p_t(\boldsymbol{x}_0 \mid \boldsymbol{x}_t) \, \boldsymbol{v}_t(\boldsymbol{x}_t \mid \boldsymbol{x}_0) \, \mathrm{d}\boldsymbol{x}_0 = \boldsymbol{v}_t(\boldsymbol{x}_t), \tag{29}$$

and

$$\mathbb{E}_{\boldsymbol{x}_0 \sim q} \left[ \frac{p_t(\boldsymbol{x}_0 \mid \boldsymbol{x}_t)}{q(\boldsymbol{x}_0)} \right] = \int p_t(\boldsymbol{x}_0 \mid \boldsymbol{x}_t) \, \mathrm{d}\boldsymbol{x}_0 = 1. \tag{30}$$

We then continue with the lemmas.

**Lemma E.4.** *As $n \to \infty$, we have*

$$\sqrt{n-1} \left( \frac{V_{n-1}}{P_{n-1}} - \boldsymbol{v}_t^{(\ell)}(\boldsymbol{x}_t) \right)$$

$$\xrightarrow{d} \mathcal{N} \left( 0, \, \mathbb{E}_{\boldsymbol{x}_0 \sim p_t(\cdot | \boldsymbol{x}_t)} \left[ \frac{p_t(\boldsymbol{x}_t \mid \boldsymbol{x}_0)}{p_t(\boldsymbol{x}_t)} \left( \boldsymbol{v}_t^{(\ell)}(\boldsymbol{x}_t \mid \boldsymbol{x}_0) - \boldsymbol{v}_t^{(\ell)}(\boldsymbol{x}_t) \right)^2 \right] \right),$$

*where the random variables $V_{n-1}$ and $P_{n-1}$ are conditioned on $\boldsymbol{x}_t$ and $I = i$.*

*Proof.* Write

$$f(\boldsymbol{x}_0) := \boldsymbol{v}_t^{(\ell)}(\boldsymbol{x}_t \mid \boldsymbol{x}_0), \quad w(\boldsymbol{x}_0) := \frac{p_t(\boldsymbol{x}_0 \mid \boldsymbol{x}_t)}{q(\boldsymbol{x}_0)}.$$

Then we have

$$\frac{V_{n-1}}{P_{n-1}} = \frac{\sum_{k \neq i} p_t(\boldsymbol{x}_t \mid \boldsymbol{x}_0^k) \, \boldsymbol{v}_t^{(\ell)}(\boldsymbol{x}_t \mid \boldsymbol{x}_0^k)}{\sum_{k \neq i} p_t(\boldsymbol{x}_t \mid \boldsymbol{x}_0^k)} = \frac{\sum_{k \neq i} p_t(\boldsymbol{x}_t \mid \boldsymbol{x}_0^k) \, f(\boldsymbol{x}_0^k)}{\sum_{k \neq i} p_t(\boldsymbol{x}_t \mid \boldsymbol{x}_0^k)}$$

$$= \frac{\sum_{k \neq i} \frac{p_t(\boldsymbol{x}_t | \boldsymbol{x}_0^k) \, q(\boldsymbol{x}_0^k)}{p_t(\boldsymbol{x}_t) \, q(\boldsymbol{x}_0^k)} f(\boldsymbol{x}_0^k)}{\sum_{k \neq i} \frac{p_t(\boldsymbol{x}_t | \boldsymbol{x}_0^k) \, q(\boldsymbol{x}_0^k)}{p_t(\boldsymbol{x}_t) \, q(\boldsymbol{x}_0^k)}} = \frac{\sum_{k \neq i} \frac{p_t(\boldsymbol{x}_0^k | \boldsymbol{x}_t)}{q(\boldsymbol{x}_0^k)} f(\boldsymbol{x}_0^k)}{\sum_{k \neq i} \frac{p_t(\boldsymbol{x}_0^k | \boldsymbol{x}_t)}{q(\boldsymbol{x}_0^k)}}$$

$$= \frac{\sum_{k \neq i} w(\boldsymbol{x}_0^k) \, f(\boldsymbol{x}_0^k)}{\sum_{k \neq i} w(\boldsymbol{x}_0^k)}. \tag{31}$$

Recall from Lemma E.2 that conditioned on $\boldsymbol{x}_t$ and $I = i$, the $\boldsymbol{x}_0^i$'s are all independent. Therefore, Eq. (31) is a self-normalized importance sampling estimator (Chapter 9 of Owen (2013)) with importance distribution $q(\boldsymbol{x}_0)$, nominal distribution $p_t(\boldsymbol{x}_0 \mid \boldsymbol{x}_t)$, and importance weight ratio $w(\boldsymbol{x}_0)$.

Note that by Eq. (29) and Eq. (30), we have

$$\mathbb{E}_{\boldsymbol{x}_0 \sim p_t(\cdot | \boldsymbol{x}_t)}[f(\boldsymbol{x}_0)] = \boldsymbol{v}_t^{(\ell)}(\boldsymbol{x}_t), \quad \mathbb{E}_{\boldsymbol{x}_0 \sim q}[w(\boldsymbol{x}_0)] = 1.$$

Following results using the delta method as in Lehmann & Romano (2023) (Theorem 11.2.14) and Owen (2013) (Eq. (9.8)), we have that

$$\sqrt{n-1} \left( \frac{V_{n-1}}{P_{n-1}} - \boldsymbol{v}_t^{(\ell)}(\boldsymbol{x}_t) \right) \xrightarrow{d} \mathcal{N} \left( 0, \, \mathbb{E}_{\boldsymbol{x}_0 \sim q} \left[ w(\boldsymbol{x}_0)^2 \left( f(\boldsymbol{x}_0) - \boldsymbol{v}_t^{(\ell)}(\boldsymbol{x}_t) \right)^2 \right] \right).$$

Expanding the variance term above gives us

$$\mathbb{E}_{\boldsymbol{x}_0 \sim q} \left[ w(\boldsymbol{x}_0)^2 \left( f(\boldsymbol{x}_0) - \boldsymbol{v}_t^{(\ell)}(\boldsymbol{x}_t) \right)^2 \right]$$

$$= \mathbb{E}_{\boldsymbol{x}_0 \sim q} \left[ \frac{p_t(\boldsymbol{x}_0 \mid \boldsymbol{x}_t)^2}{q(\boldsymbol{x}_0)^2} \cdot \left( \boldsymbol{v}_t^{(\ell)}(\boldsymbol{x}_t \mid \boldsymbol{x}_0) - \boldsymbol{v}_t^{(\ell)}(\boldsymbol{x}_t) \right)^2 \right]$$

$$= \int \frac{p_t(\boldsymbol{x}_0 \mid \boldsymbol{x}_t)^2}{q(\boldsymbol{x}_0)} \cdot \left( \boldsymbol{v}_t^{(\ell)}(\boldsymbol{x}_t \mid \boldsymbol{x}_0) - \boldsymbol{v}_t^{(\ell)}(\boldsymbol{x}_t) \right)^2 \, \mathrm{d}\boldsymbol{x}_0$$

$$= \frac{1}{p_t(\boldsymbol{x}_t)} \int p_t(\boldsymbol{x}_0 \mid \boldsymbol{x}_t) \cdot \frac{p_t(\boldsymbol{x}_0 \mid \boldsymbol{x}_t) \, p_t(\boldsymbol{x}_t)}{q(\boldsymbol{x}_0)} \left( \boldsymbol{v}_t^{(\ell)}(\boldsymbol{x}_t \mid \boldsymbol{x}_0) - \boldsymbol{v}_t^{(\ell)}(\boldsymbol{x}_t) \right)^2 \, \mathrm{d}\boldsymbol{x}_0$$

$$= \frac{1}{p_t(\boldsymbol{x}_t)} \int p_t(\boldsymbol{x}_0 \mid \boldsymbol{x}_t) \cdot p_t(\boldsymbol{x}_t \mid \boldsymbol{x}_0) \left( \boldsymbol{v}_t^{(\ell)}(\boldsymbol{x}_t \mid \boldsymbol{x}_0) - \boldsymbol{v}_t^{(\ell)}(\boldsymbol{x}_t) \right)^2 \, \mathrm{d}\boldsymbol{x}_0$$

$$= \frac{1}{p_t(\boldsymbol{x}_t)} \cdot \mathbb{E}_{\boldsymbol{x}_0 \sim p_t(\cdot | \boldsymbol{x}_t)} \left[ p_t(\boldsymbol{x}_t \mid \boldsymbol{x}_0) \left( \boldsymbol{v}_t^{(\ell)}(\boldsymbol{x}_t \mid \boldsymbol{x}_0) - \boldsymbol{v}_t^{(\ell)}(\boldsymbol{x}_t) \right)^2 \right],$$

as desired. □

**Lemma E.5.** *Assume $\left((n-1)\|\widehat{\boldsymbol{v}}_t - \boldsymbol{v}_t(\boldsymbol{x}_t)\|^2\right)_{n=1}^{\infty}$ is uniformly integrable. Then we have*

$$\mathrm{Var}\left(\widehat{\boldsymbol{v}}_t^{(\ell)} \mid \boldsymbol{x}_t, I = i\right) = \frac{1}{n-1}\mathbb{E}_{\boldsymbol{x}_0 \sim p_t(\cdot\mid\boldsymbol{x}_t)}\left[\frac{p_t(\boldsymbol{x}_t \mid \boldsymbol{x}_0)}{p_t(\boldsymbol{x}_t)}\left(\boldsymbol{v}_t^{(\ell)}(\boldsymbol{x}_t \mid \boldsymbol{x}_0) - \boldsymbol{v}_t^{(\ell)}(\boldsymbol{x}_t)\right)^2\right] + o\left(\frac{1}{n}\right)$$

*for large $n$.*

*Proof.* Let $g(x, y) = x/y$ so that $\widehat{\boldsymbol{v}}_t^{(\ell)} = g(V_{n-1} + v_i, P_{n-1} + p_i)$. Performing a Taylor expansion of $g$ at the point $(V_{n-1}, P_{n-1})$ gives us

$$\widehat{\boldsymbol{v}}_t^{(\ell)} = g(V_{n-1}, P_{n-1}) + \nabla g(V_{n-1} + \lambda_{n-1}v_i, P_{n-1} + \lambda_{n-1}p_i)\begin{bmatrix} v_i \\ p_i \end{bmatrix}$$

$$= \frac{V_{n-1}}{P_{n-1}} + \frac{v_i}{P_{n-1} + \lambda_{n-1}p_i} - \frac{V_{n-1} + \lambda_{n-1}v_i}{(P_{n-1} + \lambda_{n-1}p_i)^2}\, p_i$$

for some $[0, 1]$-valued random variable $\lambda_{n-1}$. It follows that

$$\sqrt{n-1}\left(\widehat{\boldsymbol{v}}_t^{(\ell)} - \boldsymbol{v}_t^{(\ell)}(\boldsymbol{x}_t)\right)$$

$$= \sqrt{n-1}\left(\frac{V_{n-1}}{P_{n-1}} - \boldsymbol{v}_t^{(\ell)}(\boldsymbol{x}_t)\right) + \frac{\sqrt{n-1}\cdot v_i}{P_{n-1} + \lambda_{n-1}p_i} - \sqrt{n-1}\cdot\frac{V_{n-1} + \lambda_{n-1}v_i}{(P_{n-1} + \lambda_{n-1}p_i)^2}\cdot p_i \qquad (32)$$

For the second term in Eq. (32), we first note that by the law of large numbers and Eq. (27), we have $\frac{P_{n-1}}{n-1} \xrightarrow{p} p_t(\boldsymbol{x}_t)$ as $n \to \infty$. Also note that $\frac{\lambda_{n-1}}{n-1} \to 0$ as $n \to \infty$ deterministically. It follows that $\frac{1}{n-1}(P_{n-1} + \lambda_{n-1}p_i) \xrightarrow{p} p_t(\boldsymbol{x}_t)$. We also have $\frac{v_i}{\sqrt{n-1}} \to 0$ deterministically since $\boldsymbol{v}_t$ is bounded. Hence, we have

$$\frac{\sqrt{n-1}\cdot v_i}{P_{n-1} + \lambda_{n-1}p_i} = \frac{\frac{1}{\sqrt{n-1}}v_i}{\frac{1}{n-1}(P_{n-1} + \lambda_{n-1}p_i)} \xrightarrow{p} 0$$

by Slutsky's theorem, so the second term of Eq. (32) converges to 0 in probability.

For the third term in Eq. (32), we have $\frac{1}{(n-1)^2}(P_{n-1} + \lambda_{n-1}p_i)^2 \xrightarrow{p} p_t(\boldsymbol{x}_t)^2$ by Slutsky's theorem. At the same time, by the law of large numbers and Eq. (28), we have $\frac{V_{n-1}}{n-1} \xrightarrow{p} p_t(\boldsymbol{x}_t)\boldsymbol{v}_t^{(\ell)}(\boldsymbol{x}_t)$. Since $\boldsymbol{v}_t$ is bounded, we have $\frac{\lambda_{n-1}v_i}{n-1} \to 0$ deterministically. We also have $\frac{p_i}{\sqrt{n-1}} \to 0$ deterministically. Therefore, we get that

$$\frac{1}{(n-1)^{3/2}}\left(V_{n-1} + \lambda_{n-1}v_i\right)p_i = \frac{p_i}{\sqrt{n-1}}\cdot\left(\frac{V_{n-1}}{n-1} + \frac{\lambda_{n-1}v_i}{n-1}\right)$$

$$\xrightarrow{p} 0\cdot\left(p_t(\boldsymbol{x}_t)\boldsymbol{v}_t^{(\ell)}(\boldsymbol{x}_t) + 0\right) = 0,$$

by Slutsky's theorem. Then by Slutsky's theorem again, we have

$$\sqrt{n-1}\cdot\frac{V_{n-1} + \lambda_{n-1}v_i}{(P_{n-1} + \lambda_{n-1}p_i)^2}\cdot p_i = \frac{\frac{1}{(n-1)^{3/2}}\left(V_{n-1} + \lambda_{n-1}v_i\right)p_i}{\frac{1}{(n-1)^2}\left(P_{n-1} + \lambda_{n-1}p_i\right)^2} \xrightarrow{p} 0,$$

so the third term in Eq. (32) also converges to 0 in probability.

Therefore, combining Lemma E.4 and the results above using Slutsky's theorem, we conclude that

$$\sqrt{n-1}\left(\widehat{\boldsymbol{v}}_t^{(\ell)} - \boldsymbol{v}_t^{(\ell)}(\boldsymbol{x}_t)\right) \xrightarrow{d} \mathcal{N}\left(0, \mathbb{E}_{\boldsymbol{x}_0 \sim p_t(\cdot\mid\boldsymbol{x}_t)}\left[\frac{p_t(\boldsymbol{x}_t \mid \boldsymbol{x}_0)}{p_t(\boldsymbol{x}_t)}\left(\boldsymbol{v}_t^{(\ell)}(\boldsymbol{x}_t \mid \boldsymbol{x}_0) - \boldsymbol{v}_t^{(\ell)}(\boldsymbol{x}_t)\right)^2\right]\right).$$

Then by the uniform integrability assumption, we have that $\widehat{\boldsymbol{v}}_t^{(\ell)}$ has variance

$$\frac{1}{n-1}\mathbb{E}_{\boldsymbol{x}_0 \sim p_t(\cdot\mid\boldsymbol{x}_t)}\left[\frac{p_t(\boldsymbol{x}_t \mid \boldsymbol{x}_0)}{p_t(\boldsymbol{x}_t)}\left(\boldsymbol{v}_t^{(\ell)}(\boldsymbol{x}_t \mid \boldsymbol{x}_0) - \boldsymbol{v}_t^{(\ell)}(\boldsymbol{x}_t)\right)^2\right] + o\left(\frac{1}{n}\right),$$

as desired. □

We are now ready to state and prove the main theorem.

**Theorem 3.3.** *Fix $t \in [0,1]$. Assume $\left((n-1)\|\widehat{\boldsymbol{v}}_t - \boldsymbol{v}_t(\boldsymbol{x}_t)\|^2\right)_{n=1}^{\infty}$ is uniformly integrable. Define $r_t(\boldsymbol{x}_t, \boldsymbol{x}_0) := \frac{p_t(\boldsymbol{x}_0|\boldsymbol{x}_t)}{q(\boldsymbol{x}_0)}$ and $\Delta_t(\boldsymbol{x}_t, \boldsymbol{x}_0) := \|\boldsymbol{v}_t(\boldsymbol{x}_t \mid \boldsymbol{x}_0) - \boldsymbol{v}_t(\boldsymbol{x}_t)\|^2$. Then, for large enough $n$, we have*

$$\mathcal{V}_{\text{StableVM}}(t) = \frac{1}{n-1}\left(\mathcal{V}_{\text{CFM}}(t) + \mathbb{E}_{\boldsymbol{x}_0 \sim q, \boldsymbol{x}_t \sim p_t(\cdot|\boldsymbol{x}_0)}\left[\left(r_t(\boldsymbol{x}_t, \boldsymbol{x}_0) - 1\right)\|\boldsymbol{v}_t(\boldsymbol{x}_t \mid \boldsymbol{x}_0) - \boldsymbol{v}_t(\boldsymbol{x}_t)\|^2\right]\right) + o\left(\frac{1}{n}\right).$$

*Proof.* Recall from Eq. (9) that we have

$$\mathcal{V}_{\text{StableVM}}(t) = \mathbb{E}\left[\operatorname{Tr}\operatorname{Cov}\left(\widehat{\boldsymbol{v}}_t \mid \boldsymbol{x}_t\right)\right]$$
$$= \mathbb{E}\left[\operatorname{Tr}\left(\mathbb{E}\left[\operatorname{Cov}\left(\widehat{\boldsymbol{v}}_t \mid \boldsymbol{x}_t, I\right) \mid \boldsymbol{x}_t\right]\right)\right]$$
$$= \mathbb{E}\left[\mathbb{E}\left[\operatorname{Tr}\operatorname{Cov}\left(\widehat{\boldsymbol{v}}_t \mid \boldsymbol{x}_t, I\right) \mid \boldsymbol{x}_t\right]\right],$$

where the second line follows from Lemma E.3, and the third line follows from the linearity of expectation. Note that the inner expectation is over the random variable $I$, and the outer expectation is over the random variable $\boldsymbol{x}_t$. From Lemma E.5, we have

$$\operatorname{Var}\left(\widehat{\boldsymbol{v}}_t^{(\ell)} \mid \boldsymbol{x}_t, I = i\right) = \frac{1}{n-1}\mathbb{E}_{\boldsymbol{x}_0 \sim p_t(\cdot|\boldsymbol{x}_t)}\left[\frac{p_t(\boldsymbol{x}_t \mid \boldsymbol{x}_0)}{p_t(\boldsymbol{x}_t)}\left(\boldsymbol{v}_t^{(\ell)}(\boldsymbol{x}_t \mid \boldsymbol{x}_0) - \boldsymbol{v}_t^{(\ell)}(\boldsymbol{x}_t)\right)^2\right] + o\left(\frac{1}{n}\right).$$

Then

$$\operatorname{Tr}\operatorname{Cov}\left(\widehat{\boldsymbol{v}}_t \mid \boldsymbol{x}_t, I = i\right) = \sum_{\ell=1}^{d}\operatorname{Var}\left(\widehat{\boldsymbol{v}}_t^{(\ell)} \mid \boldsymbol{x}_t, I = i\right)$$
$$= \frac{1}{n-1}\sum_{\ell=1}^{d}\mathbb{E}_{\boldsymbol{x}_0 \sim p_t(\cdot|\boldsymbol{x}_t)}\left[\frac{p_t(\boldsymbol{x}_t \mid \boldsymbol{x}_0)}{p_t(\boldsymbol{x}_t)}\left(\boldsymbol{v}_t^{(\ell)}(\boldsymbol{x}_t \mid \boldsymbol{x}_0) - \boldsymbol{v}_t^{(\ell)}(\boldsymbol{x}_t)\right)^2\right] + o\left(\frac{1}{n}\right)$$
$$= \frac{1}{n-1}\mathbb{E}_{\boldsymbol{x}_0 \sim p_t(\cdot|\boldsymbol{x}_t)}\left[\frac{p_t(\boldsymbol{x}_t \mid \boldsymbol{x}_0)}{p_t(\boldsymbol{x}_t)}\|\boldsymbol{v}_t(\boldsymbol{x}_t \mid \boldsymbol{x}_0) - \boldsymbol{v}_t(\boldsymbol{x}_t)\|^2\right] + o\left(\frac{1}{n}\right).$$

Since $i$ does not appear in the last line above, we get

$$\mathcal{V}_{\text{StableVM}}(t) = \frac{1}{n-1}\mathbb{E}_{\boldsymbol{x}_t \sim p_t}\left[\mathbb{E}_{\boldsymbol{x}_0 \sim p_t(\cdot|\boldsymbol{x}_t)}\left[\frac{p_t(\boldsymbol{x}_t \mid \boldsymbol{x}_0)}{p_t(\boldsymbol{x}_t)}\|\boldsymbol{v}_t(\boldsymbol{x}_t \mid \boldsymbol{x}_0) - \boldsymbol{v}_t(\boldsymbol{x}_t)\|^2\right]\right] + o\left(\frac{1}{n}\right). \tag{33}$$

For simplicity, we write

$$r_t(\boldsymbol{x}_t, \boldsymbol{x}_0) := \frac{p_t(\boldsymbol{x}_t \mid \boldsymbol{x}_0)}{p_t(\boldsymbol{x}_t)} = \frac{p_t(\boldsymbol{x}_0 \mid \boldsymbol{x}_t)}{q(\boldsymbol{x}_0)}, \quad \Delta_t(\boldsymbol{x}_t, \boldsymbol{x}_0) := \|\boldsymbol{v}_t(\boldsymbol{x}_t \mid \boldsymbol{x}_0) - \boldsymbol{v}_t(\boldsymbol{x}_t)\|^2.$$

Then Eq. (33) becomes

$$\mathcal{V}_{\text{StableVM}}(t) = \frac{1}{n-1}\left(\mathbb{E}_{\boldsymbol{x}_t \sim p_t}\left[\mathbb{E}_{\boldsymbol{x}_0 \sim p_t(\cdot|\boldsymbol{x}_t)}\left[\Delta_t(\boldsymbol{x}_t, \boldsymbol{x}_0)\right]\right]\right.$$
$$\left.+ \mathbb{E}_{\boldsymbol{x}_t \sim p_t}\left[\mathbb{E}_{\boldsymbol{x}_0 \sim p_t(\cdot|\boldsymbol{x}_t)}\left[\left(r_t(\boldsymbol{x}_t, \boldsymbol{x}_0) - 1\right)\Delta_t(\boldsymbol{x}_t, \boldsymbol{x}_0)\right]\right]\right) + o\left(\frac{1}{n}\right)$$
$$= \frac{1}{n-1}\left(\mathcal{V}_{\text{CFM}}(t) + \mathbb{E}_{\boldsymbol{x}_t \sim p_t, \boldsymbol{x}_0 \sim p_t(\cdot|\boldsymbol{x}_t)}\left[\left(r_t(\boldsymbol{x}_t, \boldsymbol{x}_0) - 1\right)\Delta_t(\boldsymbol{x}_t, \boldsymbol{x}_0)\right]\right) + o\left(\frac{1}{n}\right)$$
$$= \frac{1}{n-1}\left(\mathcal{V}_{\text{CFM}}(t) + \mathbb{E}_{\boldsymbol{x}_0 \sim q, \boldsymbol{x}_t \sim p_t(\cdot|\boldsymbol{x}_0)}\left[\left(r_t(\boldsymbol{x}_t, \boldsymbol{x}_0) - 1\right)\Delta_t(\boldsymbol{x}_t, \boldsymbol{x}_0)\right]\right) + o\left(\frac{1}{n}\right),$$

as desired. $\qquad\square$

### E.4. Simulating the Reverse SDE in Low-Variance Regime

Ma et al. (2024) show that the reverse-time SDE (Eq. (17)) with score function $s_t(x) = \nabla_x \log p_t(x)$ and arbitrary diffusion strength $w_t \geq 0$ yields the correct marginal density $p_t(x)$ at each time $t$. Furthermore, as established in Anderson (1982); Ma et al. (2024), if $x_t \sim p_t(x)$, then the reverse-time solution $x_\tau$ at any $\tau \in [0, t]$ is distributed according to the posterior:

$$p_\tau(x_\tau \mid x_t) = \mathbb{E}_{p_t(x_0 \mid x_t)}[p_\tau(x_\tau \mid x_0, x_t)] \approx p_\tau(x_\tau \mid x_0, x_t). \tag{34}$$

**Proposition E.6.** *Let $x_t \sim \mathcal{N}(\alpha_t x_0, \sigma_t^2 I)$ and $x_\tau \sim \mathcal{N}(\alpha_\tau x_0, \sigma_\tau^2 I)$, where $\tau < t$, and $x_0 \sim p(x_0)$ is the clean data sample. For any fixed variance parameter $\beta_t^2 \in (0, \sigma_\tau^2)$, define the posterior distribution as*

$$p_\tau^{\alpha_t}(x_\tau \mid x_t, x_0) = \mathcal{N}(k_t x_t + \lambda_t x_0, \ \beta_t^2 I),$$

*then the coefficients*

$$k_t = \sqrt{\frac{\sigma_\tau^2 - \beta_t^2}{\sigma_t^2}}, \quad \lambda_t = \alpha_\tau - \alpha_t \cdot \sqrt{\frac{\sigma_\tau^2 - \beta_t^2}{\sigma_t^2}}$$

*guarantee that the marginal of $x_\tau$ is $\mathcal{N}(\alpha_\tau x_0, \ \sigma_\tau^2 I)$.*

*Proof.* We begin by expressing $x_t$ using the forward diffusion process:

$$x_t = \alpha_t x_0 + \sigma_t \varepsilon, \quad \varepsilon \sim \mathcal{N}(0, I).$$

We define the reverse model as a Gaussian conditional:

$$x_\tau = k_t x_t + \lambda_t x_0 + \eta, \quad \eta \sim \mathcal{N}(0, \beta_t^2 I).$$

Substituting $x_t$ yields:

$$x_\tau = k_t(\alpha_t x_0 + \sigma_t \varepsilon) + \lambda_t x_0 + \eta = (k_t \alpha_t + \lambda_t)x_0 + k_t \sigma_t \varepsilon + \eta.$$

Hence, the conditional distribution of $x_\tau$ given $x_0$ is:

$$x_\tau \mid x_0 \sim \mathcal{N}\left((k_t \alpha_t + \lambda_t)x_0, \ (k_t^2 \sigma_t^2 + \beta_t^2)I\right).$$

To match the desired marginal $x_\tau \sim \mathcal{N}(\alpha_\tau x_0, \ \sigma_\tau^2 I)$, we require:

$$k_t \alpha_t + \lambda_t = \alpha_\tau,$$
$$k_t^2 \sigma_t^2 + \beta_t^2 = \sigma_\tau^2.$$

Solving the second equation above for $k_t$, we obtain:

$$k_t = \sqrt{\frac{\sigma_\tau^2 - \beta_t^2}{\sigma_t^2}}.$$

Substituting into first equation, we get:

$$\lambda_t = \alpha_\tau - \alpha_t \cdot \sqrt{\frac{\sigma_\tau^2 - \beta_t^2}{\sigma_t^2}}.$$

Thus, the choice of $k_t$ and $\lambda_t$ ensures that the conditional distribution of $x_\tau$ is consistent with the marginal. $\square$

Within this low variance area, we also have

$$v_t(x_t) \approx v_t(x_t \mid x_0) = \frac{\sigma_t'}{\sigma_t}(x_t - \alpha_t x_0) + \alpha_t' x_0, \tag{35}$$

Thus, given the velocity field $v_t(x_t)$ and the current state $x_t$, the target $x_0$ can be extracted as:

$$x_0 = \frac{v_t(x_t) - \frac{\sigma_t'}{\sigma_t} x_t}{\alpha_t' - \frac{\sigma_t'}{\sigma_t} \alpha_t} \tag{36}$$

Plugging in this equation into the original expression, thus the posterior distribution with $\boldsymbol{x}_0$ eliminated via $\boldsymbol{v}_t(\boldsymbol{x}_t)$, is given by:

$$p_\tau(\boldsymbol{x}_\tau \mid \boldsymbol{x}_t, \boldsymbol{v}_t(\boldsymbol{x}_t)) = \mathcal{N}\left(\boldsymbol{\mu}_{\tau|t}, \ \beta_t^2 \mathbf{I}\right)$$

where the posterior mean is explicitly:

$$\boldsymbol{\mu}_{\tau|t} = \left(\sqrt{\frac{\sigma_\tau^2 - \beta_t^2}{\sigma_t^2}} - \left(\alpha_\tau - \alpha_t\sqrt{\frac{\sigma_\tau^2 - \beta_t^2}{\sigma_t^2}}\right) \cdot \frac{\frac{\sigma_t'}{\sigma_t}}{\alpha_t' - \frac{\sigma_t'}{\sigma_t}\alpha_t}\right)\boldsymbol{x}_t + \left(\alpha_\tau - \alpha_t\sqrt{\frac{\sigma_\tau^2 - \beta_t^2}{\sigma_t^2}}\right) \cdot \frac{\boldsymbol{v}_t(\boldsymbol{x}_t)}{\alpha_t' - \frac{\sigma_t'}{\sigma_t}\alpha_t}$$

Assuming $\alpha_t = 1 - t$ and $\sigma_t = t$, the DDIM-style posterior becomes:

$$p_\tau(\boldsymbol{x}_\tau \mid \boldsymbol{x}_t, \boldsymbol{v}_t(\boldsymbol{x}_t)) = \mathcal{N}\left(\boldsymbol{\mu}_{\tau|t}, \ \beta_t^2 \mathbf{I}\right)$$

with mean:

$$\boldsymbol{\mu}_{\tau|t} = \left(\sqrt{\frac{\tau^2 - \beta_t^2}{t^2}} + \left((1-\tau) - (1-t)\sqrt{\frac{\tau^2 - \beta_t^2}{t^2}}\right)\right)\boldsymbol{x}_t - \left((1-\tau) - (1-t)\sqrt{\frac{\tau^2 - \beta_t^2}{t^2}}\right)t\boldsymbol{v}_t(\boldsymbol{x}_t)$$

If we set $\beta_t = 0$, we obtain the deterministic sampler:

$$\boldsymbol{x}_\tau = \boldsymbol{x}_t + (\tau - t)\boldsymbol{v}_t(\boldsymbol{x}_t)$$

### E.5. Explicit PF-ODE Solution in Low-Variance Regime

In *low-variance regime* ($0 \leq t \leq \xi$), the conditional velocity field simplifies as $\boldsymbol{v}_t(\boldsymbol{x}_t) \approx \boldsymbol{v}_t(\boldsymbol{x}_t \mid \boldsymbol{x}_0)$. We can thus derive explicit solutions to the Probability Flow ODE (PF-ODE) under both the stochastic interpolant and VP diffusion frameworks. We consider the **Probability Flow ODE (PF-ODE)** under the stochastic interpolant framework:

$$\frac{\mathrm{d}\boldsymbol{x}_t}{\mathrm{d}t} = \boldsymbol{v}_t(\boldsymbol{x}_t) \approx \boldsymbol{v}_t(\boldsymbol{x}_t \mid \boldsymbol{x}_0) = \frac{\sigma_t'}{\sigma_t}(\boldsymbol{x}_t - \alpha_t\boldsymbol{x}_0) + \alpha_t'\boldsymbol{x}_0, \tag{37}$$

where $\alpha_t$ and $\sigma_t$ define a stochastic interpolant, and $\boldsymbol{x}_0$ is the data point to be matched.

**Closed-form of the Target $\boldsymbol{x}_0$**    Given the velocity field $\boldsymbol{v}_t(\boldsymbol{x}_t)$ and the current state $\boldsymbol{x}_t$, the target $\boldsymbol{x}_0$ can be extracted as:

$$\boldsymbol{x}_0 = \frac{\boldsymbol{v}_t(\boldsymbol{x}_t) - \frac{\sigma_t'}{\sigma_t}\boldsymbol{x}_t}{\alpha_t' - \frac{\sigma_t'}{\sigma_t}\alpha_t} = \frac{\boldsymbol{v}_t(\boldsymbol{x}_t) - \frac{\sigma_t'}{\sigma_t}\boldsymbol{x}_t}{C_t}, \tag{38}$$

where we define the coefficient

$$C_t := \alpha_t' - \frac{\sigma_t'}{\sigma_t}\alpha_t.$$

**Solving the PF-ODE from $t$ to $0 \leq \tau < t$**    We aim to integrate the PF-ODE backward in time from a known terminal state $\boldsymbol{x}_t$. The PF-ODE can be rewritten as:

$$\frac{\mathrm{d}\boldsymbol{x}_t}{\mathrm{d}t} + a(t)\boldsymbol{x}_t = b(t), \quad \text{where} \quad a(t) = -\frac{\sigma_t'}{\sigma_t}, \quad b(t) = C_t\boldsymbol{x}_0. \tag{39}$$

This is a linear nonhomogeneous first-order ODE. The integrating factor is:

$$\mu(t) = \exp\left(\int a(t)\,\mathrm{d}t\right) = \exp\left(-\int \frac{\sigma_t'}{\sigma_t}\,\mathrm{d}t\right) = \frac{1}{\sigma_t}.$$

Multiplying both sides by $\mu(t)$ yields:

$$\frac{\mathrm{d}}{\mathrm{d}t}\left(\frac{\boldsymbol{x}_t}{\sigma_t}\right) = \frac{C_t}{\sigma_t}\boldsymbol{x}_0.$$

Integrating both sides from $t$ to $\tau < t$:

$$\frac{\boldsymbol{x}_\tau}{\sigma_\tau} = \frac{\boldsymbol{x}_t}{\sigma_t} + \int_t^\tau \frac{C(s)}{\sigma_s} \, \mathrm{d}s \cdot \boldsymbol{x}_0, \tag{40}$$

which gives:

$$\boldsymbol{x}_\tau = \sigma_\tau \left( \frac{\boldsymbol{x}_t}{\sigma_t} + I(t, \tau) \cdot \boldsymbol{x}_0 \right), \tag{41}$$

where

$$I(t, \tau) := \int_t^\tau \frac{C(s)}{\sigma_s} \, \mathrm{d}s.$$

**Substituting $\boldsymbol{x}_0$ in Closed Form**   We now substitute the expression for $\boldsymbol{x}_0$ evaluated at time $t$:

$$\boldsymbol{x}_0 = \frac{\boldsymbol{v}_t(\boldsymbol{x}_t) - \frac{\sigma_t'}{\sigma_t} \boldsymbol{x}_t}{C_t}.$$

Substitute this into the solution:

$$\boldsymbol{x}_\tau = \sigma_\tau \left( \frac{\boldsymbol{x}_t}{\sigma_t} + I(t, \tau) \cdot \frac{\boldsymbol{v}_t(\boldsymbol{x}_t) - \frac{\sigma_t'}{\sigma_t} \boldsymbol{x}_t}{C_t} \right) \tag{42}$$

$$= \sigma_\tau \left[ \left( \frac{1}{\sigma_t} - \frac{\sigma_t'}{\sigma_t} \cdot \frac{I(t, \tau)}{C_t} \right) \boldsymbol{x}_t + \frac{I(t, \tau)}{C_t} \boldsymbol{v}_t(\boldsymbol{x}_t) \right]. \tag{43}$$

**Final Expression (Only in Terms of $\boldsymbol{x}_t$)**

$$\boxed{\boldsymbol{x}_\tau = \sigma_\tau \left[ \left( \frac{1}{\sigma_t} - \frac{\sigma_t'}{\sigma_t} \cdot \frac{I(t, \tau)}{C_t} \right) \boldsymbol{x}_t + \frac{I(t, \tau)}{C_t} \boldsymbol{v}_t(\boldsymbol{x}_t) \right]} \tag{44}$$

This provides a fully explicit backward solution to the PF-ODE, depending only on $\boldsymbol{x}_t$ and the velocity field $\boldsymbol{v}_t(\boldsymbol{x}_t)$.

**Special Case: Linear Interpolant**   For the linear interpolant with $\alpha_t = 1 - t$ and $\sigma_t = t$, we have:

$$\alpha_t' = -1, \quad \sigma_t' = 1, \quad \Rightarrow \quad C_t = -\frac{1}{t}, \quad \frac{C_t}{\sigma_t} = -\frac{1}{t^2}.$$

Then:

$$I(t, \tau) = \int_t^\tau -\frac{1}{s^2} ds = \frac{1}{\tau} - \frac{1}{t}.$$

Also note:

$$\frac{\sigma_t'}{\sigma_t} = \frac{1}{t}, \quad C_t = -\frac{1}{t}.$$

Plug into the general expression:

$$\boldsymbol{x}_\tau = \tau \left[ \left( \frac{1}{t} - \frac{1}{t} \cdot \frac{1/\tau - 1/t}{-1/t} \right) \boldsymbol{x}_t + \left( \frac{1/\tau - 1/t}{-1/t} \right) \boldsymbol{v}_t(\boldsymbol{x}_t) \right] \tag{45}$$

$$= \tau \left[ \left( \frac{1}{t} + \left( \frac{1}{\tau} - \frac{1}{t} \right) \right) \boldsymbol{x}_t + \left( \frac{1}{t} - \frac{1}{\tau} \right) \boldsymbol{v}_t(\boldsymbol{x}_t) \right] \tag{46}$$

$$= \boldsymbol{x}_t + (\tau - t) \boldsymbol{v}_t(\boldsymbol{x}_t). \tag{47}$$

*Table 8.* **Two-stage training experiment, comparing CFM and StableVM (bank size $K = 256$).** Models are trained for 500k steps with checkpoints evaluated every 40k steps. The results are reported with classifier-free guidance. StableVM consistently achieves better FID and IS across training, demonstrating improved optimization stability when training is restricted to the high-variance regime.

| # Steps | SiT | | | | | StableVM ($K = 256$) | | | | |
|---|---|---|---|---|---|---|---|---|---|---|
| | IS ↑ | FID ↓ | sFID ↓ | Precision ↑ | Recall ↑ | IS ↑ | FID ↓ | sFID ↓ | Precision ↑ | Recall ↑ |
| 40k (SiT) / 50k (StableVM) | 212.5 | 4.33 | 6.29 | 0.71 | 0.69 | **219.2** | **3.92** | **6.04** | **0.72** | 0.69 |
| 80k | 223.8 | 3.53 | 5.61 | 0.73 | **0.68** | **227.3** | **3.35** | 5.61 | 0.73 | 0.67 |
| 120k | 234.4 | 3.07 | **5.34** | 0.74 | 0.67 | **234.8** | **3.02** | 5.40 | 0.74 | 0.67 |
| 160k | **239.7** | **2.81** | **5.24** | 0.74 | **0.67** | 238.9 | 2.87 | 5.74 | 0.74 | 0.66 |
| 200k | 244.7 | 2.62 | **5.20** | 0.75 | 0.67 | **247.0** | **2.59** | 5.21 | 0.75 | 0.67 |
| 240k | 247.4 | 2.54 | **5.17** | 0.75 | 0.67 | **250.3** | **2.48** | 5.20 | 0.75 | 0.67 |
| 280k | 250.9 | 2.47 | 5.14 | 0.75 | 0.66 | **253.3** | **2.40** | **5.13** | 0.75 | **0.67** |
| 320k | 251.7 | 2.39 | 5.17 | 0.75 | **0.67** | **255.4** | **2.34** | **5.13** | 0.75 | 0.66 |
| 360k | 255.4 | 2.35 | 5.16 | 0.75 | **0.67** | **258.5** | **2.28** | **5.11** | **0.76** | 0.66 |
| 400k | 258.6 | 2.28 | 5.17 | 0.76 | 0.67 | **261.7** | **2.17** | **4.98** | 0.76 | 0.67 |
| 440k | 259.3 | 2.29 | 5.20 | 0.76 | 0.67 | **262.3** | **2.16** | **5.00** | 0.76 | 0.67 |
| 480k | 259.7 | 2.26 | 5.22 | 0.76 | **0.67** | **263.1** | **2.16** | **5.03** | 0.76 | 0.66 |
| 500k | 260.1 | 2.26 | 5.23 | 0.76 | 0.67 | **262.7** | **2.17** | **5.06** | 0.76 | 0.67 |

**Final Linear Interpolant Result**

$$\boxed{\boldsymbol{x}_\tau = \boldsymbol{x}_t + (\tau - t)\boldsymbol{v}_t(\boldsymbol{x}_t)} \tag{48}$$

In the case of the linear interpolant, the PF-ODE corresponds to a straight-line trajectory with constant velocity $\boldsymbol{v}_t(\boldsymbol{x}_t)$, enabling exact integration via Euler steps of arbitrary size.

## F. More Experimental Results

### F.1. Two-Stage Training and Sampling

For the two-stage experiment, we set $\xi = 0.7$ as the split point between the low-variance and high-variance regimes. During training, timesteps are sampled uniformly from $[\xi, 1]$, and optimization is performed using either the standard CFM loss or our StableVM loss, while all other configurations follow Sec. 4.1. Specifically, an SiT-XL/2 model is trained from scratch on $t \in [\xi, 1]$, while a pretrained SiT-XL/2 model from REPA is used for stepping in $t \in [0, \xi]$. Each model is trained for 500k steps, with checkpoints evaluated every 40k steps. As shown in Tab. 8, StableVM consistently outperforms CFM across training stages, indicating improved optimization stability in the high-variance regime.

### F.2. Unconditional Synthetic GMM Generation

We also evaluate Algorithm 1 in the unconditional generation setting. Specifically, we construct a synthetic Gaussian Mixture Model (GMM) distribution and train the model to learn it using either the standard CFM loss or our proposed StableVM loss.

The GMM is defined with 100 modes. For each component $k$, the mean vector $\boldsymbol{\mu}_k$ is sampled independently from a uniform distribution over $[-1, 1]$ in each dimension. The variances are drawn independently per component and per dimension from a uniform distribution over $[10^{-2}, 10^{-1}]$, yielding anisotropic Gaussian components. The mixing coefficients $\boldsymbol{\pi}$ are obtained by sampling each entry from Uniform$(0.1, 1.0)$ and normalizing so that they sum to one. To generate samples, we first draw a component index $k$ via multinomial sampling according to $\boldsymbol{\pi}$, then sample from the corresponding Gaussian using the reparameterization trick:

$$\boldsymbol{x} = \boldsymbol{\mu}_k + \boldsymbol{\sigma}_k \odot \boldsymbol{\varepsilon}, \quad \text{where} \quad \boldsymbol{\varepsilon} \sim \mathcal{N}(0, \mathbf{I}),$$

and $\odot$ denotes element-wise multiplication.

We fix the data dimensionality to 10 and evaluate both the CFM loss and our proposed StableVM loss with a reference batch size of 2048 on this distribution. Model performance is assessed by computing the *second-order moment* of the discrepancy between the model's predicted velocity field $\boldsymbol{v_\theta}(\boldsymbol{x}_t, t)$ and the true velocity field $\boldsymbol{v}_t(\boldsymbol{x}_t)$, under the marginal distribution

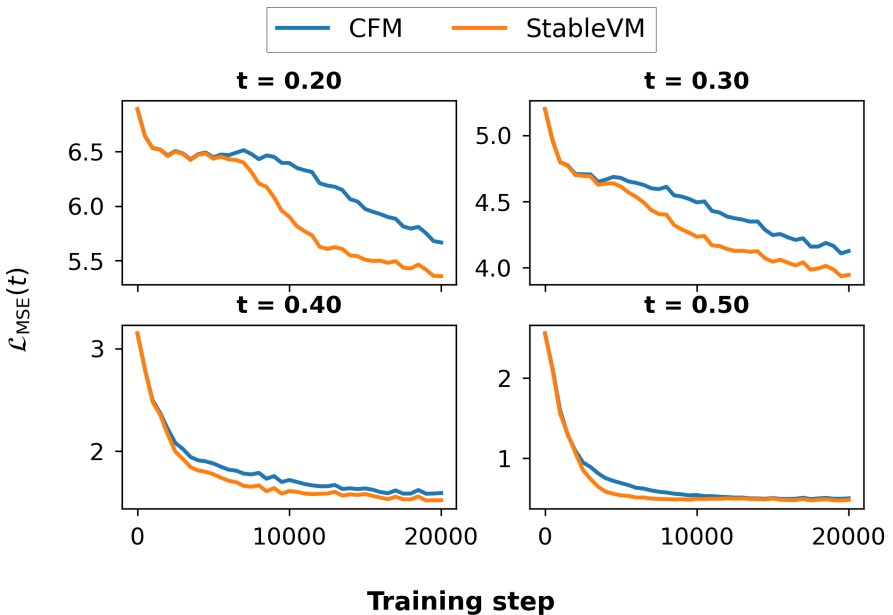

*Figure 7.* **Comparison of CFM and StableVM with** $n = 2048$ **on the synthetic GMM distribution.** We plot the *second-order moment* as a function of training iterations at four time steps: $t = 0.20, 0.30, 0.40,$ and $0.50$.

$p_t(\boldsymbol{x}_t)$:

$$\mathcal{L}_{\mathrm{MSE}}(t) \;=\; \tfrac{1}{2}\,\mathbb{E}_{\boldsymbol{x}_t \sim p_t}\left[\left\|\boldsymbol{v}_{\boldsymbol{\theta}}(\boldsymbol{x}_t, t) - \boldsymbol{v}_t(\boldsymbol{x}_t)\right\|^2\right]. \tag{49}$$

However, the exact velocity field $\boldsymbol{v}_t(\boldsymbol{x}_t) = \mathbb{E}_{p_t(\boldsymbol{x}_0|\boldsymbol{x}_t)}[\boldsymbol{v}_t(\boldsymbol{x}_t \mid \boldsymbol{x}_0)]$ is intractable. We therefore approximate it using a self-normalized importance sampling estimator (Hesterberg, 1995). Concretely, given $N = 50{,}000$ samples $\{\boldsymbol{x}_0^i\}_{i=1}^N$ drawn from the data distribution, we approximate:

$$\boldsymbol{v}_t(\boldsymbol{x}_t) \approx \sum_{i=1}^{N} w_i(\boldsymbol{x}_t)\,\boldsymbol{v}_t(\boldsymbol{x}_t \mid \boldsymbol{x}_0^i), \quad \text{where} \quad w_i(\boldsymbol{x}_t) = \frac{p_t(\boldsymbol{x}_t \mid \boldsymbol{x}_0^i)}{\sum_{j=1}^{N} p_t(\boldsymbol{x}_t \mid \boldsymbol{x}_0^j)}.$$

This approximation enables us to estimate the *second-order moment* at any time step $t$.

As shown in Fig. 7, StableVM consistently achieves faster convergence and lower final error than standard CFM across all evaluated timesteps. This behavior directly reflects the variance reduction predicted by our analysis, even in a fully controlled setting where the true velocity field is accessible.

**Reproduction of Fig. 1.** To reproduce the variance curves in Fig. 1, we follow the same procedure described above but increase the dimensionality of the synthetic GMM to 100 and 500, using identical configuration settings. The true velocity field is again approximated using the self-normalized importance estimator with $N = 10{,}000$ samples. For CIFAR-10 and ImageNet latents, we evaluate on the full training set or a 50,000-sample subset.

### F.3. Ablation Studies of Hyperparameters in StableVS

Tab. 9 presents the effect of three key hyperparameters—the variance factor $f_\beta$, the number of steps in the *low-variance regime*, and the split point $\xi$—on Stable Velocity Sampling with *SD3.5-Large* at $1024 \times 1024$ resolution on the GenEval benchmark.

Tab. 9 shows that StableVS is robust to moderate variations in all three hyperparameters. Increasing the variance factor $f_\beta$ introduces additional stochasticity and degrades reference metrics, consistent with our analysis that the low-variance regime admits nearly deterministic dynamics. Similarly, allocating too few steps to the low-variance regime reduces fidelity, while overly aggressive step reduction beyond the regime boundary ($\xi$ too large) leads to quality degradation. These trends support the variance-regime interpretation underlying StableVS and justify our default configuration.

*Table 9.* **Ablation study on hyperparameters of StableVS.** We analyze the effects of the split point $\xi$, the number of steps in the *low-variance regime* $[0, \xi]$, and the variance factor $f_\beta$ of Stable Velocity Sampling for ***SD3.5-Large*** (Esser et al., 2024) at $1024 \times 1024$ resolution. All experiments use Euler as the base solver.

| Total steps | $\xi$ | $[0, \xi]$ steps | $f_\beta$ | Overall ↑ | PSNR ↑ | SSIM ↑ | LPIPS ↓ |
|---|---|---|---|---|---|---|---|
| **Baseline** | | | | | | | |
| 20 | – | – | – | 0.710 | 16.93 | 0.753 | 0.333 |
| **Default StableVS setting** | | | | | | | |
| 20 | 0.85 | 9 | 0.0 | 0.723 | 36.92 | 0.980 | 0.021 |
| **Variance factor** $f_\beta$ | | | | | | | |
| 20 | 0.85 | 9 | 0.1 | 0.719 | 32.69 | 0.956 | 0.036 |
| 20 | 0.85 | 9 | 0.2 | 0.723 | 29.08 | 0.908 | 0.062 |
| **Low-variance regime steps** $[0, \xi]$ | | | | | | | |
| 15 | 0.85 | 4 | 0.0 | 0.708 | 29.30 | 0.873 | 0.155 |
| 25 | 0.85 | 14 | 0.0 | 0.719 | 39.79 | 0.988 | 0.010 |
| **Split point** $\xi$ | | | | | | | |
| 26 | 0.70 | 9 | 0.0 | 0.726 | 43.65 | 0.992 | 0.006 |
| 22 | 0.80 | 9 | 0.0 | 0.724 | 39.15 | 0.974 | 0.014 |
| 17 | 0.90 | 9 | 0.0 | 0.728 | 31.99 | 0.959 | 0.045 |

## F.4. Full Results on *Geneval*

Tab. 10 provides the full GenEval category-wise breakdown for all models and solver configurations.

# G. Experimental Details

Data preprocessing follows the ADM protocol (Dhariwal & Nichol, 2021), where original images are center-cropped and resized to $256 \times 256$ resolution. For optimization, we employ AdamW (Kingma, 2015; Loshchilov, 2017) with a constant learning rate of $1 \times 10^{-4}$ and a global batch size of 256. Training efficiency and numerical stability are enhanced via mixed-precision (fp16) training, gradient clipping, and an adaptive exponential moving average (EMA) decay schedule following Lipman et al. (2024b). For sigmoid weighting, the sharpness hyperparameters $k = 20$.

StableVS is implemented on top of the Hugging Face diffusers library (von Platen et al.), which provides robust support for state-of-the-art pre-trained models and flexible sampling pipelines.

# H. More Qualitative Results

We provide additional qualitative results for SD3.5 (Esser et al., 2024), Flux (Labs, 2024), SD3 (Esser et al., 2024), and Wan2.2 (Wan et al., 2025), shown in Fig. 8, Fig. 9, Fig. 10, Fig. 11, and Fig. 12, respectively.

*Table 10.* **Detailed evaluation on *GenEval* at** $1024 \times 1024$ **resolution.** We report the overall GenEval score together with per-category breakdowns. StableVS replaces the base solver in the *low-variance regime* $[0, \xi]$ (with the number of steps indicated in parentheses), while keeping the *high-variance regime* $[\xi, 1]$ unchanged. The split point is fixed to $\xi = 0.85$ for all models.

| Solver configuration | | | GenEval Metrics | | | | | | | Reference metrics | | |
|---|---|---|---|---|---|---|---|---|---|---|---|---|
| Base solver | Solver in $[0, \xi]$ | Total steps | Overall ↑ | Single ↑ | Two ↑ | Counting ↑ | Colors ↑ | Position ↑ | Color Attr ↑ | PSNR ↑ | SSIM ↑ | LPIPS ↓ |
| *SD3.5-Large* (Esser et al., 2024) | | | | | | | | | | | | |
| | Euler(19) | 30 | 0.723 | 0.994 | 0.907 | 0.697 | 0.843 | 0.293 | 0.608 | – | – | – |
| Euler | Euler(13) | 20 | 0.710 | 0.994 | 0.884 | 0.650 | 0.838 | 0.290 | 0.603 | 16.93 | 0.753 | 0.333 |
| | **StableVS(9)** | 20 | 0.723 | 0.994 | 0.891 | 0.700 | 0.838 | 0.298 | 0.615 | **36.92** | **0.980** | **0.021** |
| | DPM++(19) | 30 | 0.724 | 0.997 | 0.917 | 0.700 | 0.825 | 0.278 | 0.630 | – | – | – |
| DPM++ | DPM++(13) | 20 | 0.717 | 0.994 | 0.889 | 0.681 | 0.717 | 0.278 | 0.620 | 17.42 | 0.784 | 0.287 |
| | **StableVS(9)** | 20 | 0.719 | 0.997 | 0.889 | 0.697 | 0.806 | 0.275 | 0.625 | **32.61** | **0.957** | **0.063** |
| *Flux-dev* (Labs, 2024) | | | | | | | | | | | | |
| | Euler(19) | 30 | 0.660 | 0.984 | 0.828 | 0.700 | 0.801 | 0.205 | 0.443 | – | – | – |
| Euler | Euler(13) | 20 | 0.659 | 0.991 | 0.838 | 0.694 | 0.785 | 0.203 | 0.445 | 19.74 | 0.820 | 0.244 |
| | **StableVS(9)** | 20 | 0.666 | 0.981 | 0.833 | 0.697 | 0.798 | 0.210 | 0.475 | **35.45** | **0.968** | **0.025** |
| *Qwen-Image-2512* (Wu et al., 2025a) | | | | | | | | | | | | |
| | Euler(22) | 30 | 0.733 | 0.988 | 0.907 | 0.572 | 0.859 | 0.450 | 0.623 | – | – | – |
| Euler | Euler(12) | 17 | 0.721 | 0.994 | 0.899 | 0.556 | 0.859 | 0.430 | 0.590 | 17.01 | 0.767 | 0.277 |
| | **StableVS(9)** | 17 | 0.731 | 0.988 | 0.907 | 0.569 | 0.859 | 0.450 | 0.615 | **32.27** | **0.962** | **0.031** |

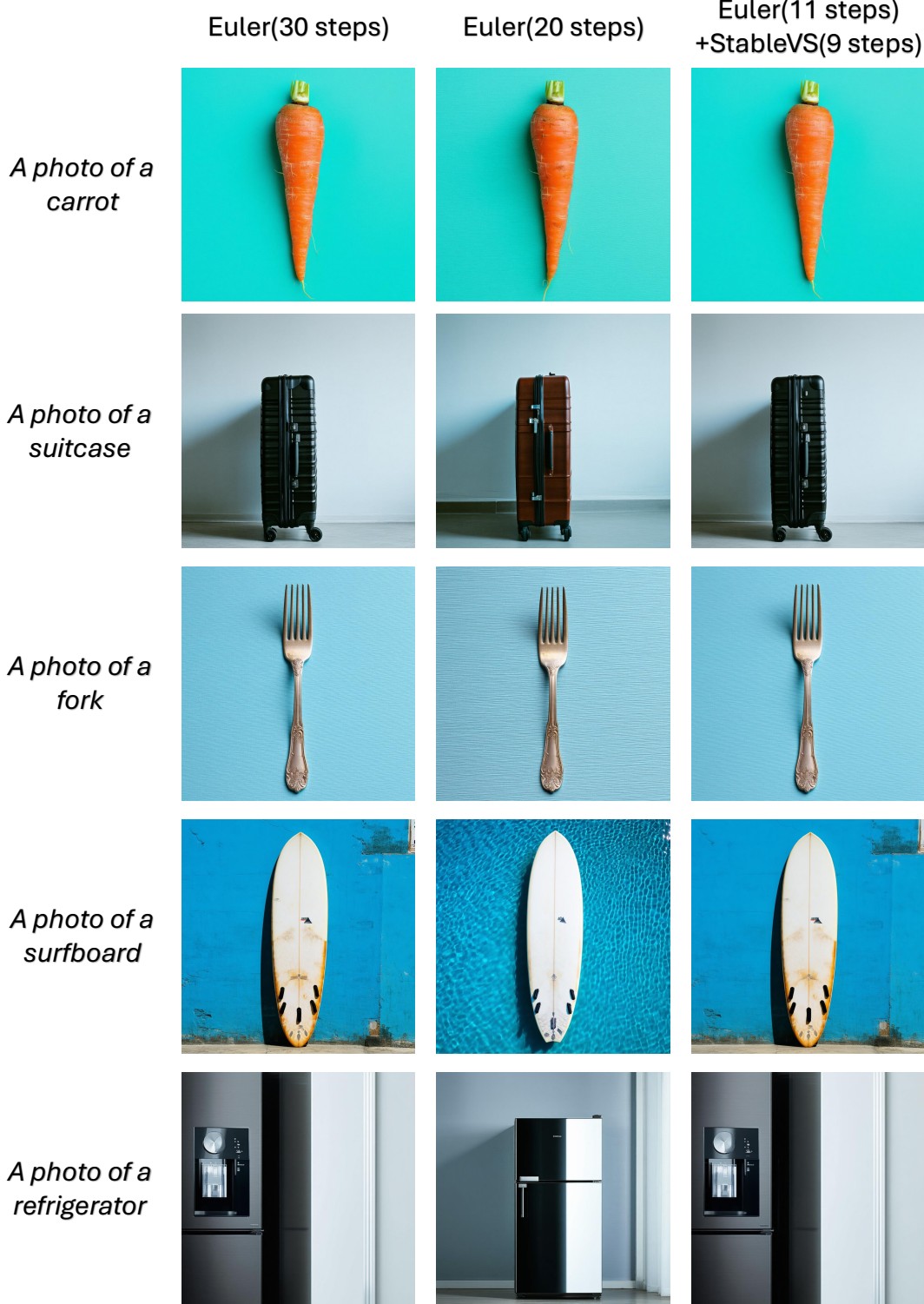

*Figure 8.* **Visual comparison of SD3.5-Large (Esser et al., 2024) on different prompts.** Results are generated using the Euler solver with 30 and 20 steps, and with StableVS replacing Euler in the *low-variance regime*, all under the same random seed. Compared to the standard 20-step solver, StableVS yields outputs that more closely resemble the 30-step results. Zoom in for details.

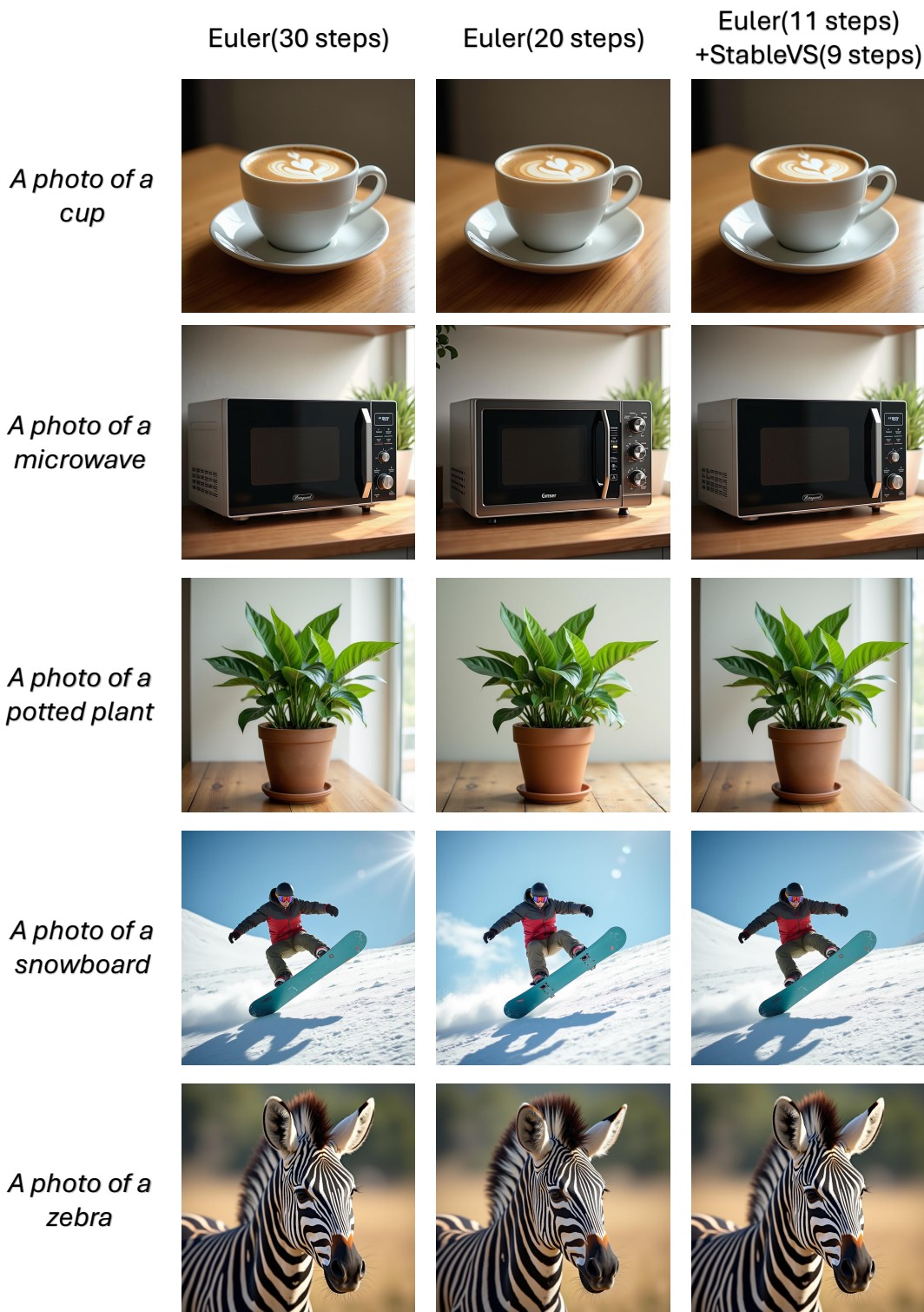

*Figure 9.* **Visual comparison of Flux-dev (**Labs, 2024**) on different prompts.** Results are generated using the Euler solver with 30 and 20 steps, and with StableVS replacing Euler in the *low-variance regime*, all under the same random seed. Compared to the standard 20-step solver, StableVS yields outputs that more closely resemble the 30-step results. Zoom in for details.

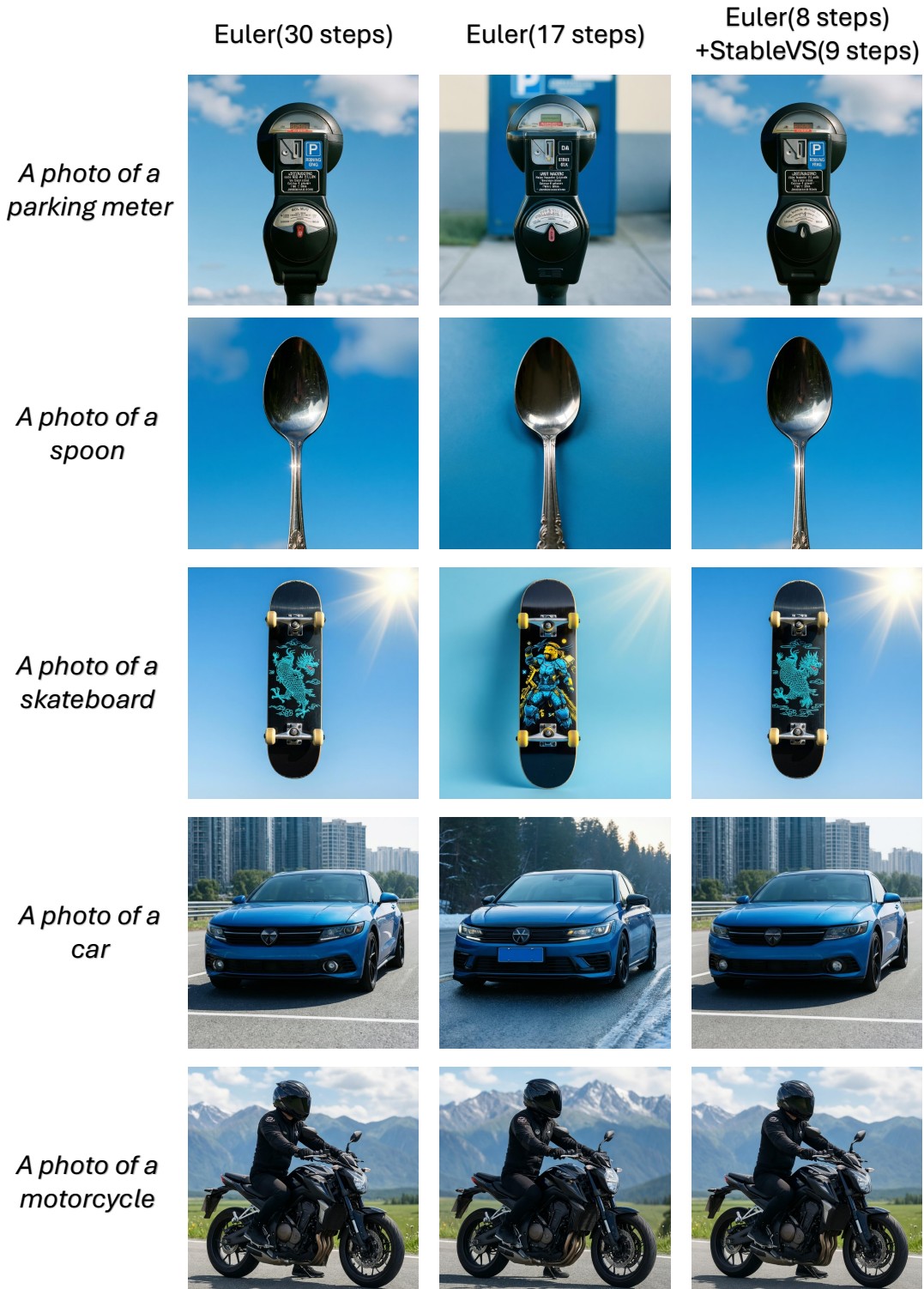

*Figure 10.* **Visual comparison of Qwen-Image-2512 ([Wu et al., 2025a](#)) on different prompts.** Results are generated using the Euler solver with 30 and 17 steps, and with StableVS replacing Euler in the *low-variance regime*, all under the same random seed. Compared to the standard 17-step solver, StableVS yields outputs that more closely resemble the 30-step results. Zoom in for details.

Prompt: *A dog plays guitar while a cat takes a selfie.*

Prompt: *Green taxi drives past a yellow building.*

Prompt: *A timelapse of a green banana turning yellow as it ripens.*

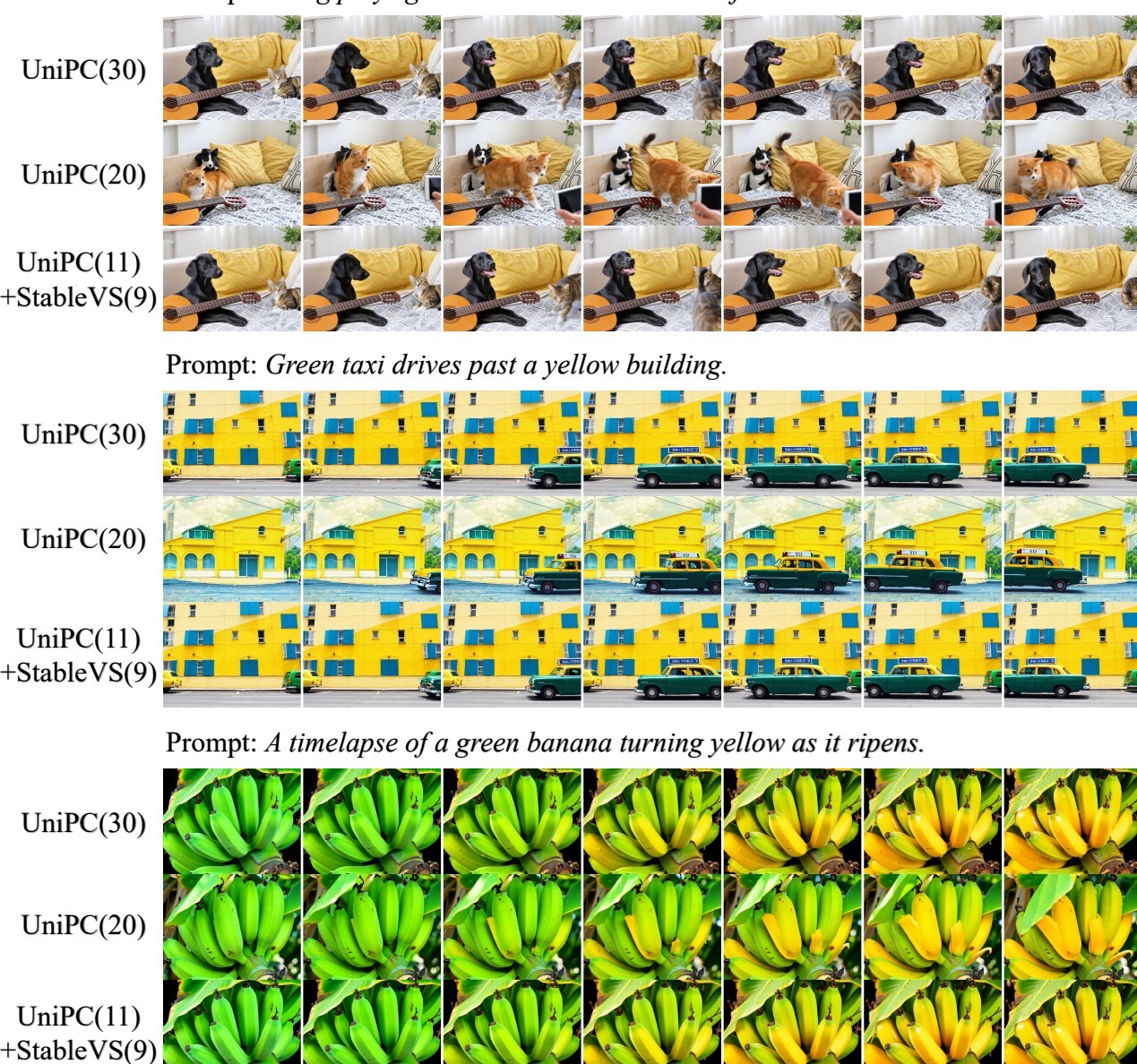

*Figure 11.* **Visual comparison of Wan2.2 (Wan et al., 2025) on different prompts.** Results are generated using UniPC solver with 30 and 20 steps, and with StableVS replacing UniPC in the *low-variance regime*, all under the same random seed. Compared to the standard 20-step solver, StableVS yields outputs that more closely resemble the 30-step results. Zoom in for details.

Prompt: *A horse jumps over a fence.*

Prompt: *A boat sails to the left on the ocean.*

Prompt: *A dog running on the left of a bicycle.*

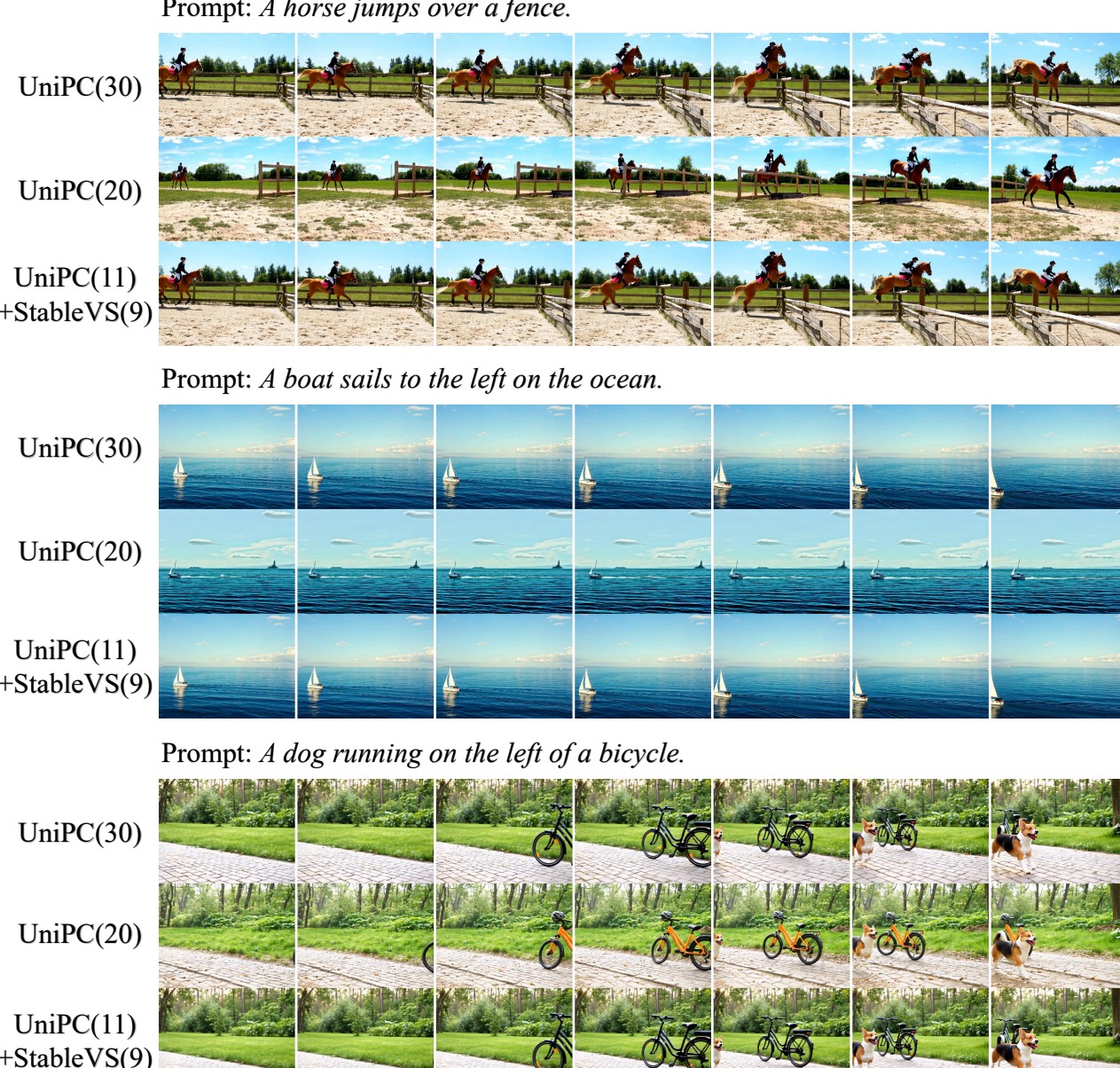

*Figure 12.* **Visual comparison of Wan2.2 (Wan et al., 2025) on different prompts.** Results are generated using UniPC solver with 30 and 20 steps, and with StableVS replacing UniPC in the *low-variance regime*, all under the same random seed. Compared to the standard 20-step solver, StableVS yields outputs that more closely resemble the 30-step results. Zoom in for details.

