# OpenReview forum: "Stable Velocity: A Variance Perspective on Flow Matching"
_ICML.cc/2026/Conference — ICML 2026 regular_

### Official Review · Reviewer_eBwN · 2026-03-12

**Soundness:** 2
**Presentation:** 3
**Significance:** 3
**Originality:** 3
**Overall Recommendation:** 4
**Confidence:** 3

**Summary:**

This paper analyzes the variance of velocity targets in flow matching, revealing a high-variance regime near noise and low-variance regime near data. They exploit this structure to propose variance-reduced training and accelerated inference methods that improve image and video generation quality

**Compliance With Llm Reviewing Policy:**

Affirmed.

**Final Justification:**

(a) Fully resolved - My concerns have been adequately addressed.

**Key Questions For Authors:**

1. The proof of Theorem 2 applies Jensen's inequality and then simplifies the upper bound to exactly $\mathcal{V}\_{\text{CFM}}(t)$, independent of $n$. This means for any $n \geq 1$, the theorem only guarantees $\mathcal{V}\_{\text{StableVM}}(t) \leq \mathcal{V}\_{\text{CFM}}(t)$ — the same bound whether you use 2 or 1000 reference samples. Could the authors clarify why this result is meaningful beyond stating that StableVM is "no worse" than CFM? In particular, is there a simple example or condition under which the bound is tight (i.e., equality holds for $n > 1$), or can you show that the inequality is always strict for $n > 1$?
2. Theorem 2 upper bounds $\mathcal{V}\_{\text{StableVM}}(t)$ by bounding the squared norm of a weighted average with the weighted average of squared norms (Jensen). In doing so, the cross-term structure of the StableVM estimator — which is precisely where variance reduction comes from — is entirely discarded. Given that the resulting bound does not depend on $n$ at all, could the authors comment on whether a tighter $n$-dependent bound is achievable for Theorem 2 without resorting to the asymptotic/delta-method machinery of Theorem 3? It seems the "weak" bound contributes little beyond what one would expect from a convexity argument, and the actual variance reduction story rests entirely on Theorem 3's asymptotic analysis.

**Limitations:**

- no training validation on models that requires large scale setup

**Strengths And Weaknesses:**

# Strengths

- Not applying REPA in the high-noise regime makes sense — the posterior becomes multimodal with no meaningful alignment target. Fig. 3 confirms alignment loss stays high there and restricting REPA to the low-variance regime gives the best FID.
- FID improves 1.98→1.80 over REPA (80 epochs), generalizes across scales and REPA variants. StableVS gives >2x step reduction on SD3.5/Flux/Qwen/Wan2.2 with negligible quality loss (PSNR 36.92 on SD3.5). Ablations are thorough.



# Weakness

- In the proof of Theorem 3 (Appendix C.3, the step from the Lemma 5 result to the final bound), the authors write "where we have used $p\_t(x\_t \mid x\_0) \leq 1$ in the second line" to upper-bound the ratio $p\_t(x\_t \mid x\_0)/p\_t(x\_t)$ by $1/p\_t(x\_t)$. However, $p\_t(x\_t \mid x\_0) = \mathcal{N}(x\_t; \alpha\_t x\_0, \sigma\_t^2 I)$ is a Gaussian density, not a probability mass. Its peak value is $(2\pi\sigma\_t^2)^{-d/2}$, which for the linear interpolant $\sigma\_t = t$ at, e.g., $t=0.5, d=4096$ equals $(2\pi \cdot 0.25)^{-2048} \approx 10^{1600}$. The inequality $p\_t(x\_t \mid x\_0) \leq 1$ is only valid for discrete distributions or CDFs. And it is generically false for continuous densities in high dimensions. This error propagates into the $\varepsilon$-splitting argument that follows, invalidating the stated bound. Notably, the expression before the erroneous step (in the appendix) is already a valid and clean result: $\mathcal{V}\_{\text{StableVM}}(t) = \frac{1}{n-1} \mathbb{E}\_{x\_t \sim p\_t}\left[\mathbb{E}\_{x\_0 \sim p\_t(\cdot|x\_t)}\left[\frac{p\_t(x\_t|x\_0)}{p\_t(x\_t)} \lVert v\_t(x\_t|x\_0) - v\_t(x\_t)\rVert^2\right]\right] + o(1/n)$, which can be taken as the final answer directly — no bounding by 1, no $\varepsilon$-splitting, no $M$ tail term needed. The corrected bound is simpler, requires fewer assumptions, and for Gaussian priors with linear interpolants, the reweighting factor $p\_t(x\_t|x\_0)/p\_t(x\_t)$ actually makes the bound tighter than $\frac{1}{n-1}\mathcal{V}\_{\text{CFM}}(t)$. Even if corrected, the Theorem 3 bound with $1/\varepsilon$ is vacuous in high dimensions. Setting aside the proof error, the structure of the Theorem 3 bound $\frac{1}{(n-1)\varepsilon}\mathcal{V}\_{\text{CFM}}(t) + \frac{M}{n-1}$ is problematic. The marginal density $p\_t(x\_t)$ in $\mathbb{R}^d$ has typical values of order $(2\pi\sigma\_t^2)^{-d/2}$. For the set $\lbrace x: p\_t(x) > \varepsilon\rbrace$ to capture most of the probability mass, $\varepsilon$ must be exponentially small in $d$, making the $1/\varepsilon$ factor exponentially large. There is a fundamental tension: smaller $\varepsilon$ grows $M$, while larger $\varepsilon$ inflates $1/\varepsilon$. The main text's informal claim that $\mathcal{V}\_{\text{StableVM}}(t) \lessapprox \frac{1}{n}\mathcal{V}\_{\text{CFM}}(t)$ is not supported by the formal bound as stated.

- Theorem 2 provides no information about variance reduction. It states $\mathcal{V}\_{\text{StableVM}}(t) \leq \mathcal{V}\_{\text{CFM}}(t)$. This bound has no dependence on $n$. It merely says the $n$-sample estimator is no worse than the 1-sample estimator. This is a trivial sanity check, not a demonstration of variance reduction. The looseness enters when Jensen's inequality is applied to the squared norm of the weighted average, which replaces the variance of an average with the average of variances, discarding all benefit from multi-sample aggregation. A meaningful result would establish an $O(1/n)$ rate, which the authors defer entirely to Theorem 3 (whose proof contains an error, see above).

- The paper states that in the low-variance regime, $x\_\tau = x\_t + (\tau - t)v\_t(x\_t)$ allows "exact integration via Euler steps of arbitrary size." This is only true if $v\_t(x\_t) = v\_t(x\_t \mid x\_0)$ exactly, i.e., the posterior is a delta function. No formal error bound is provided for the approximation $v\_t(x\_t) \approx v\_t(x\_t \mid x\_0)$, so the quality of the step-skipping and the choice of $\xi$ remain entirely heuristic.

---

> ### Author Rebuttal · Authors · 2026-03-31
>
> > **Q1.** Error in the proof of Theorem 3.3
> >
>
> **A1.**  We thank the reviewer for catching this issue. You are right that one cannot assume $p\_t(x\_t\\mid x\_0)\\le 1$ in general. This only affects the last step of the proof of Theorem 3.3; the preceding derivation remains valid. We will therefore replace Theorem 3.3 by the corrected statement:
>
> $$\\mathcal{V}\_{\\mathrm{StableVM}}(t)=\\frac{1}{n-1}\\left(\\mathcal{V}\_{\\mathrm{CFM}}(t)+\\mathbb{E}\\big[(r\_t(x\_t,x\_0)-1)\\Delta\_t(x\_t,x\_0)\\big]\\right)+o(1/n),$$
>
> where
>
> $$
> r\_t(x\_t,x\_0):=\\frac{p\_t(x\_0\\mid x\_t)}{q(x\_0)},
> \\qquad
> \\Delta\_t(x\_t,x\_0):=
> \\|v\_t(x\_t\\mid x\_0)-v\_t(x\_t)\\|^2.
> $$
> The proof should be immediate from the line right before the incorrect simplification in the appendix.
>
> We observe that this correction term is almost negligible for $t<0.5$ and as $t\to 1$ for CIFAR-10. Intuitively, for early timesteps, we are in the low-variance regime, so $\Delta_t({x}_t,{x}_0)$ should be small. As $t\to 1$, we are in the high-variance regime, so ${x}_t$ carries little information about ${x}_0$ and the conditional distribution $p_t({x}_0\mid {x}_t)$ approaches the unconditional $q({x}_0)$. The ratio $r_t({x}_t,{x}_0)$ would then be close to 1. For the intermediate timesteps, neither effect is dominant, so the correction term may peak.
>
> | Timestep $t$ | $\mathbb{E}[(r_t-1)\Delta_t] / \sqrt{d}$|
> | --- | --- |
> | 0.01 | 6.48e-07 |
> | 0.50 | 1.52e-08 |
> | 0.70 | 596 |
> | **0.84** | **19,186** |
> | 0.96 | 41.0 |
> | 0.999 | 6.1e-04 |
>
> > **Q2.** For Theorem 3.2, can you show strict inequality for $n>1$? Is there a tighter $n$-dependent bound without asymptotic/delta-method machinery?
>
> **A2.** We agree that Theorem 3.2 is qualitative rather than a quantitative $n$-rate result. Its role is to give an assumption-free guarantee that StableVM is never worse than CFM.
> Nevertheless, we did find a very simple condition to guarantee strict inequality. In particular, for affine conditional flows, if the event $\\{x\_0^1=\\cdots=x\_0^n\\}$ has probability 0 (which is a reasonable assumption if $n$ is large), then $\\mathcal{V}\_{\\text{StableVM}}(t) < \\mathcal{V}\_{\\text{CFM}}(t)$ strictly. Due to space limit, we omit the proof here, but this is a simple consequence of the equality condition of Jensen's inequality with the convex function being $\\|\\cdot\\|^2$. This shows that we truly have a variance reduction when using StableVM, and Theorem 3.3 shows a stronger result that the variance reduction scales at the rate of $O(1/n)$.
>
> At the same time, we do not believe a universal tighter $n$-dependent bound can be obtained from Theorem 3.2 alone. Without extra assumptions, the normalized likelihood weights can become highly concentrated on one reference sample, in which case StableVM becomes arbitrarily close to the one-sample CFM target for fixed $n$. Thus Theorem 3.2 is essentially sharp as a distribution-free result; any stronger $n$-dependent rate requires additional structure, which is exactly the role of the corrected Theorem 3.3.
>
> > **Q3.** The "exact integration via Euler steps of arbitrary size" claim in the low-variance regime lacks a formal error bound.
>
> **A3.** The statement that ${x}_\tau = {x}_t + (\tau - t)\,{v}_t({x}_t)$ holds only under the specific linear scheduler $\alpha_t = 1 - t, \sigma_t = t$, where the velocity field becomes locally time-invariant in the low-variance regime. For general schedulers, we do not claim arbitrary-step euler exactness; instead, we rely on Eq. (14–15) for shortcut sampling, which still enable large-step integration but are not strictly "exact Euler" steps. We will clarify this distinction in the revision.
>
> We agree that the approximation ${v}_t({x}_t) \approx {v}_t({x}_t\mid{x}_0)$ is not supported by a formal error bound. Our justification is mainly empirical:
>
> - In Sec. 2, we show that for a substantial region ($t < \xi$), the conditional variance is consistently very low, indicating that the posterior is sharply concentrated and the conditional and marginal velocity fields are close.
> - This directly motivates treating the dynamics in this region as effectively deterministic, enabling stable large-step updates.
>
> Although a formal bound is intractable due to the unknown data distribution, our experiments provide consistent supporting evidence:
>
> - Tab. 4 and Tab. 9 validate the existence of a clear two-regime structure.
> - Choosing $\xi \approx 0.7$ consistently places the split within the low-variance regime, where shortcut sampling preserves quality.
> - When $\xi$ moves outside this regime, performance degrades significantly, confirming that the approximation breaks down.
>
> Importantly, the choice of $\xi$ is not entirely unconstrained. Based on the dimension-dependent shift discussed in the paper:
>
> - Higher-dimensional data (e.g., T2I/T2V) exhibit a larger low-variance region, implying $\xi > 0.7$.
> - This significantly reduces the search space and makes the method practical without exhaustive tuning.

---

> > ### Author Rebuttal · Reviewer_eBwN · 2026-04-01
> >
> > please remember to update the proof in the next version

---

> > > ### Author Response · Authors · 2026-04-02
> > >
> > > Thank you for your careful review and constructive comments. We will revise the proof accordingly in the updated version.

---

### Official Review · Reviewer_XCu3 · 2026-03-13

**Soundness:** 3
**Presentation:** 4
**Significance:** 3
**Originality:** 3
**Overall Recommendation:** 5
**Confidence:** 3

**Summary:**

The paper analyzes variance in Conditional Flow Matching (CFM) training and proposes Stable Velocity incorporating StableVM and VA-REPA at the training stage and StableVS at the inference stage. Extensive experiments are conducted to validate the effectiveness of the proposed method across multiple benchmarks.

**Compliance With Llm Reviewing Policy:**

Affirmed.

**Final Justification:**

After taking all the reviews and the rebuttal into consideration, I would like to maintain my score.

**Key Questions For Authors:**

1. Is it possible to provide more ablation results on different split point settings?
2. Why is the split point set to 0.7 for StableVM and VA-REPA, and 0.85 for StableVS?

**Limitations:**

I think this is a great work with clear motivation, and the proposed method proves to be effective across multiple benchmarks. The primary limitation is that inevitable manual settings on the hyperparameter of the split point and the weighting function are introduced, which may require some post adjustment during the real-world applications.

**Strengths And Weaknesses:**

Strength:
1. The authors conduct extensive and comprehensive experiments across multiple tasks, including image generation, text2image, and text2video generations on multiple state-of-the-art baselines, which strongly validate the effectiveness of the proposed method.
2. The paper is well-written with clear textual expression and intuitive visualization figures, making it easy for readers to understand.
3. The target of this research is to tackle the challenge that the training target can have a large variance given one reference sample, especially when $x_t$ is close to the prior distribution, which is an important problem in various conditional generation subtasks.
4. The proposed method proposes stableVM along with VA-REPA at the training stage and StableVS at the inference stage, which are straightforward and novel approaches to tackle the problem.

Weakness:
1. The split point still requires manual adjustment, which may somehow limit its performance in real-world applications.
2. The weighting functions involve multiple factors and the setting of the split point, which may introduce sensitivity to the hyperparameter.

---

> ### Author Rebuttal · Authors · 2026-03-31
>
> We thank the reviewer for the positive evaluation and for recognizing the clarity and contributions of the paper.
>
> > **Q1.** The split point still requires manual adjustment.
> >
>
> **A1.** We agree that the split point $\xi$ is a hyperparameter. However, it is not entirely unconstrained and can be chosen based on a principled heuristic derived from our analysis.
>
> Specifically, while the exact split point depends on the (unknown) data distribution, our theory predicts a dimensionality-dependent shift in the boundary between the low-variance and high-variance regimes. For higher-dimensional data (e.g., high-resolution T2I or T2V models), the split point is expected to move closer to 1. In practice, this leads to a simple and effective guideline:
>
> - Since CIFAR-10 empirically yields an optimal split around $\xi \approx 0.7$,
> - for higher-dimensional settings, one can restrict the search to $\xi > 0.7$.
>
> Our ablation in Tab. 9 supports this claim: choosing $\xi=0.7$ yields almost identical results (PSNR = 43.65) compared to the original 30-step sampler, indicating that the true split point for higher-dimensional data is indeed larger than that of CIFAR-10.
>
> Therefore, rather than requiring extensive tuning, our analysis substantially narrows the feasible range of $\xi$, making the method practical and robust in real-world settings. In practice, we fix $\xi = 0.7$ across all experiments unless otherwise noted, and observe consistently strong performance. We will clarify this heuristic in the revision to better guide practitioners.
>
> > **Q2.** Is it possible to provide more ablation results on different split point settings? Why is the split point set to 0.7 for StableVM and VA-REPA, and 0.85 for StableVS?
> >
>
> **A2.** We thank the reviewer for the question regarding the choice of the split point $\xi$. We would like to clarify that we have already provided detailed ablations on this hyperparameter in Tab. 4 (VA-REPA) and Tab. 9 (StableVS).
>
> For VA-REPA, the optimal split point depends mildly on the training stage:
>
> - At early training (100k iterations), a smaller split point ($\xi = 0.6$) achieves the best FID. This suggests that weaker alignment (i.e., focusing more on the low-variance regime) provides an easier supervisory signal and accelerates early convergence.
> - As training progresses, $\xi = 0.7$ consistently yields the best performance at 400k iterations.
> - A larger split point ($\xi = 0.8$) degrades performance, which we attribute to noisy supervision from the high-variance regime.
>
> For StableVS, we observe a similar trend:
>
> - $\xi = 0.7$ achieves the best reconstruction quality (PSNR = 43.65), indicating that selecting the split point within the low-variance regime enables accurate shortcut sampling (i.e., matching the full 30-step results).
> - When the split point moves into the high-variance regime (e.g., $\xi = 0.9$), performance drops significantly (PSNR = 31.99).
>
> Regarding the use of $\xi = 0.85$ in some StableVS experiments, this choice is made solely for fair comparison with the 20-step baseline setting, rather than performance tuning.
>
> Overall, these ablations consistently support our theoretical analysis in Sec. 2–3: there exists a two-regime structure in flow matching, and the split point should be placed around $\xi \approx 0.7$ to effectively exploit the low-variance regime for both stable supervision (VA-REPA) and efficient sampling (StableVS).

---

> > ### Author Rebuttal · Reviewer_XCu3 · 2026-04-04
> >
> > My concerns are well addressed.

---

### Official Review · Reviewer_NVq8 · 2026-03-13

**Soundness:** 3
**Presentation:** 3
**Significance:** 2
**Originality:** 3
**Overall Recommendation:** 4
**Confidence:** 3

**Summary:**

This paper presents a variance-centric framework for flow matching and stochastic interpolants, identifying a two-regime variance structure that governs the generative trajectory. To mitigate the training instabilities caused by single-sample Monte Carlo velocity targets, the authors introduce stable velocity matching, an unbiased variance-reduced objective, alongside variance-aware representation alignment, which adaptively applies auxiliary semantic supervision where signal-to-noise ratios are highest. For inference, they derive stable velocity sampling, which exploits the effectively deterministic nature of the low-variance regime to enable significant sampling acceleration without additional finetuning.

**Compliance With Llm Reviewing Policy:**

Affirmed.

**Key Questions For Authors:**

* How does the optimal split point $\xi$ vary across different noise schedules or interpolants beyond the standard linear case?
* Can you provide a more granular breakdown of the training wall-clock time overhead introduced by StableVM as the batch size or memory bank capacity $K$ scales up?
* In Algorithm 2, how does the FIFO update policy for the class-conditional memory bank compare to other sampling strategies in terms of maintaining target diversity during long-duration training?
* The exact split point $\xi$ is dependent on the underlying data distribution. Does this necessitate per-task calibration to avoid noisy supervision or integration errors?

**Limitations:**

yes

**Strengths And Weaknesses:**

Strengths:
* The framework is successfully applied to a diverse array of recent, large-scale models across both image and video modalities, such as Flux, Qwen-Image, and Wan2.2.
* StableVS offers a notable inference speedup (over 2x) by leveraging the closed-form simplifications of the low-variance regime without the need for expensive distillation or retraining.

Weaknesses:
* StableVM introduces extra memory requirements proportional to the bank capacity $K$ and added computation for self-normalized targets .
* The efficacy of the framework relies on the selection of the split point $\xi$, which requires empirical tuning and differs between the training and sampling stages.
* While the paper notes that the split point $\xi$ shifts with dimensionality, more guidance on how to automatically determine this point for arbitrary new datasets would enhance the method's robustness.

---

> ### Author Rebuttal · Authors · 2026-03-31
>
> > **Q1.** How does the optimal split point vary across different noise schedules or interpolants beyond the standard linear case?
> >
>
> **A1.** We thank the reviewer for this question. Our variance analysis of the split point (Fig. 1) are indeed derived under the commonly adopted linear interpolant. However, this choice is made for analytical simplicity, and the result is not restricted to this specific schedule.
>
> As discussed in Sec. 4.9.2 of [1], affine conditional flows admit closed-form transformations between different noise schedules. This implies that, for a fixed data coupling, one can map velocity fields from the linear scheduler to another scheduler via a deterministic time reparameterization. More importantly, Eq. (4.65) in [1] shows that velocities under two schedulers are related through a linear transformation with a time reparameterization defined via the SNR, i.e., $t_r = \rho^{-1}(\bar{\rho}(r))$, which induces a monotone correspondence between the two time parameterizations. This transformation further implies that the variance is scaled by a positive, smoothly varying function of time. While this may change the absolute magnitude of the variance, it does not alter the two-regime structure under standard linear schedulers.
>
> Consequently, the split point transfers consistently under such transformations. Since both the variance ordering and the time parameterization are preserved, the mapped split point remains optimal under the new schedule. In this sense, the split point reflects an intrinsic property rather than a particular choice of time parameterization.
>
> > **Q2.** Can you provide a more granular breakdown of the training wall-clock time overhead introduced by StableVM?
> >
>
> **A2.** We thank the reviewer for raising the question regarding training efficiency. We provide a direct comparison of throughput with the baseline under the same hardware setting (1 node with 4× NVIDIA L40S):
>
> - The original REPA implementation achieves 1.22 iters/s.
> - Our implementation, using StableVM (bank = 256) together with VA-REPA, runs at 1.02 iters/s.
>
> This corresponds to an approximate 16% reduction in throughput. The overhead primarily comes from StableVM, which requires computing a stabilized target over a reference batch (memory bank). Importantly, this cost is modular:
>
> - VA-REPA alone (without StableVM) introduces negligible overhead and runs at essentially the same speed as vanilla REPA.
> - The additional cost is therefore isolated to StableVM and can be adjusted via the memory bank size. The additional cost scales linearly with memory bank size $K$, since StableVM requires $\mathcal{O}(K)$ aggregation per step.
>
> > **Q3.** How does the FIFO update policy compare to other sampling strategies in terms of maintaining target diversity during training?
> >
>
> **A3.** The FIFO update policy is designed to balance **sample diversity** and **temporal relevance** of the target distribution during training.
>
> From a statistical perspective, the FIFO memory bank can be viewed as a **sliding-window Monte Carlo estimator**. While it does not maintain a uniform sample over the entire training history, it approximates the data distribution over a recent window, which empirically suffices for stable training.  Empirically, Figure 4 shows that a relatively small buffer ($K=256$) achieves performance comparable to a near-static large buffer ($K=1024$), where the latter effectively stores almost the entire dataset and receives minimal updates. This indicates that FIFO serves as an effective and computationally efficient proxy for the target distribution.
>
> > **Q4.** The exact split point is dependent on the underlying data distribution. Does this necessitate per-task calibration to avoid noisy supervision or integration errors?
> >
>
> **A4.** We agree that the exact split point $\xi$ is intractable and depends on the underlying data distribution. However, it is not entirely unconstrained and can be chosen based on a principled heuristic derived from our analysis. In Sec. 2, our theory predicts a dimensionality-dependent shift of the split point $\xi$. For higher-dimensional data (e.g., high-resolution T2I or T2V models), the split point is expected to move closer to 1. In practice, this leads to a simple guideline: Since CIFAR-10 empirically yields an split around $\xi \approx 0.7$, for higher-dimensional settings, one can restrict the search to $\xi > 0.7$.
>
> Our ablation in Tab. 9 supports this claim: choosing $\xi=0.7$ yields almost identical results (PNSR = 43.65) compared to the original 30-step sampler, indicating that the true split point for higher-dimensional data is indeed larger than that of CIFAR-10.
>
> Therefore, our analysis narrows the feasible range of $\xi$, making the method practical in real-world settings. We will clarify this heuristic in the revision to better guide practitioners.
>
> ### References
>
> [1] Lipman, Yaron, et al. *Flow matching guide and code.* arXiv preprint arXiv:2412.06264 (2024).

---

### Official Review · Reviewer_qyuc · 2026-03-14

**Soundness:** 3
**Presentation:** 3
**Significance:** 3
**Originality:** 3
**Overall Recommendation:** 4
**Confidence:** 4

**Summary:**

This paper conducts a variance analysis of Conditional Flow Matching (CFM) training and reveals, based on empirical observations, an interesting two-regime structure: (1) a high-variance regime near the prior distribution, and (2) a low-variance regime near the data distribution.
Based on this observation, the paper proposes three modifications to the CFM framework to improve training performance and sampling efficiency.
Stable Velocity Matching (StableVM). The authors argue that the high variance in CFM arises because the training target is effectively a single-sample Monte Carlo estimate of the marginal velocity. To address this issue, they propose the StableVM target, which replaces the single-sample estimator with a multi-sample estimator constructed using Gaussian mixture models (GMM) and reference samples. Theoretical justification is provided (Theorem 3.2), showing that the variance of StableVM is guaranteed to be smaller in theory than that of the original CFM objective.

Variance-Aware Representation Alignment (VA-REPA). The authors empirically observe that the effectiveness of REPA mainly arises in the low-variance regime. Based on this observation, they propose applying representation alignment selectively by introducing a variance-aware weighting scheme, which focuses REPA on the low-variance region.

Stable Velocity Sampling (StableVS). The paper further proposes StableVS, a sampling acceleration strategy designed for the low-variance regime, where the dynamics can be simplified to allow larger integration steps without degrading generation quality.

Extensive experimental results are provided to demonstrate the effectiveness of the proposed methods, showing consistent improvements in generation performance across multiple benchmarks.

**Compliance With Llm Reviewing Policy:**

Affirmed.

**Key Questions For Authors:**

1) Sensitivity to hyperparameters. Many of the reported improvements appear relatively modest, while the proposed framework introduces several additional hyperparameters, such as the split point, the number of steps used in StableVS, the memory bank capacity  and the weighting function. It is therefore unclear how robust the reported improvements are with respect to these hyperparameter choices. The authors are encouraged to perform a more detailed ablation study focusing on a few of the most significant improvements reported in the tables. In particular, it would be helpful to show how the performance varies when these hyperparameters are changed, in order to demonstrate that the observed gains are stable and not sensitive to specific hyperparameter selections.

2) Fairness of epoch comparisons in Table 1. In Table 1, the comparisons at 80 epochs appear fair across methods. However, the results reported for the proposed method at 240 epochs do not have a corresponding comparison with REPA-E trained for the same number of epochs. This makes it difficult to directly assess the relative performance of the methods. A clearer comparison would be to report results for all methods at matched training durations, or alternatively provide performance curves versus training epochs for each method. Such plots would make the comparison more transparent and help readers understand the relative convergence behavior of the approaches.

3). It is not clear whether  the training time of Stable Velocity Matching is significantly increased.  A comparison of the training time to CFM will be helpful to clarify the significance of the proposed new method. Also although the variance reduction is justified theoretically, empirical evidence is not provided. A figure similar to Figure 1 to show the variance curves of the new method would be useful.

**Limitations:**

The major weakness of this paper includes: 1)  the performance improvements are not very significant; 2) The additional components introduced by Stable Velocity Matching (e.g., multi-sample estimators and memory bank) may increase training cost, but the paper does not provide a clear comparison of training time or computational complexity relative to CFM; 3) Although theoretical results show variance reduction, the paper does not provide direct empirical measurements (e.g., variance curves analogous to Figure 1) to verify the practical impact of the proposed estimator.

**Strengths And Weaknesses:**

1) Clear and insightful variance perspective. The paper provides a useful analysis of variance in Conditional Flow Matching and identifies a meaningful two-regime structure (high variance near the prior and low variance near the data), which offers a helpful perspective for understanding training dynamics of CFM.

2) Variance reduction method. The proposed StableVM introduces a multi-sample estimator that theoretically reduces the variance of the training target.

3) Improvements to both training stability and sampling efficiency. The paper proposes a coherent framework consisting of StableVM, VA-REPA, and StableVS, addressing both training stability and sampling efficiency.

---

> ### Author Rebuttal · Authors · 2026-03-31
>
> > **Q1.** It would be helpful to show how the performance varies when these hyperparameters are changed, in order to demonstrate that the observed gains are stable.
> >
>
> **A1.** We appreciate the reviewer’s concern regarding sensitivity to hyperparameters and the robustness of the reported gains. We would like to clarify that the paper already includes a set of targeted ablation studies covering the key hyperparameters:
>
> - Split point: analyzed in Tab. 4 (VA-REPA, $\xi=0.6,0.7,0.8$) and Tab. 9 (StableVS, $\xi=0.7,0.8,0.85,0.9$)
> - Memory bank capacity and weighting function: Fig. 4 ($K=256,1024;w_{\text{hard}},w_{\text{sigmoid}},w_{\text{SNR}}$)
> - Number of steps in the low-variance regime: Tab. 9 (steps: 4, 9, 14)
> - Variance factor $f_\beta$: Tab. 9 ($f_\beta=0.0,0.1,0.2$)
>
> Across all these studies, we observe consistent and stable improvements of StableVM and VA-REPA over the vanilla REPA baseline across a broad range of hyperparameter choices, rather than only at finely tuned settings. In particular, performance varies smoothly with respect to these parameters, and no sharp degradation is observed, indicating that the method is not overly sensitive to hyperparameter selection.
>
> Moreover, the ablation on the split point provides additional insight: it empirically validates the two-regime structure of flow matching discussed in Sec. 2, which is the core motivation of our method. The fact that improvements persist across different split locations further supports that the gains are structural rather than incidental.
>
> > **Q2.** Fairness of epoch comparisons in Table 1. In Table 1, the comparisons at 80 epochs appear fair across methods. However, the results at 240 epochs do not have a corresponding comparison with REPA-E.
> >
>
> **A2.** We thank the reviewer for pointing out the importance of fair comparisons across matched training durations. To address this concern, we provide additional results with longer training schedules:
>
> | Method | Epochs | FID | IS |
> | --- | --- | --- | --- |
> | REPA | 80 | 1.98 | 263 |
> | REPA | 800 | 1.42 | 305.7 |
> | REG | 80 | 1.86 | 321.4 |
> | REG | 480 | 1.4 | 296.9 |
> | REPA-E | 80 | 1.67 | 266.3 |
> | REPA-E | 800 | 1.12 | 302.9 |
> | Ours | 80 | 1.71 | 274.2 |
> | Ours | 480 | 1.33 | 307.8 |
>
> Both REPA-E and our method adopt class-balanced sampling. At 80 epochs, all methods are directly comparable, and our method already achieves competitive performance. At 480 epochs, our method achieves the best IS and competitive FID, outperforming REG and REPA, and approaching the performance of REPA-E trained significantly longer**. It is important to note that REPA-E requires end-to-end fine-tuning of both the autoencoder and diffusion transformer, which is substantially more computationally expensive**. In contrast, our method achieves strong performance without such full-model fine-tuning.
>
> > **Q3.** It is not clear whether the training time of Stable Velocity Matching is significantly increased.
> >
>
> **A3.** We thank the reviewer for raising the question regarding training efficiency. We provide a direct comparison of throughput with the baseline under the same hardware setting (1 node with 4× NVIDIA L40S):
>
> - The original REPA implementation achieves 1.22 iters/s.
> - Our implementation, using StableVM (bank = 256) together with VA-REPA, runs at 1.02 iters/s.
>
> This corresponds to an approximate 16% reduction in throughput. The overhead primarily comes from StableVM, which requires computing a stabilized target over a reference batch (memory bank). Importantly, this cost is modular:
>
> - VA-REPA alone (without StableVM) introduces negligible overhead and runs at essentially the same speed as vanilla REPA.
> - The additional cost is therefore isolated to StableVM and can be adjusted via the memory bank size.
>
> Overall, we emphasize that the slowdown is moderate and predictable. We will include these throughput comparisons in the revision for clarity.
>
> > **Q4.** Also although the variance reduction is justified theoretically, empirical evidence is not provided.
> >
>
> **A4.** We thank the reviewer for the helpful suggestion. To complement our theoretical variance analysis, we provide additional empirical evidence by directly measuring the variance $V_{\text{StableVM}}$ (with $n=2048$) and comparing it against $\mathcal{V}_{\text{CFM}}$.
>
> **GMM (d = 500, high-variance regime):**
>
> | **Timestep** | **0.87** | **0.89** | **0.91** | **0.93** | **0.95** | **0.97** | **0.99** |
> | --- | --- | --- | --- | --- | --- | --- | --- |
> | $\mathcal{V}_{\text{CFM}}/\sqrt{d}$ | 10.12 | 10.75 | 11.24 | 11.72 | 12.3 | 13.03 | 13.92 |
> | $\mathcal{V}_{\text{StableVM}}/\sqrt{d}$ | **0.53** | **0.86** | **1.36** | **2.05** | **2.93** | **3.99** | **5.22** |
>
> We observe a consistent and substantial reduction in variance often by an order of magnitude. These results are fully consistent with our theoretical analysis. We will incorporate variance curves in the final version to better visualize the trend across timesteps.

---

> > ### Author Rebuttal · Reviewer_qyuc · 2026-04-07
> >
> > My concerns are well addressed.

---

### Decision · Program_Chairs · 2026-04-30

**Decision:**

Accept (regular)

**Comment:**

This paper proposes Stable Velocity, an extension of standard flow matching in which the velocity target in the loss is computed in terms of a self-normalized estimate computed from a set of data points in a memory bank. This approach is informed by the (in itself not novel) observation that the variance of the velocity field (which arises from the variance in the posterior p_t(x_0 | x_t) of the probability path) varies with time, being highest early during generation near the prior, at t=1 in the notation of the paper, and lowest near t=0 close to the data). This perspective informs two secondary contributions: Variance-aware representation alignment (VA-REPA), which applies REPA selectively in the low-variance regime and Stable velocity sampling (StableVS), a finetuning-free method for switching from ODE integration in the high-variance regime to larger steps in the low-variance regime. Empirical validation extends beyond ImageNet to large pretrained image and video models, which supports the practical relevance of the approach.

Reviewers are uniformly positive about the submission. Reviews uncovered some issues with the theoretical exposition which authors addressed in the response. There seem to be no major glaring issues. Overall this is a solid contribution that leverages (not entirely novel) insights about the the time dependence of the posterior variance of the probability path to identify improvements to flow matching training and generation.